# SCHEDULED RESTART MOMENTUM FOR ACCELERATED STOCHASTIC GRADIENT DESCENT

## ABSTRACT

Stochastic gradient descent (SGD) algorithms, with constant momentum and its variants such as Adam, are the optimization methods of choice for training deep neural networks (DNNs). There is great interest in speeding up the convergence of these methods due to their high computational expense. Nesterov accelerated gradient (NAG) with a time-varying momentum, denoted as NAG below, improves the convergence rate of gradient descent (GD) for convex optimization using a specially designed momentum; however, it accumulates error when an inexact gradient is used (such as in SGD), slowing convergence at best and diverging at worst. In this paper, we propose *scheduled restart SGD* (SRSGD), a new NAG-style scheme for training DNNs. SRSGD replaces the constant momentum in SGD by the increasing momentum in NAG but stabilizes the iterations by resetting the momentum to zero according to a schedule. Using a variety of models and benchmarks for image classification, we demonstrate that, in training DNNs, SRSGD significantly improves convergence and generalization; for instance, in training ResNet-200 for ImageNet classification, SRSGD achieves an error rate of 20.93% vs. the benchmark of 22.13%. These improvements become more significant as the network grows deeper. Furthermore, on both CIFAR and ImageNet, SRSGD reaches similar or even better error rates with significantly fewer training epochs compared to the SGD baseline.

## 1 INTRODUCTION

Training many machine learning (ML) models reduces to solving the following finite-sum optimization problem

$$\min_{\boldsymbol{w}} f(\boldsymbol{w}) := \min_{\boldsymbol{w}} \frac{1}{N} \sum_{i=1}^{N} f_i(\boldsymbol{w}) := \min_{\boldsymbol{w}} \frac{1}{N} \sum_{i=1}^{N} \mathcal{L}(g(\boldsymbol{x}_i, \boldsymbol{w}), y_i), \ \ \boldsymbol{w} \in \mathbb{R}^d, \tag{1}$$

where $\{\boldsymbol{x}_i, y_i\}_{i=1}^{N}$ are the training samples and $\mathcal{L}$ is the loss function, e.g., cross-entropy loss for a classification task, that measure the discrepancy between the ground-truth label $y_i$ and the prediction by the model $g(\cdot, \boldsymbol{w})$, parametrized by $\boldsymbol{w}$. The problem (1) is known as *empirical risk minimization* (ERM). In many applications, $f(\boldsymbol{w})$ is non-convex, and $g(\cdot, \boldsymbol{w})$ is chosen among deep neural networks (DNNs) due to their preeminent performance across various tasks. These deep models are heavily overparametrized and require large amounts of training data. Thus, both $N$ and the dimension of $\boldsymbol{w}$ can scale up to millions or even billions. These complications pose serious computational challenges.

One of the simplest algorithms to solve (1) is gradient descent (GD), which updates $\boldsymbol{w}$ according to:

$$\boldsymbol{w}^{k+1} = \boldsymbol{w}^k - s_k \frac{1}{N} \sum_{i=1}^{N} \nabla f_i(\boldsymbol{w}^k), \tag{2}$$

where $s_k > 0$ is the step size at the $k$-th iteration. Computing $\nabla f(\boldsymbol{w}^k)$ on the entire training set is memory intensive and often prohibitive for devices with limited random access memory (RAM) such as graphics processing units (GPUs) used for deep learning (DL). In practice, we sample a subset of the training set, of size $m$ with $m \ll N$, to approximate $\nabla f(\boldsymbol{w}^k)$ by the mini-batch gradient $1/m \sum_{j=1}^{m} \nabla f_{i_j}(\boldsymbol{w}^k)$, resulting in the (mini-batch)-stochastic gradient descent (SGD). SGD and its

accelerated variants are among the most used optimization algorithms in ML. These gradient-based algorithms have low computational complexity, and they are easy to parallelize, making them suitable for large scale and high dimensional problems (Zinkevich et al., 2010; Zhang et al., 2015).

Nevertheless, GD and SGD have issues with slow convergence, especially when the problem is ill-conditioned. There are two common techniques to accelerate GD and SGD: adaptive step size (Duchi et al., 2011; Hinton et al.; Zeiler, 2012) and momentum (Polyak, 1964). The integration of both adaptive step size and momentum with SGD leads to Adam (Kingma & Ba, 2014), one of the most used optimizers for training DNNs. Many recent developments have improved Adam (Reddi et al., 2019; Dozat, 2016; Loshchilov & Hutter, 2018; Liu et al., 2020). GD with constant momentum leverages the previous step to accelerate GD according to:

$$\boldsymbol{v}^{k+1} = \boldsymbol{w}^k - s_k \nabla f(\boldsymbol{w}^k); \ \ \boldsymbol{w}^{k+1} = \boldsymbol{v}^{k+1} + \mu(\boldsymbol{v}^{k+1} - \boldsymbol{v}^k), \tag{3}$$

where $\mu > 0$ is a constant. A similar acceleration can be achieved by the heavy-ball (HB) method (Polyak, 1964). The momentum update in both (3) and HB have the same convergence rate of $O(1/k)$ as that of GD for convex smooth optimization. A breakthrough due to Nesterov (1983; 2018) replaces $\mu$ with $(k-1)/(k+2)$, which is known as the Nesterov accelerated gradient (NAG) with time-varying momentum. For simplicity, we denote this method as NAG below. NAG accelerates the convergence rate to $O(1/k^2)$, which is optimal for convex and smooth loss functions (Nesterov, 1983; 2018). NAG can also speed up the process of escaping from saddle points (Jin et al., 2017). In practice, NAG momentum can accelerate GD for nonconvex optimization, especially when the underlying problem is poorly conditioned (Goh, 2017). However, NAG accumulates error and causes instability when the gradient is inexact (Devolder et al., 2014; Assran & Rabbat, 2020). In many DL applications, constant momentum achieves state-of-the-art result. For instance, training DNNs for image classification. Since NAG momentum achieves a much better convergence rate than constant momentum with exact gradient for general convex optimization, we consider the following question:

*Can we leverage NAG with a time-varying momentum parameter to accelerate SGD in training DNNs and improve the test accuracy of the trained models?*

**Contributions.** We answer the above question by proposing the first algorithm that integrates scheduled restart NAG momentum with plain SGD. Here, we restart the momentum, which is orthogonal to the learning rate restart (Loshchilov & Hutter, 2016). We name the resulting algorithm ***scheduled restart SGD (SRSGD)***. *Theoretically, we prove the error accumulation of Nesterov accelerated SGD (NASGD) and the convergence of SRSGD.* The major practical benefits of SRSGD are fourfold:

- SRSGD remarkably speeds up DNN training. For image classification, SRSGD *significantly reduces the number of training epochs while preserving or even improving the network's accuracy*. In particular, on CIFAR10/100, the number of training epochs is reduced by half with SRSGD, while on ImageNet the reduction in training epochs is also remarkable.
- DNNs trained by SRSGD *generalize significantly better* than the current benchmark optimizers. *The improvement becomes more significant as the network grows deeper* as shown in Fig. 1.
- SRSGD *reduces overfitting in training very deep networks* such as ResNet-200 for ImageNet classification, enabling the accuracy to keep increasing with depth.
- SRSGD is *straightforward to implement* and only requires changes in a few lines of the SGD code. There is also no additional computational or memory overhead.

*We focus on image classification with DNNs, in which SGD with constant momentum is the choice.*

**Related Work.** Momentum has long been used to accelerate SGD. SGD with scheduled momentum and a good initialization can handle the curvature issues in training DNNs and enable the trained models to generalize well (Sutskever et al., 2013). Kingma & Ba (2014) and Dozat (2016) integrated momentum with adaptive step size to accelerate SGD. In this work, we study the time-varying momentum version of NAG with restart for stochastic optimization. Adaptive and scheduled restart have been used to accelerate NAG with the exact gradient (Nemirovskii & Nesterov, 1985; Nesterov, 2013; Iouditski & Nesterov, 2014; Lin & Xiao, 2014; Renegar, 2014; Freund & Lu, 2018; Roulet et al., 2015; O'donoghue & Candes, 2015; Giselsson & Boyd, 2014; Su et al., 2014). These studies of restart NAG momentum are for convex optimization with the exact gradient. Restart techniques have also been used for stochastic optimization (Kulunchakov & Mairal, 2019). In particular, Aybat et al. (2019) developed a multistage variant of NAG with momentum restart between stages. Our work focuses on developing NAG-based optimization for training DNNs. Many efforts have also been

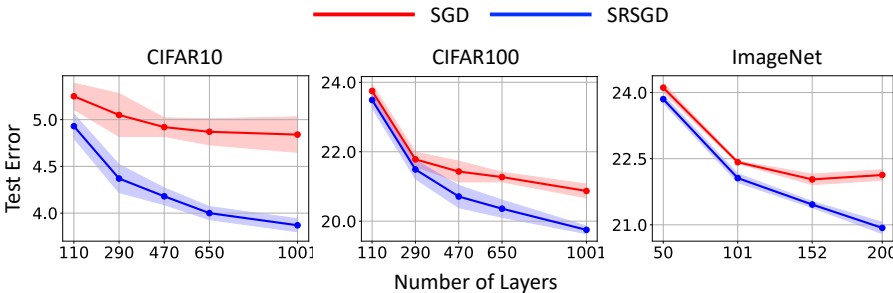

Figure 1: Error rate vs. depth of ResNet models trained with SRSGD and the baseline SGD with constant momemtum. Advantage of SRSGD continues to grow with depth.

devoted to studying the non-acceleration issues of SGD with HB and NAG momentum (Kidambi et al., 2018; Liu & Belkin, 2020), as well as accelerating first-order algorithms with noise-corrupted gradients (Cohen et al., 2018; Aybat et al., 2018; Lan, 2012). Ghadimi & Lan (2013; 2016) provides analysis for the general stochastic gradient-based optimization algorithms. .

**Organization.** In Section 2, we review and discuss momentum for accelerating GD for convex smooth optimization. In Section 3, we present the SRSGD algorithm and its theoretical guarantees. In Section 4, we verify the efficacy of the proposed SRSGD in training DNNs for image classification on CIFAR and ImageNet. In Section 4.3, we perform empirical analysis of SRSGD. We end with some concluding remarks. Technical proofs, some experimental details, and more results in training LSTMs (Hochreiter & Schmidhuber, 1997) and WGANs (Arjovsky et al., 2017; Gulrajani et al., 2017) are provided in the Appendix.

**Notation.** We denote scalars and vectors by lower case and lower case bold face letters, respectively, and matrices by upper case bold face letters. For a vector $\boldsymbol{x} = (x_1, \cdots, x_d) \in \mathbb{R}^d$, we denote its $\ell_p$ norm ($p \geq 1$) by $\|\boldsymbol{x}\|_p = (\sum_{i=1}^d |x_i|^p)^{1/p}$. For a matrix $\mathbf{A}$, we use $\|\mathbf{A}\|_p$ to denote its induced norm by the vector $\ell_p$ norm. Given two sequences $\{a_n\}$ and $\{b_n\}$, we write $a_n = O(b_n)$ if there exists a positive constant s.t. $a_n \leq Cb_n$. We denote the interval $a$ to $b$ (included) as $(a, b]$. For a function $f(\boldsymbol{w}) : \mathbb{R}^d \to \mathbb{R}$, we denote its gradient as $\nabla f(\boldsymbol{w})$ and its Hessian as $\nabla^2 f(\boldsymbol{w})$.

## 2 REVIEW: MOMENTUM IN GRADIENT DESCENT

**GD.** GD (2) is a popular approach to solve (1), which dates back to Cauchy (1847). If $f(\boldsymbol{w})$ is convex and $L$-smooth (i.e., $\|\nabla^2 f(\boldsymbol{w})\|_2 \leq L$), then GD converges with rate $O(1/k)$ by letting $s_k \equiv 1/L$ (we use this $s_k$ in all the discussion below), which is independent of the dimension of $\boldsymbol{w}$.

**HB.** HB (4) (Polyak, 1964) accelerates GD by using the historical information, which gives

$$\boldsymbol{w}^{k+1} = \boldsymbol{w}^k - s_k \nabla f(\boldsymbol{w}^k) + \mu(\boldsymbol{w}^k - \boldsymbol{w}^{k-1}), \ \ \mu > 0. \tag{4}$$

We can also accelerate GD by using the Nesterov/lookahead momentum, which leads to (3). Both (3) and (4) have a convergence rate of $O(1/k)$ for convex smooth optimization. Recently, several variants of (3) have been proposed for DL, e.g., (Sutskever et al., 2013) and (Bengio et al., 2013).

**NAG.** NAG (Nesterov, 1983; 2018; Beck & Teboulle, 2009) replaces $\mu$ with $(t_k - 1)/t_{k+1}$, where $t_{k+1} = (1 + \sqrt{1 + 4t_k^2})/2$ with $t_0 = 1$. NAG iterates as following

$$\boldsymbol{v}^{k+1} = \boldsymbol{w}^k - s_k \nabla f(\boldsymbol{w}^k); \ \ \boldsymbol{w}^{k+1} = \boldsymbol{v}^{k+1} + \frac{t_k - 1}{t_{k+1}}(\boldsymbol{v}^{k+1} - \boldsymbol{v}^k). \tag{5}$$

NAG achieves a convergence rate $O(1/k^2)$ with the step size $s_k = 1/L$.

**Remark 1.** *Su et al. (2014) showed that $(k-1)/(k+2)$ is the asymptotic limit of $(t_k - 1)/t_{k+1}$. In the following presentation of NAG with restart, for the ease of notation, we will replace the momentum coefficient $(t_k - 1)/t_{k+1}$ with $(k-1)/(k+2)$.*

**Adaptive Restart NAG (ARNAG).** The sequences, $\{f(\boldsymbol{w}^k) - f(\boldsymbol{w}^*)\}$ where $\boldsymbol{w}^*$ is the minimum of $f(\boldsymbol{w})$, generated by GD and GD with constant momentum (GD + Momentum, which follows (3)) converge monotonically to zero. However, that sequence generated by NAG oscillates, as illustrated in Fig. 2 (a) when $f(\boldsymbol{w})$ is a quadratic function. O'donoghue & Candes (2015) proposed ARNAG

(6), which restart the time-varying momentum of NAG according to the change of function values, to alleviate this oscillatory phenomenon. ARNAG iterates as following

$$\boldsymbol{v}^{k+1} = \boldsymbol{w}^k - s_k\nabla f(\boldsymbol{w}^k);\ \ \boldsymbol{w}^{k+1} = \boldsymbol{v}^{k+1} + \frac{m(k)-1}{m(k)+2}(\boldsymbol{v}^{k+1} - \boldsymbol{v}^k), \tag{6}$$

where $m(1) = 1$; $m(k+1) = m(k) + 1$ if $f(\boldsymbol{w}^{k+1}) \leq f(\boldsymbol{w}^k)$, and $m(k+1) = 1$ otherwise.

**Scheduled Restart NAG (SRNAG).** SR is another strategy to restart the time-varying momentum of NAG. We first divide the total iterations $(0, T]$ (integers only) into a few intervals $\{I_i\}_{i=1}^m = (T_{i-1}, T_i]$, such that $(0, T] = \bigcup_{i=1}^m I_i$. In each $I_i$ we restart the momentum after every $F_i$ iterations. The update rule is then given by:

$$\boldsymbol{v}^{k+1} = \boldsymbol{w}^k - s_k\nabla f(\boldsymbol{w}^k);\ \ \boldsymbol{w}^{k+1} = \boldsymbol{v}^{k+1} + \frac{(k \bmod F_i)}{(k \bmod F_i)+3}(\boldsymbol{v}^{k+1} - \boldsymbol{v}^k). \tag{7}$$

Both AR and SR accelerate NAG to linear convergence for convex problems with the Polyak-Lojasiewicz (PL) condition (Roulet & d'Aspremont, 2017).

**Case Study – Quadratic Function.** Consider the following quadratic optimization (Hardt, 2014)

$$\min_{\boldsymbol{x}} f(\boldsymbol{x}) = \frac{1}{2}\boldsymbol{x}^T\mathbf{L}\boldsymbol{x} - \boldsymbol{x}^T\boldsymbol{b}, \tag{8}$$

where $\mathbf{L} \in \mathbb{R}^{d \times d}$ is the Laplacian of a cycle graph, and $\boldsymbol{b}$ is a $d$-dimensional vector whose first entry is 1 and all the other entries are 0. Note that $f(\boldsymbol{x})$ is convex with Lipschitz constant 4. In particular, we set $d = 1\text{K}$ ($1\text{K}:= 10^3$). We run $T = 50\text{K}$ iterations with step size $1/4$. In SRNAG, we restart, i.e., we set the momentum to 0, after every 1K iterations. Fig. 2 (a) shows that GD + Momentum as in (3) converges faster than GD, while NAG speeds up GD + Momentum dramatically and converges to the minimum in an oscillatory fashion. Both AR and SR accelerate NAG significantly.

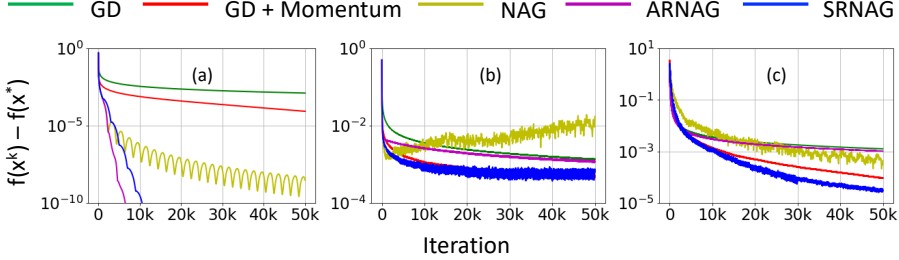

Figure 2: Comparison between different schemes in optimizing the quadratic function in (8) with (a) exact gradient, (b) gradient with constant variance Gaussian noise, and (c) gradient with decaying variance Gaussian noise. NAG, ARNAG, and SRNAG can speed up convergence remarkably when exact gradient is used. Also, SRNAG is more robust to noisy gradient than NAG and ARNAG.

# 3 ALGORITHM PROPOSED: SCHEDULED RESTART SGD (SRSGD)

Computing gradient for ERM, (1), can be computational costly and memory intensive, especially when the training set is large. In many applications, such as training DNNs, SGD is used. In this section, we first prove that the error bound of SGD with NAG cannot be bounded by a convergent sequence, then we formulate our new SRSGD as a solution to accelerate the convergence of SGD using the NAG momentum.

## 3.1 UNCONTROLLED BOUND OF NESTEROV ACCELERATED SGD (NASGD)

Replacing $\nabla f(\boldsymbol{w}^k) := 1/N \sum_{i=1}^N \nabla f_i(\boldsymbol{w}^k)$ in (5) with the mini-batch gradient $1/m \sum_{j=1}^m \nabla f_{i_j}(\boldsymbol{w}^k)$ will lead to uncontrolled error bound. Theorem 1 formulates this observation for NASGD.

**Theorem 1** (Uncontrolled Bound of NASGD). *Let $f(\boldsymbol{w})$ be a convex and $L$-smooth function with $\|\nabla f(\boldsymbol{w})\| \leq R$, where $R > 0$ is a constant. The sequence $\{\boldsymbol{w}^k\}_{k\geq 0}$ generated by (5), with stochastic gradient of bounded variance (Bubeck, 2014; Bottou et al., 2018)[1] and using any constant step size $s_k \equiv s \leq 1/L$, satisfies*

$$\mathbb{E}\left(f(\boldsymbol{w}^k) - f(\boldsymbol{w}^*)\right) = O(k), \tag{9}$$

---

[1]We leave the analysis under the other assumptions (Jain et al., 2018) as a future work.

*where $\boldsymbol{w}^*$ is the minimum of $f$, and the expectation is taken over the generation of the stochastic gradient.*

One idea to prove Theorem 1 is by leveraging the established resulting in Lan (2012). We will provide a new proof of Theorem 1 in Appendix A. The proof shows that the uncontrolled error bound is because the time-varying momentum gets close to 1 as iteration increases. To remedy this, we can restart the momentum in order to guarantee that the time-varying momentum with restart is less than a number that is strictly less than 1. Devolder et al. (2014) proved a similar error bound for the $\delta$-inexact gradient, and we provide a brief review of NAG with $\delta$-inexact gradient in Appendix B. As far as we know that there is no lower bound of $\mathbb{E}(f(\boldsymbol{w}^k) - f(\boldsymbol{w}^*))$ available even for the $\delta$-inexact gradient, and we leave the lower bound estimation as an open problem.

We consider three different inexact gradients: Gaussian noise with constant and decaying variance corrupted gradients for the quadratic optimization (8), and training logistic regression model for MNIST (LeCun & Cortes, 2010) classification. The detailed settings and discussion are provided in Appendix B. We denote SGD with NAG momentum as NASGD and NASGD with AR and SR as ARSGD and SRSGD, respectively. The results shown in Fig. 2 (b) and (c) (iteration vs. optimal gap for quadratic optimization (8)) and Fig. 3 (a) (iteration vs. loss for training logistic regression) confirm Theorem 1. For these cases, SR improves the performance of NAG with inexact gradients. Moreover, when an inexact gradient is used, ARNAG/ARSGD performs almost the same as GD/SGD asymptotically because ARNAG/ARSGD restarts too often and almost degenerates to GD/SGD.

## 3.2 SRSGD AND ITS CONVERGENCE

For ERM (1), SRSGD replaces $\nabla f(\boldsymbol{w})$ in (7) with stochastic gradient with batch size $m$ and gives

$$\boldsymbol{v}^{k+1} = \boldsymbol{w}^k - s_k \frac{1}{m} \sum_{j=1}^m \nabla f_{i_j}(\boldsymbol{w}^k); \ \ \boldsymbol{w}^{k+1} = \boldsymbol{v}^{k+1} + \frac{(k \bmod F_i)}{(k \bmod F_i) + 3}(\boldsymbol{v}^{k+1} - \boldsymbol{v}^k), \tag{10}$$

where $F_i$ is the restart frequency used in the interval $I_i$. We implemented SRSGD, in both PyTorch (Paszke et al., 2019) and Keras (Chollet et al., 2015), by changing just a few lines of code on top of the existing implementation of the SGD optimizer. We provide a snippet of SRSGD code in Appendix J (PyTorch) and K (Keras). We formulate the convergence of SRSGD for general convex and nonconvex problems in Theorem 2 and provide its proof in Appendix C.

**Theorem 2** (Convergence of SRSGD). *Suppose $f(\boldsymbol{w})$ is $L$-smooth. Consider the sequence $\{\boldsymbol{w}^k\}_{k \geq 0}$ generated by (10) with stochastic gradient that is bounded and has bounded variance, and consider any restart frequency $F$ using any constant step size $s_k := s \leq 1/L$. Assume that $\sum_{k \in \mathcal{A}} \left( \mathbb{E}f(\boldsymbol{w}^{k+1}) - \mathbb{E}f(\boldsymbol{w}^k) \right) = \bar{R} < +\infty$ with $\bar{R}$ being a constant and the set $\mathcal{A} := \{k \in \mathbb{Z}^+ | \mathbb{E}f(\boldsymbol{w}^{k+1}) \geq \mathbb{E}f(\boldsymbol{w}^k)\}$, then we have*

$$\min_{1 \leq k \leq K} \left\{ \mathbb{E}\|\nabla f(\boldsymbol{w}^k)\|_2^2 \right\} = O\left( s + \frac{1}{sK} \right). \tag{11}$$

*If $f(\boldsymbol{w})$ is further convex and $\sum_{k \in \mathcal{B}} \left( \mathbb{E}f(\boldsymbol{w}^{k+1}) - \mathbb{E}f(\boldsymbol{w}^k) \right) = \hat{R} < +\infty$ with $\hat{R}$ being a constant and the set $\mathcal{B} := \{k \in \mathbb{Z}^+ | \mathbb{E}\|\boldsymbol{w}^{k+1} - \boldsymbol{w}^*\|^2 \geq \mathbb{E}\|\boldsymbol{w}^k - \boldsymbol{w}^*\|^2\}$, then*

$$\min_{1 \leq k \leq K} \left\{ \mathbb{E} \left( f(\boldsymbol{w}^k) - f(\boldsymbol{w}^*) \right) \right\} = O\left( s + \frac{1}{sK} \right), \tag{12}$$

*where $\boldsymbol{w}^*$ is the minimum of $f$. To obtain $\epsilon$ ($\forall \epsilon > 0$) error, we set $s = O(\epsilon)$ and $K = O(1/\epsilon^2)$.*

Theorem 2 relies on the assumption that $\sum_{k \in \mathcal{A} \text{ or } \mathcal{B}} \left( \mathbb{E}f(\boldsymbol{w}^{k+1}) - \mathbb{E}f(\boldsymbol{w}^k) \right)$ is bounded, and we provide an empirical verification in Appendix C.1. We leave it open for how to establish the convergence result for SRSGD without this assumption.

## 4 EXPERIMENTAL RESULTS

We evaluate SRSGD on a variety of benchmarks for image classification, including CIFAR10, CIFAR100, and ImageNet. In all experiments, we show the advantage of SRSGD over the widely used and well-calibrated SGD baselines with a constant momentum of 0.9 and decreasing learning

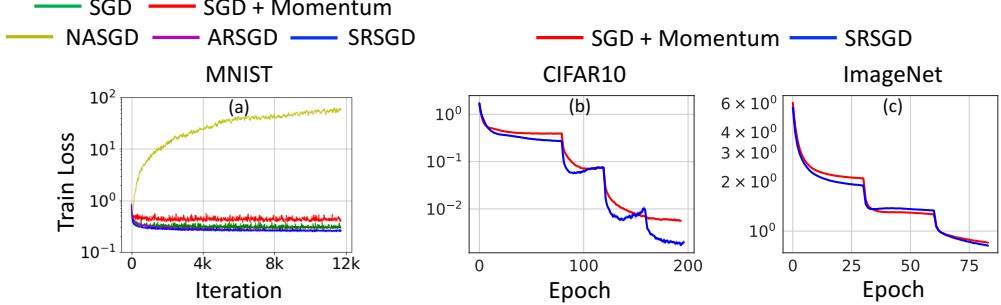

Figure 3: (a) Training loss comparison between different schemes in training logistic regression for MNIST classification. Here, SGD is the plain SGD without momentum, and SGD + Momentum that follows (3) and replaces gradient with the mini-batch stochastic gradient. NASGD is not robust to noisy gradient, ARSGD almost degenerates to SGD, and SRSGD performs the best in this case. (b, c) Training loss vs. training epoch of ResNet models trained with SRSGD (blue) and the SGD baseline with constant momentum as in PyTorch implementation, which is denoted by SGD in Section 4 (red).

rate at certain epochs, which we denote as SGD. We also compare SRSGD with the well-calibrated SGD in which we switch momentum to the Nesterov momentum of 0.9, and we denote this optimizer as SGD + NM. We fine tune the SGD and SGD + NM baselines to obtain the best validation performance, and we then adopt the same set of parameters for training with SRSGD. In the SRSGD experiments, *we tune the restart frequencies on small DNNs for each task based on the validation performance and apply the calibrated restart frequencies to large DNNs for the same task.* Note that ARSGD is impractical for training on large-scale datasets since it requires to compute the loss over the whole training set at each iteration, which is very computationally inefficient. Alternatively, ARSGD can estimate loss and restart using mini-batches, but then ARSGD restarts too often and degenerates to SGD without momentum as we mentioned in Section 3. Thus, we do not compare with ARSGD in our CIFAR and ImageNet experiments. The details about hyper-parameters calibration can be found in Appendix D.4. We provide the detailed description of datasets and experimental settings in Appendix D. Additional experimental results in training LSTMs (Hochreiter & Schmidhuber, 1997) and WGANs (Arjovsky et al., 2017; Gulrajani et al., 2017) with SRSGD, as well as the comparison between SRSGD and SGD + NM on ImageNet classification task, are provided in Appendix E. We also note that in all the following experiments, the training loss will blow up if we apply NASGD without restart. These further confirm the stabilizing effect of scheduled restart in training DNNs.

## 4.1 CIFAR10 AND CIFAR100

We summarize our results for CIFAR in Tables 1 and 2. We also explore two different restarting frequency schedules for SRSGD: *linear* and *exponential* schedule. These schedules are governed by two parameters: the initial restarting frequency $F_1$ and the growth rate $r$. In both scheduling schemes, the restarting frequency at the 1st learning rate stage is set to $F_1$ during training. Then the restarting frequency at the $(k + 1)$-th learning rate stage is determined by:

$$F_{k+1} = \begin{cases} F_1 \times r^k, & \text{exponential schedule} \\ F_1 \times (1 + (r - 1) \times k), & \text{linear schedule.} \end{cases}$$

We search $F_1$ and $r$ using the method outlined in Appendix D.4. For CIFAR10, $(F_1 = 40, r = 1.25)$ and $(F_1 = 30, r = 2)$ are good initial restarting frequencies and growth rates for the exponential and linear schedules, respectively. For CIFAR100, those values are $(F_1 = 45, r = 1.5)$ for the exponential schedule and $(F_1 = 50, r = 2)$ for the linear schedule.

**Improvement in Accuracy Increases with Depth.** We observe that the linear schedule of restart yields better test error on CIFAR than the exponential schedule for most of the models except for Pre-ResNet-470 and Pre-ResNet-1001 on CIFAR100 (see Tables 1 and 2). SRSGD with either linear or exponential restart schedule outperforms SGD. Furthermore, the advantage of SRSGD over SGD is more significant for deeper networks. This observation holds strictly when using the linear schedule (see Fig. 1) and is generally true when using the exponential schedule with only a few exceptions.

**Faster Convergence Reduces the Training Time by Half.** SRSGD also converges faster than SGD. This result is consistent with our MNIST case study in Section 3 and indeed expected since SRSGD

Table 1: Classification test error (%) on CIFAR10 using SGD, SGD + NM, and SRSGD. We report the results of SRSGD with two restarting schedules: linear (lin) and exponential (exp). The numbers of iterations after which we restart the momentum in the lin schedule are 30, 60, 90, 120 for the 1st, 2nd, 3rd, and 4th stage. Those numbers for the exp schedule are 40, 50, 63, 78. We include the reported results from (He et al., 2016b) (in parentheses) in addition to our reproduced results.

| Network | # Params | SGD (baseline) | SGD+NM | SRSGD (lin) | SRSGD (exp) | Improve over SGD (lin/exp) | Improve over SGD+NM (lin/exp) |
|---|---|---|---|---|---|---|---|
| Pre-ResNet-110 | 1.1M | $5.25 \pm 0.14$ (6.37) | $5.24 \pm 0.16$ | $\mathbf{4.93 \pm 0.13}$ | $5.00 \pm 0.47$ | $\mathbf{0.32}/0.25$ | $\mathbf{0.31}/0.24$ |
| Pre-ResNet-290 | 3.0M | $5.05 \pm 0.23$ | $5.04 \pm 0.12$ | $\mathbf{4.37 \pm 0.15}$ | $4.50 \pm 0.18$ | $\mathbf{0.68}/0.55$ | $\mathbf{0.67}/0.54$ |
| Pre-ResNet-470 | 4.9M | $4.92 \pm 0.10$ | $4.97 \pm 0.15$ | $\mathbf{4.18 \pm 0.09}$ | $4.49 \pm 0.19$ | $\mathbf{0.74}/0.43$ | $\mathbf{0.79}/0.48$ |
| Pre-ResNet-650 | 6.7M | $4.87 \pm 0.14$ | $4.80 \pm 0.14$ | $\mathbf{4.00 \pm 0.07}$ | $4.40 \pm 0.13$ | $\mathbf{0.87}/0.47$ | $\mathbf{0.80}/0.40$ |
| Pre-ResNet-1001 | 10.3M | $4.84 \pm 0.19$ (4.92) | $4.62 \pm 0.14$ | $\mathbf{3.87 \pm 0.07}$ | $4.13 \pm 0.10$ | $\mathbf{0.97}/0.71$ | $\mathbf{0.75}/0.49$ |

Table 2: Classification test error (%) on CIFAR100 using SGD, SGD + NM, and SRSGD. We report the results of SRSGD with two restarting schedules: linear (lin) and exponential (exp). The numbers of iterations after which we restart the momentum in the lin schedule are 50, 100, 150, 200 for the 1st, 2nd, 3rd, and 4th stage. Those numbers for the exp schedule are 45, 68, 101, 152. We include the reported results from (He et al., 2016b) (in parentheses) in addition to our reproduced results.

| Network | # Params | SGD (baseline) | SGD+NM | SRSGD (lin) | SRSGD (exp) | Improve over SGD (lin/exp) | Improve over SGD+NM (lin/exp) |
|---|---|---|---|---|---|---|---|
| Pre-ResNet-110 | 1.2M | $23.75 \pm 0.20$ | $23.65 \pm 0.36$ | $\mathbf{23.49 \pm 0.23}$ | $23.50 \pm 0.39$ | $\mathbf{0.26}/0.25$ | $\mathbf{0.16}/0.15$ |
| Pre-ResNet-290 | 3.0M | $21.78 \pm 0.21$ | $21.68 \pm 0.21$ | $\mathbf{21.49 \pm 0.27}$ | $21.58 \pm 0.20$ | $\mathbf{0.29}/0.20$ | $\mathbf{0.19}/0.10$ |
| Pre-ResNet-470 | 4.9M | $21.43 \pm 0.30$ | $21.21 \pm 0.30$ | $20.71 \pm 0.32$ | $\mathbf{20.64 \pm 0.18}$ | $0.72/\mathbf{0.79}$ | $0.50/\mathbf{0.57}$ |
| Pre-ResNet-650 | 6.7M | $21.27 \pm 0.14$ | $21.04 \pm 0.38$ | $\mathbf{20.36 \pm 0.25}$ | $20.41 \pm 0.21$ | $\mathbf{0.91}/0.86$ | $\mathbf{0.68}/0.63$ |
| Pre-ResNet-1001 | 10.4M | $20.87 \pm 0.20$ (22.71) | $20.13 \pm 0.16$ | $19.75 \pm 0.11$ | $\mathbf{19.53 \pm 0.19}$ | $1.12/\mathbf{1.34}$ | $0.38/\mathbf{0.60}$ |

Table 3: On CIFAR10/100 (%), SRSGD training *with only 100 epochs* achieves comparable classification errors (%) to the SGD baseline training with 200 epochs.

| | CIFAR10 | | CIFAR100 | |
|---|---|---|---|---|
| Network | SRSGD | Improvement | SRSGD | Improvement |
| Pre-ResNet-110 | $5.43 \pm 0.18$ | $-0.18$ | $23.85 \pm 0.19$ | $-0.10$ |
| Pre-ResNet-290 | $4.83 \pm 0.11$ | $0.22$ | $21.77 \pm 0.43$ | $0.01$ |
| Pre-ResNet-470 | $4.64 \pm 0.17$ | $0.28$ | $21.42 \pm 0.19$ | $0.01$ |
| Pre-ResNet-650 | $4.43 \pm 0.14$ | $0.44$ | $21.04 \pm 0.20$ | $0.23$ |
| Pre-ResNet-1001 | $4.17 \pm 0.20$ | $0.67$ | $20.27 \pm 0.11$ | $0.60$ |
| Pre-ResNet-110 | $5.25 \pm 0.10$ (110 epochs) | $0.00$ | $23.73 \pm 0.23$ (140 epochs) | $0.02$ |

Table 4: Test errors on CIFAR10 (%) of Pre-ResNet-110 (Left)/290 (Right) using different optimizers.

| SRSGD | Adam | RMSProp | SRSGD | Adam | RMSProp |
|---|---|---|---|---|---|
| $\mathbf{4.93 \pm 0.13}$% | $6.83 \pm 0.10$% | $7.31 \pm 0.31$% | $\mathbf{4.37 \pm 0.15}$% | $6.12 \pm 0.18$% | $7.18 \pm 0.05$% |

can avoid the error accumulation when there is an inexact oracle. For CIFAR, Fig. 3 (b) shows that SRSGD yields smaller training loss than SGD during the training. Interestingly, SRSGD converges quickly to good loss values in the 2nd and 3rd stages. This suggests that *the model can be trained with SRSGD in many fewer epochs compared to SGD* while achieving a similar error rate.

Results in Table 3 confirm the hypothesis above. We train Pre-ResNet models with SRSGD *in only 100 epochs*, decreasing the learning rate by a factor of 10 at the 80th, 90th, and 95th epoch while using the same linear schedule for restarting frequency as before with ($F_1 = 30, r = 2$) for CIFAR10 and ($F_1 = 50, r = 2$) for CIFAR100. We compare the test error of the trained models with those trained by the SGD baseline in 200 epochs. We observe that SRSGD training consistently yields lower test errors than SGD except for the case of Pre-ResNet-110 even though *the number of training epochs of our method is only half of the number of training epochs required by SGD*. For Pre-ResNet-110, SRSGD needs 110 epochs with learning rate decreased at the 80th, 90th, and 100th epoch to achieve the same error rate as the 200-epoch SGD training on CIFAR10. On CIFAR100, SRSGD training for Pre-ResNet-110 needs 140 epochs with learning rate decreased at the 80th, 100th and 120th epoch to outperform the 200-epoch SGD. Comparison with SGD short training is provided in Appendix F.2.

**Comparison with Adam and RMSProp.** SRSGD outperforms not only SGD with momentum but also other popular optimizers including Adam and RMSProp (Tieleman & Hinton, 2012) for image classification tasks. In fact, for image classification tasks, Adam and RMSProp yield worse performance than the baseline SGD with momentum (Chen & Kyrillidis, 2019). Table 4 compares SRSGD with Adam and RMSprop on CIFAR10.

Table 5: Single crop validation errors (%) on ImageNet of ResNets trained with SGD baseline and SRSGD. We report the results of SRSGD with the increasing restarting frequency in the first two learning rates. In the last learning rate, the restarting frequency is linearly decreased to 1. For baseline results, we also include the reported single-crop validation errors (He et al., 2016c) (in parentheses).

| Network | # Params | SGD | | SRSGD | | Improvement | |
|---|---|---|---|---|---|---|---|
| | | top-1 | top-5 | top-1 | top-5 | top-1 | top-5 |
| ResNet-50 | 25.56M | $24.11 \pm 0.10$ (24.70) | $7.22 \pm 0.14$ (7.80) | $\mathbf{23.85 \pm 0.09}$ | $\mathbf{7.10 \pm 0.09}$ | 0.26 | 0.12 |
| ResNet-101 | 44.55M | $22.42 \pm 0.03$ (23.60) | $6.22 \pm 0.01$ (7.10) | $\mathbf{22.06 \pm 0.10}$ | $\mathbf{6.09 \pm 0.07}$ | 0.36 | 0.13 |
| ResNet-152 | 60.19M | $22.03 \pm 0.12$ (23.00) | $6.04 \pm 0.07$ (6.70) | $\mathbf{21.46 \pm 0.07}$ | $\mathbf{5.69 \pm 0.03}$ | 0.57 | 0.35 |
| ResNet-200 | 64.67M | $22.13 \pm 0.12$ | $6.00 \pm 0.07$ | $\mathbf{20.93 \pm 0.13}$ | $\mathbf{5.57 \pm 0.05}$ | 1.20 | 0.43 |

Table 6: Comparison of single crop validation errors on ImageNet (%) between SRSGD training *with fewer epochs* and SGD training with full 90 epochs.

| Network | SRSGD | Reduction | Improvement | Network | SRSGD | Reduction | Improvement |
|---|---|---|---|---|---|---|---|
| ResNet-50 | $24.30 \pm 0.21$ | 10 | $-0.19$ | ResNet-152 | $21.79 \pm 0.07$ | 15 | 0.24 |
| ResNet-101 | $22.32 \pm 0.06$ | 10 | 0.1 | ResNet-200 | $21.92 \pm 0.17$ | 30 | 0.21 |

## 4.2 IMAGENET

Next, we discuss our experimental results on the 1000-way ImageNet classification task (Russakovsky et al., 2015). We conduct our experiments on ResNet-50, 101, 152, and 200 with 5 different seeds. We use the official PyTorch implementation for all of our ResNet models (Paszke et al., 2019). Following common practice, we train each model for 90 epochs and decrease the learning rate by a factor of 10 at the 30th and 60th epoch. We use an initial learning rate of 0.1, a momentum scaled by 0.9, and a weight decay value of 0.0001. Additional details and comparisons between SRSGD and SGD + NM are given in Appendix E.

We report single crop validation errors of ResNet models trained with SGD and SRSGD on ImageNet in Table 5. In contrast to our CIFAR experiments, we observe that for ResNets trained on ImageNet with SRSGD, linearly decreasing the restarting frequency to 1 at the last stage (i.e., after the 60th epoch) helps improve the generalization of the models. Thus, in our experiments, we use linear scheduling with $(F_1 = 40, r = 2)$. From epoch 60 to 90, the restarting frequency decays to 1 linearly.

**Advantage of SRSGD continues to grow with depth.** Similar to the CIFAR experiments, we observe that SRSGD outperforms the SGD baseline for all ResNet models that we study. As shown in Fig. 1, *the advantage of SRSGD over SGD grows with network depth*, just as in our CIFAR experiments with Pre-ResNet architectures.

**Avoiding Overfitting in ResNet-200.** ResNet-200 demonstrates that *SRSGD is better than the SGD baseline at avoiding overfitting*[2]. The ResNet-200 trained with SGD has a top-1 error of 22.13%, higher than the ResNet-152 trained with SGD, which achieves a top-1 error of 22.03% (see Table 5). He et al. (2016b) pointed out that ResNet-200 suffers from overfitting. The ResNet-200 trained with our SRSGD has a top-1 error of 20.93%, which is 1.2% lower than the ResNet-200 trained with the SGD and also lower than the ResNet-152 trained with both SRSGD and SGD, an improvement by 0.53% and 1.1%, respectively. We hypothesize that SRSGD with appropriate restart frequency is locally not monotonic (see Fig. 3 (b, c)), and this property allows SRSGD to escape from bad minima in order to reach a better one, which helps avoid overfitting in very deep networks. Theoretical analysis of the observation that SRSGD is less overfitting in training DNNs is under our investigation.

**Training ImageNet in Fewer Number of Epochs.** As in the CIFAR experiments, we note that when training on ImageNet, SRSGD converges faster than SGD at the first and last learning rate while quickly reaching a good loss value at the second learning rate (see Fig. 3 (c)). This observation suggests that ResNets can be trained with SRSGD in fewer epochs while still achieving comparable error rates to the same models trained by the SGD baseline using all 90 epochs. We summarize the results in Table 6. On ImageNet, we note that SRSGD helps reduce the number of training epochs for very deep networks (ResNet-101, 152, 200). For smaller networks like ResNet-50, training with fewer epochs slightly decreases the accuracy.

## 4.3 EMPIRICAL ANALYSIS

**SRSGD Helps Reduce the Training Time.** We find that SRSGD training using fewer epochs yields comparable error rates to both the SGD baseline and the SRSGD full training with 200 epochs on CIFAR. We conduct an ablation study to understand the impact of reducing the number of epochs

---

[2]By overfitting, we mean that the model achieves low training error but high test error.

on the final error rate when training with SRSGD on CIFAR10 and ImageNet. In the CIFAR10 experiments, we vary the number of epoch reduction from 15 to 90 while in the ImageNet experiments, we vary the number of epoch reduction from 10 to 30. We summarize our results in Fig. 4, and provide detailed results in Appendix F. For CIFAR10, we can train with 30 fewer epochs while still maintaining a comparable error rate to the full SRSGD training, and with a better error rate than the SGD baseline trained in full 200 epochs. For ImageNet, SRSGD training with fewer epochs decreases the accuracy but still obtains comparable results to the 90-epoch SGD baseline.

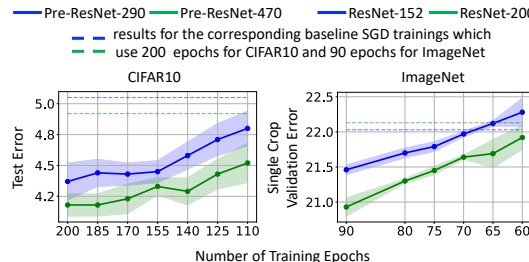

Figure 4: Test error vs. number of training epochs. Dashed lines are test errors of SGD trained with 200 epochs for CIFAR10 (left) and 90 epochs for ImageNet (right). For CIFAR, SRSGD with fewer epochs achieves comparable results to SRSGD with 200 epochs. For ImageNet, training with less epochs slightly decreases the performance of SRSGD but still achieves comparable results to 200-epoch SGD.

**Impact of Restarting Frequency.** We examine the impact of restarting frequency on the network training. We choose a case study of training a Pre-ResNet-290 on CIFAR10 using SRSGD with a linear schedule scheme for the restarting frequency. We fix the growth rate $r = 2$ and vary the initial restarting frequency $F_1$ from 1 to 80. As shown in Fig. 5, SRSGD with a large $F_1$, e.g. $F_1 = 80$, approximates NASGD (yellow). We also show the training loss and test accuracy of NASGD in red. As discussed in Section 3, it suffers from error accumulation due to stochastic gradients and converges slowly or even diverges. SRSGD with small $F_1$, e.g. $F_1 = 1$, approximates SGD without momentum (green). It converges faster initially but reaches a worse local minimum (i.e. larger loss). Typical SRSGD (blue) converges faster than NASGD and to a better local minimum than both NASGD and SGD without momentum. It also achieves the best test error. We provide more empirical analysis results in Appendix F, G and H. The impact of the growth rate $r$ is studied in Appendix G.2.

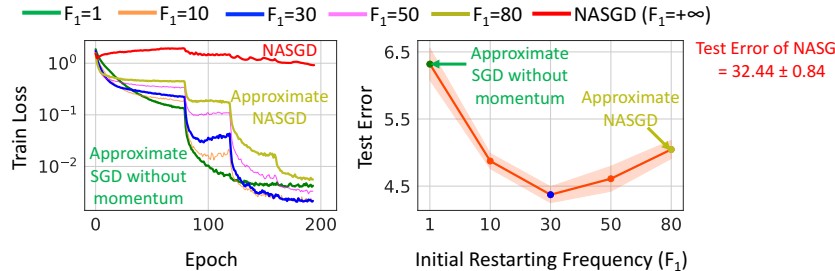

Figure 5: Training loss (left) and test error (right) of Pre-ResNet-290 trained on CIFAR10 with different initial restarting frequencies $F_1$ (linear schedule). SRSGD with small $F_1$ approximates SGD without momentum, while SRSGD with large $F_1$ approximates NASGD. The training loss curve and test accuracy of NASGD are shown in red and confirm the result of Theorem 1 that NASGD accumulates error due to the stochastic gradients.

## 5 CONCLUSIONS

We propose the Scheduled Restart SGD (SRSGD), with two major changes from the widely used SGD with constant momentum. First, we replace the momentum in SGD with the iteration-dependent momentum that used in Nesterov accelerated gradient (NAG). Second, we restart the NAG momentum according to a schedule to prevent error accumulation when the stochastic gradient is used. For image classification, SRSGD can significantly improve the accuracy of the trained DNNs. Also, compared to the SGD baseline, SRSGD requires fewer training epochs to reach the same trained model's accuracy. There are numerous avenues for future work: 1) deriving the optimal restart scheduling and the corresponding convergence rate of SRSGD and 2) integrating the scheduled restart NAG momentum with adaptive learning rate algorithms, e.g., Adam (Kingma & Ba, 2014).

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

# Part

# Appendices

The appendices are structured as follows. In Section A, we prove Theorem 1. In Section B, we review an error accumulation result of the Nesterov accelerated gradient with $\delta$-inexact gradient. In Section C, we prove Theorem 2. In Section D, we provide some experimental details; in particular, the calibration of restarting hyperparameters. In Section E, we compare SRSGD with benchmark optimization algorithms on some other tasks, including training LSTM and Wasserstein GAN. In Section F, we provide detailed experimental settings in studying the effects of reducing the number of epoch in training deep neural networks with SRSGD, and we provide some more experimental results. In Section G and H, we further study the effects of restarting frequency and training with less epochs by using SRSGD. In Section I, we visualize the optimization trajectory of SRSGD and compare it with benchmark methods. A snippet of our implementation of SRSGD in PyTorch and Keras are available in Section J and K, respectively.

## Table of Contents

# A    UNCONTROLLED BOUND OF NASGD

Consider the following optimization problem

$$\min_{\boldsymbol{w}} f(\boldsymbol{w}), \tag{13}$$

where $f(\boldsymbol{w})$ is $L$-smooth and convex.

Start from $\boldsymbol{w}^k$, GD update, with step size $\frac{1}{r}$, can be obtained based on the minimization of the function

$$Q_r(\boldsymbol{v}, \boldsymbol{w}^k) := \langle \boldsymbol{v} - \boldsymbol{w}^k, \nabla f(\boldsymbol{w}^k) \rangle + \frac{r}{2} \|\boldsymbol{v} - \boldsymbol{w}^k\|_2^2. \tag{14}$$

With direct computation, we can get that

$$Q_r(\boldsymbol{v}^{k+1}, \boldsymbol{w}^k) - \min Q_r(\boldsymbol{v}, \boldsymbol{w}^k) = \frac{\|\mathbf{g}^k - \nabla f(\mathbf{w}^k)\|_2}{2r},$$

where $\mathbf{g}^k := \frac{1}{m} \sum_{j=1}^m \nabla f_{i_j}(\boldsymbol{w}^k)$. We assume the variance is bounded, which gives The stochastic gradient rule, $\mathcal{R}_s$, satisfies $\mathbb{E}[Q_r(\boldsymbol{v}^{k+1}, \boldsymbol{w}^k) - \min Q_r(\boldsymbol{v}, \boldsymbol{w}^k)|\chi^k] \leq \delta$, with $\delta$ being a constant and $\chi^k$ being the sigma algebra generated by $\boldsymbol{w}^1, \boldsymbol{w}^2, \cdots, \boldsymbol{w}^k$, i.e.,

$$\chi^k := \sigma(\boldsymbol{w}^1, \boldsymbol{w}^2, \cdots, \boldsymbol{w}^k).$$

NASGD can be reformulated as

$$\begin{aligned} \boldsymbol{v}^{k+1} &\approx \quad \arg\min_{\boldsymbol{v}} Q_r(\boldsymbol{v}, \boldsymbol{w}^k) \text{ with rule } \mathcal{R}_s, \\ \boldsymbol{w}^{k+1} &= \quad \boldsymbol{v}^{k+1} + \frac{t_k - 1}{t_{k+1}}(\boldsymbol{v}^{k+1} - \boldsymbol{v}^k), \end{aligned} \tag{15}$$

where $t_0 = 1$ and $t_{k+1} = (1 + \sqrt{1 + 4t_k^2})/2$.

## A.1    PRELIMINARIES

To proceed, we introduce several definitions and some useful properties in variational and convex analysis. More detailed background can be found at Mordukhovich (2006); Nesterov (1998); Rockafellar & Wets (2009); Rockafellar (1970).

Let $f$ be a convex function, we say that $f$ is *L-smooth (gradient Lipschitz)* if $f$ is differentiable and

$$\|\nabla f(\boldsymbol{v}) - \nabla f(\boldsymbol{w})\|_2 \leq L\|\boldsymbol{v} - \boldsymbol{w}\|_2,$$

and we say $f$ is *$\nu$-strongly convex* if for any $\boldsymbol{w}, \boldsymbol{v} \in \mathrm{dom}(f)$

$$f(\boldsymbol{w}) \geq f(\boldsymbol{v}) + \langle \nabla f(\boldsymbol{v}), \boldsymbol{w} - \boldsymbol{v} \rangle + \frac{\nu}{2}\|\boldsymbol{w} - \boldsymbol{v}\|_2^2.$$

Below of this subsection, we list several basic but useful lemmas, the proof can be found in Nesterov (1998).

**Lemma 1.** *If $f$ is $\nu$-strongly convex, then for any $\boldsymbol{v} \in \mathrm{dom}(f)$ we have*

$$f(\boldsymbol{v}) - f(\boldsymbol{v}^*) \geq \frac{\nu}{2}\|\boldsymbol{v} - \boldsymbol{v}^*\|_2^2, \tag{16}$$

*where $\boldsymbol{v}^*$ is the minimizer of $f$.*

**Lemma 2.** *If $f$ is L-smooth, for any $\boldsymbol{w}, \boldsymbol{v} \in \mathrm{dom}(f)$,*

$$f(\boldsymbol{w}) \leq f(\boldsymbol{v}) + \langle \nabla f(\boldsymbol{v}), \boldsymbol{w} - \boldsymbol{v} \rangle + \frac{L}{2}\|\boldsymbol{w} - \boldsymbol{v}\|_2^2.$$

## A.2   UNCONTROLLED BOUND OF NASGD: ANALYSIS

In this part, we denote

$$\tilde{\boldsymbol{v}}^{k+1} := \arg\min_{\boldsymbol{v}} Q_r(\boldsymbol{v}, \boldsymbol{w}^k). \tag{17}$$

**Lemma 3.** *If the constant $r > 0$, then*

$$\mathbb{E}\left(\|\boldsymbol{v}^{k+1} - \tilde{\boldsymbol{v}}^{k+1}\|_2^2 | \chi^k\right) \leq \frac{2\delta}{r}. \tag{18}$$

*Proof.* Note that $Q_r(\boldsymbol{v}, \boldsymbol{w}^k)$ is strongly convex with constant $r$, and $\tilde{\boldsymbol{v}}^{k+1}$ in (17) is the minimizer of $Q_r(\boldsymbol{v}, \boldsymbol{w}^k)$. With Lemma 1 we have

$$Q_r(\boldsymbol{v}^{k+1}, \boldsymbol{w}^k) - Q_r(\tilde{\boldsymbol{v}}^{k+1}, \boldsymbol{w}^k) \geq \frac{r}{2}\|\boldsymbol{v}^{k+1} - \tilde{\boldsymbol{v}}^{k+1}\|_2^2. \tag{19}$$

Notice that

$$\mathbb{E}\left[Q_r(\boldsymbol{v}^{k+1}, \boldsymbol{w}^k) - Q_r(\tilde{\boldsymbol{v}}^{k+1}, \boldsymbol{w}^k)\right] = \mathbb{E}\left[Q_r(\boldsymbol{v}^{k+1}, \boldsymbol{w}^k) - \min_{\boldsymbol{v}} Q_r(\boldsymbol{v}, \boldsymbol{w}^k)\right] \leq \delta.$$

The inequality (18) can be established by combining the above two inequalities. ☐

**Lemma 4.** *If the constant satisfy $r > L$, then we have*

$$\mathbb{E}\left(f(\tilde{\boldsymbol{v}}^{k+1}) + \frac{r}{2}\|\tilde{\boldsymbol{v}}^{k+1} - \boldsymbol{w}^k\|_2^2 - (f(\boldsymbol{v}^{k+1}) + \frac{r}{2}\|\boldsymbol{v}^{k+1} - \boldsymbol{w}^k\|_2^2)\right) \tag{20}$$

$$\geq -\tau\delta - \frac{r-L}{2}\mathbb{E}[\|\boldsymbol{w}^k - \tilde{\boldsymbol{v}}^{k+1}\|_2^2],$$

*where $\tau = \frac{L^2}{r(r-L)} + 1$.*

*Proof.* The convexity of $f$ gives us

$$0 \leq \langle \nabla f(\boldsymbol{v}^{k+1}), \boldsymbol{v}^{k+1} - \tilde{\boldsymbol{v}}^{k+1}\rangle + f(\tilde{\boldsymbol{v}}^{k+1}) - f(\boldsymbol{v}^{k+1}). \tag{21}$$

From the definition of the stochastic gradient rule $\mathcal{R}_s$, we have

$$-\delta \leq \mathbb{E}\left(Q_r(\tilde{\boldsymbol{v}}^{k+1}, \boldsymbol{w}^k) - Q_r(\boldsymbol{v}^{k+1}, \boldsymbol{w}^k)\right) \tag{22}$$

$$= \mathbb{E}\left[\langle \tilde{\boldsymbol{v}}^{k+1} - \boldsymbol{w}^k, \nabla f(\boldsymbol{w}^k)\rangle + \frac{r}{2}\|\tilde{\boldsymbol{v}}^{k+1} - \boldsymbol{w}^k\|_2^2\right] -$$

$$\mathbb{E}\left[\langle \boldsymbol{v}^{k+1} - \boldsymbol{w}^k, \nabla f(\boldsymbol{w}^k)\rangle + \frac{r}{2}\|\boldsymbol{v}^{k+1} - \boldsymbol{w}^k\|_2^2\right].$$

With (21) and (22), we have

$$-\delta \leq \left(f(\tilde{\boldsymbol{v}}^{k+1}) + \frac{r}{2}\|\tilde{\boldsymbol{v}}^{k+1} - \boldsymbol{w}^k\|_2^2\right) - \left(f(\boldsymbol{v}^{k+1}) + \frac{r}{2}\|\boldsymbol{v}^{k+1} - \boldsymbol{w}^k\|_2^2\right) + \tag{23}$$

$$\mathbb{E}\langle \nabla f(\boldsymbol{w}^k) - \nabla f(\tilde{\boldsymbol{v}}^{k+1}), \tilde{\boldsymbol{v}}^{k+1} - \boldsymbol{v}^{k+1}\rangle.$$

With the Schwarz inequality $\langle \boldsymbol{a}, \boldsymbol{b}\rangle \leq \frac{\|\boldsymbol{a}\|_2^2}{2\mu} + \frac{\mu}{2}\|\boldsymbol{b}\|_2^2$ with $\mu = \frac{L^2}{r-L}$, $\boldsymbol{a} = \nabla f(\boldsymbol{v}^{k+1}) - \nabla f(\tilde{\boldsymbol{v}}^{k+1})$ and $\boldsymbol{b} = \boldsymbol{w}^k - \tilde{\boldsymbol{v}}^{k+1}$,

$$\langle \nabla f(\boldsymbol{w}^k) - \nabla f(\tilde{\boldsymbol{v}}^{k+1}), \tilde{\boldsymbol{v}}^{k+1} - \boldsymbol{v}^{k+1}\rangle \tag{24}$$

$$\leq \frac{(r-L)}{2L^2}\|\nabla f(\boldsymbol{w}^k) - \nabla f(\tilde{\boldsymbol{v}}^{k+1})\|_2^2 + \frac{L^2}{2(r-L)}\|\boldsymbol{v}^{k+1} - \tilde{\boldsymbol{v}}^{k+1}\|_2^2$$

$$\leq \frac{(r-L)}{2}\|\boldsymbol{w}^k - \tilde{\boldsymbol{v}}^{k+1}\|_2^2 + \frac{L^2}{2(r-L)}\|\boldsymbol{v}^{k+1} - \tilde{\boldsymbol{v}}^{k+1}\|_2^2.$$

Combining (23) and (24), we have

$$-\delta \leq \mathbb{E}\left(f(\tilde{\boldsymbol{v}}^{k+1}) + \frac{r}{2}\|\tilde{\boldsymbol{v}}^{k+1} - \boldsymbol{w}^k\|_2^2\right) - \mathbb{E}\left(f(\boldsymbol{v}^{k+1}) + \frac{r}{2}\|\boldsymbol{v}^{k+1} - \boldsymbol{w}^k\|_2^2\right) \tag{25}$$

$$+ \frac{L^2}{2(r-L)}\mathbb{E}\|\boldsymbol{v}^{k+1} - \tilde{\boldsymbol{v}}^{k+1}\|_2^2 + \frac{r-L}{2}\mathbb{E}\|\boldsymbol{w}^k - \tilde{\boldsymbol{v}}^{k+1}\|_2^2.$$

By rearrangement of the above inequality (25) and using Lemma 3, we obtain the result. ☐

**Lemma 5.** *If the constants satisfy $r > L$, then we have the following bounds*

$$\mathbb{E}\left(f(\boldsymbol{v}^k) - f(\boldsymbol{v}^{k+1})\right) \geq \frac{r}{2}\mathbb{E}\|\boldsymbol{w}^k - \boldsymbol{v}^{k+1}\|_2^2 + r\mathbb{E}\langle \boldsymbol{w}^k - \boldsymbol{v}^k, \tilde{\boldsymbol{v}}^{k+1} - \boldsymbol{w}^k\rangle - \tau\delta, \qquad (26)$$

$$\mathbb{E}\left(f(\boldsymbol{v}^*) - f(\boldsymbol{v}^{k+1})\right) \geq \frac{r}{2}\mathbb{E}\|\boldsymbol{w}^k - \boldsymbol{v}^{k+1}\|_2^2 + r\mathbb{E}\langle \boldsymbol{w}^k - \boldsymbol{v}^*, \tilde{\boldsymbol{v}}^{k+1} - \boldsymbol{w}^k\rangle - \tau\delta, \qquad (27)$$

*where $\tau := \frac{L^2}{r(r-L)} + 1$ and $\boldsymbol{v}^*$ is the minimum .*

*Proof.* With Lemma 2, we have

$$-f(\tilde{\boldsymbol{v}}^{k+1}) \geq -f(\boldsymbol{w}^k) - \langle \tilde{\boldsymbol{v}}^{k+1} - \boldsymbol{w}^k, \nabla f(\boldsymbol{w}^k)\rangle - \frac{L}{2}\|\tilde{\boldsymbol{v}}^{k+1} - \boldsymbol{w}^k\|_2^2. \qquad (28)$$

Using the convexity of $f$, we have

$$f(\boldsymbol{v}^k) - f(\boldsymbol{w}^k) \geq \langle \boldsymbol{v}^k - \boldsymbol{w}^k, \nabla f(\boldsymbol{w}^k)\rangle,$$

i.e.,

$$f(\boldsymbol{v}^k) \geq f(\boldsymbol{w}^k) + \langle \boldsymbol{v}^k - \boldsymbol{w}^k, \nabla f(\boldsymbol{w}^k)\rangle. \qquad (29)$$

According to the definition of $\tilde{\boldsymbol{v}}^{k+1}$ in (14), i.e.,

$$\tilde{\boldsymbol{v}}^{k+1} = \arg\min_{\boldsymbol{v}} Q_r(\boldsymbol{v}, \boldsymbol{w}^k) = \arg\min_{\boldsymbol{v}} \langle \boldsymbol{v} - \boldsymbol{w}^k, \nabla f(\boldsymbol{w}^k)\rangle + \frac{r}{2}\|\boldsymbol{v} - \boldsymbol{w}^k\|_2^2,$$

and the optimization condition gives

$$\tilde{\boldsymbol{v}}^{k+1} = \boldsymbol{w}^k - \frac{1}{r}\nabla f(\boldsymbol{w}^k). \qquad (30)$$

Substituting (30) into (29), we obtain

$$f(\boldsymbol{v}^k) \geq f(\boldsymbol{w}^k) + \langle \boldsymbol{v}^k - \boldsymbol{w}^k, r(\boldsymbol{w}^k - \tilde{\boldsymbol{v}}^{k+1})\rangle. \qquad (31)$$

Direct summation of (28) and (31) gives

$$f(\boldsymbol{v}^k) - f(\tilde{\boldsymbol{v}}^{k+1}) \geq \left(r - \frac{L}{2}\right)\|\tilde{\boldsymbol{v}}^{k+1} - \boldsymbol{w}^k\|_2^2 + r\langle \boldsymbol{w}^k - \boldsymbol{v}^k, \tilde{\boldsymbol{v}}^{k+1} - \boldsymbol{w}^k\rangle. \qquad (32)$$

Summing (32) and (20), we obtain the inequality (26)

$$\mathbb{E}\left[f(\boldsymbol{v}^k) - f(\boldsymbol{v}^{k+1})\right] \geq \frac{r}{2}\mathbb{E}\|\boldsymbol{w}^k - \boldsymbol{v}^{k+1}\|_2^2 + r\mathbb{E}\langle \boldsymbol{w}^k - \boldsymbol{v}^k, \tilde{\boldsymbol{v}}^{k+1} - \boldsymbol{w}^k\rangle - \tau\delta. \qquad (33)$$

On the other hand, with the convexity of $f$, we have

$$f(\boldsymbol{v}^*) - f(\boldsymbol{w}^k) \geq \langle \boldsymbol{v}^* - \boldsymbol{w}^k, \nabla f(\boldsymbol{w}^k)\rangle = \langle \boldsymbol{v}^* - \boldsymbol{w}^k, r(\boldsymbol{w}^k - \tilde{\boldsymbol{v}}^{k+1})\rangle. \qquad (34)$$

The summation of (28) and (34) results in

$$f(\boldsymbol{v}^*) - f(\tilde{\boldsymbol{v}}^{k+1}) \geq \left(r - \frac{L}{2}\right)\|\boldsymbol{w}^k - \tilde{\boldsymbol{v}}^{k+1}\|_2^2 + r\langle \boldsymbol{w}^k - \boldsymbol{v}^*, \tilde{\boldsymbol{v}}^{k+1} - \boldsymbol{w}^k\rangle. \qquad (35)$$

Summing (35) and (20), we obtain

$$\mathbb{E}\left(f(\boldsymbol{v}^*) - f(\boldsymbol{v}^{k+1})\right) \geq \frac{r}{2}\mathbb{E}\|\boldsymbol{w}^k - \boldsymbol{v}^{k+1}\|_2^2 + r\mathbb{E}\langle \boldsymbol{w}^k - \boldsymbol{v}^*, \tilde{\boldsymbol{v}}^{k+1} - \boldsymbol{w}^k\rangle - \tau\delta, \qquad (36)$$

which is the same as (27). $\qquad\square$

**Theorem 3** (Uncontrolled Bound of NASGD (Theorem 1 with detailed bounded)). *Let the constant $r$ satisfy $r < L$ and the sequence $\{\boldsymbol{v}^k\}_{k\geq 0}$ be generated by NASGD with stochastic gradient that has bounded variance. By using any constant step size $s_k \equiv s \leq 1/L$, then we have*

$$\mathbb{E}[f(\boldsymbol{v}^k) - \min_{\boldsymbol{v}} f(\boldsymbol{v})] \leq \left(\frac{2\tau\delta}{r} + R^2\right)\frac{4k}{3}. \qquad (37)$$

*Proof.* We denote
$$F^k := \mathbb{E}(f(\boldsymbol{v}^k) - f(\boldsymbol{v}^*)).$$

By (26) $\times (t_k - 1) + $ (27), we have

$$
\frac{2[(t_k - 1)F^k - t_k F^{k+1}]}{r} \geq t_k \mathbb{E}\|\boldsymbol{v}^{k+1} - \boldsymbol{w}^k\|_2^2 \tag{38}
$$
$$
+ 2\mathbb{E}\langle \tilde{\boldsymbol{v}}^{k+1} - \boldsymbol{w}^k, t_k \boldsymbol{w}^k - (t_k - 1)\boldsymbol{v}^k - \boldsymbol{v}^*\rangle - \frac{2\tau t_k \delta}{r}.
$$

With $t_{k-1}^2 = t_k^2 - t_k$, (38) $\times t_k$ yields

$$
\frac{2[t_{k-1}^2 F^k - t_k^2 F^{k+1}]}{r} \geq \mathbb{E}\|t_k \boldsymbol{v}^{k+1} - t_k \boldsymbol{w}^k\|_2^2 \tag{39}
$$
$$
+ 2t_k \mathbb{E}\langle \tilde{\boldsymbol{v}}^{k+1} - \boldsymbol{w}^k, t_k \boldsymbol{w}^k - (t_k - 1)\boldsymbol{v}^k - \boldsymbol{v}^*\rangle - \frac{2\tau t_k^2 \delta}{r}
$$

Substituting $\boldsymbol{a} = t_k \boldsymbol{v}^{k+1} - (t_k - 1)\boldsymbol{v}^k - \boldsymbol{v}^*$ and $\boldsymbol{b} = t_k \boldsymbol{w}^k - (t_k - 1)\boldsymbol{v}^k - \boldsymbol{v}^*$ into identity

$$
\|\boldsymbol{a} - \boldsymbol{b}\|_2^2 + 2\langle \boldsymbol{a} - \boldsymbol{b}, \boldsymbol{b}\rangle = \|\boldsymbol{a}\|_2^2 - \|\boldsymbol{b}\|_2^2. \tag{40}
$$

It follows that

$$
\mathbb{E}\|t_k \boldsymbol{v}^{k+1} - t_k \boldsymbol{w}^k\|_2^2 + 2t_k \mathbb{E}\langle \tilde{\boldsymbol{v}}^{k+1} - \boldsymbol{w}^k, t_k \boldsymbol{w}^k - (t_k - 1)\boldsymbol{v}^k - \boldsymbol{v}^*\rangle \tag{41}
$$
$$
= \mathbb{E}\|t_k \boldsymbol{v}^{k+1} - t_k \boldsymbol{w}^k\|_2^2 + 2t_k \mathbb{E}\langle \boldsymbol{v}^{k+1} - \boldsymbol{w}^k, t_k \boldsymbol{w}^k - (t_k - 1)\boldsymbol{v}^k - \boldsymbol{v}^*\rangle
$$
$$
+ 2t_k \mathbb{E}\langle \tilde{\boldsymbol{v}}^{k+1} - \boldsymbol{v}^{k+1}, t_k \boldsymbol{w}^k - (t_k - 1)\boldsymbol{v}^k - \boldsymbol{v}^*\rangle
$$
$$
\underset{(40)}{=} \mathbb{E}\|t_k \boldsymbol{v}^{k+1} - (t_k - 1)\boldsymbol{v}^k - \boldsymbol{v}^*\|_2^2 - \|t_k \boldsymbol{w}^k - (t_k - 1)\boldsymbol{v}^k - \boldsymbol{v}^*\|_2^2
$$
$$
+ 2t_k \mathbb{E}\langle \tilde{\boldsymbol{v}}^{k+1} - \boldsymbol{v}^{k+1}, t_k \boldsymbol{w}^k - (t_k - 1)\boldsymbol{v}^k - \boldsymbol{v}^*\rangle
$$
$$
= \mathbb{E}\|t_k \boldsymbol{v}^{k+1} - (t_k - 1)\boldsymbol{v}^k - \boldsymbol{v}^*\|_2^2 - \mathbb{E}\|t_{k-1}\boldsymbol{v}^k - (t_{k-1} - 1)\boldsymbol{v}^{k-1} - \boldsymbol{v}^*\|_2^2
$$
$$
+ 2t_k \mathbb{E}\langle \tilde{\boldsymbol{v}}^{k+1} - \boldsymbol{v}^{k+1}, t_{k-1}\boldsymbol{v}^k - (t_{k-1} - 1)\boldsymbol{v}^{k-1} - \boldsymbol{v}^*\rangle.
$$

In the third identity, we used the fact $t_k \boldsymbol{w}^k = t_k \boldsymbol{v}^k + (t_{k-1} - 1)(\boldsymbol{v}^k - \boldsymbol{v}^{k-1})$. If we denote $u^k = \mathbb{E}\|t_{k-1}\boldsymbol{v}^k - (t_{k-1} - 1)\boldsymbol{v}^{k-1} - \boldsymbol{v}^*\|_2^2$, (39) can be rewritten as

$$
\frac{2t_k^2 F^{k+1}}{r} + u^{k+1} \leq \frac{2t_{k-1}^2 F^k}{r} + u^k + \frac{2\tau t_k^2 \delta}{r} \tag{42}
$$
$$
+ 2t_k \mathbb{E}\langle \boldsymbol{v}^{k+1} - \tilde{\boldsymbol{v}}^{k+1}, t_{k-1}\boldsymbol{v}^k - (t_{k-1} - 1)\boldsymbol{v}^{k-1} - \boldsymbol{v}^*\rangle
$$
$$
\leq \frac{2t_k^2 F^k}{r} + u^k + \frac{2\tau t_k^2 \delta}{r} + t_{k-1}^2 R^2,
$$

where we used

$$
2t_k \mathbb{E}\langle \boldsymbol{v}^{k+1} - \tilde{\boldsymbol{v}}^{k+1}, t_{k-1}\boldsymbol{v}^k - (t_{k-1} - 1)\boldsymbol{v}^{k-1} - \boldsymbol{v}^*\rangle
$$
$$
\leq t_k^2 \mathbb{E}\|\boldsymbol{v}^{k+1} - \tilde{\boldsymbol{v}}^{k+1}\|_2^2 + \mathbb{E}\|t_{k-1}\boldsymbol{v}^k - (t_{k-1}\boldsymbol{v}^k - (t_{k-1} - 1)\boldsymbol{v}^{k-1} - \boldsymbol{v}^*)\|_2^2
$$
$$
= 2t_k^2 \delta / r + t_{k-1}^2 R^2.
$$

Denoting

$$
\xi_k := \frac{2t_{k-1}^2 F^k}{r} + u^k,
$$

then, we have

$$
\xi_{k+1} \leq \xi_0 + \left(\frac{2\tau\delta}{r} + R^2\right)\sum_{i=1}^k t_i^2 = \left(\frac{2\tau\delta}{r} + R^2\right)\frac{k^3}{3}. \tag{43}
$$

With the fact, $\xi_k \geq \frac{2t_{k-1}^2 F^k}{r} \geq k^2 F^k / 4$, we then proved the result. $\qquad\square$

# B  NAG WITH $\delta$-INEXACT ORACLE & EXPERIMENTAL SETTINGS IN SECTION 3.1

In Devolder et al. (2014), the authors defines $\delta$-inexact gradient oracle for convex smooth optimization as follows:

**Definition 1** ($\delta$-Inexact Oracle). *Devolder et al. (2014) For a convex L-smooth function $f : \mathbb{R}^d \to \mathbb{R}$. For $\forall w \in \mathbb{R}^d$ and exact first-order oracle returns a pair $(f(w), \nabla f(w)) \in \mathbb{R} \times \mathbb{R}^d$ so that for $\forall v \in \mathbb{R}^d$ we have*

$$0 \leq f(v) - \big(f(w) + \langle \nabla f(w), v - w \rangle\big) \leq \frac{L}{2}\|w - v\|_2^2.$$

*A $\delta$-inexact oracle returns a pair $(f^\delta(w), \nabla f^\delta(w)) \in \mathbb{R} \times \mathbb{R}^d$ so that $\forall v \in \mathbb{R}^d$ we have*

$$0 \leq f(v) - \big(f^\delta(w) + \langle \nabla f^\delta(w), v - w \rangle\big) \leq \frac{L}{2}\|w - v\|_2^2 + \delta.$$

We have the following convergence results of GD and NAG under a $\delta$-Inexact Oracle for convex smooth optimization.

**Theorem 4.** *Devolder et al. (2014)[3] Consider*

$$\min f(w), \ \ w \in \mathbb{R}^d,$$

*where $f(w)$ is convex and L-smooth with $w^*$ being the minimum. Given access to $\delta$-inexact oracle, GD with step size $1/L$ returns a point $w^k$ after $k$ steps so that*

$$f(w^k) - f(w^*) = O\left(\frac{L}{k}\right) + \delta.$$

*On the other hand, NAG, with step size $1/L$ returns*

$$f(w^k) - f(w^*) = O\left(\frac{L}{k^2}\right) + O(k\delta).$$

Theorem 4 says that NAG may not robust to a $\delta$-inexact gradient. In the following, we will study the numerical behavior of a variety of first-order algorithms for convex smooth optimizations with the following different inexact gradients.

**Constant Variance Gaussian Noise:** We consider the inexact oracle where the true gradient is contaminated with a Gaussian noise $\mathcal{N}(0, 0.001^2)$. We run 50K iterations of different algorithms. For SRNAG, we restart after every 200 iterations. Fig. 2 (b) shows the iteration vs. optimal gap, $f(x^k) - f(x^*)$, with $x^*$ being the minimum. NAG with the inexact gradient due to constant variance noise does not converge. GD performs almost the same as ARNAG asymptotically, because ARNAG restarts too often and almost degenerates into GD. GD with constant momentum outperforms the three schemes above, and SRNAG slightly outperforms GD with constant momentum.

**Decaying Variance Gaussian Noise:** Again, consider minimizing (8) with the same experimental setting as before except that $\nabla f(x)$ is now contaminated with a decaying Gaussian noise $\mathcal{N}(0, (\frac{0.1}{\lfloor t/100 \rfloor + 1})^2)$. For SRNAG, we restart every 200 iterations in the first $10k$ iterations, and restart every 400 iterations in the remaining 40K iterations. Fig. 2 (c) shows the iteration vs. optimal gap by different schemes. ARNAG still performs almost the same as GD. The path of NAG is oscillatory. GD with constant momentum again outperforms the previous three schemes. Here SRNAG significantly outperforms all the other schemes.

**Logisitic Regression for MNIST Classification:** We apply the above schemes with stochastic gradient to train a logistic regression model for MNIST classification LeCun & Cortes (2010). We consider five different schemes, namely, SGD, SGD + (constant) momentum, NASGD, ASGD, and SRSGD. In ARSGD, we perform restart based on the loss value of the mini-batch training data. In SRSGD, we restart the NAG momentum after every 10 iterations. We train the logistic regression model with a $\ell_2$ weight decay of $10^{-4}$ by running 20 epochs using different schemes with batch size of 128. The step sizes for all the schemes are set to 0.01. Fig. 3 (a) plots the training loss vs. iteration. In this case, NASGD does not converge, and SGD with momentum does not speed up SGD. ARSGD's performance is on par with SGD's. Again, SRSGD gives the best performance with the smallest training loss among these five schemes.

---

[3]We adopt the result from Hardt (2014).

## C Convergence of SRSGD

We prove the convergence of Nesterov accelerated SGD with scheduled restart, i.e., the convergence of SRSGD. We denote that $\theta^k := \frac{t_k-1}{t_{k+1}}$ in the Nesterov iteration and $\hat{\theta}^k$ is its use in the restart version, i.e., SRSGD. For any restart frequency $F$ (positive integer), we have $\hat{\theta}^k = \theta^{k-\lfloor k/F \rfloor * F}$. In the restart version, we can see that

$$\hat{\theta}^k \leq \theta^F =: \bar{\theta} < 1.$$

**Lemma 6.** *Let the constant satisfies $r > L$ and the sequence $\{v^k\}_{k\geq0}$ be generated by the SRSGD with restart frequency $F$ (any positive integer), we have*

$$\sum_{i=1}^{k} \|v^i - v^{i-1}\|_2^2 \leq \frac{r^2 k R^2}{(1-\bar{\theta})^2}, \tag{44}$$

*where $\bar{\theta} := \theta^F < 1$ and $R := \sup_{\mathbf{x}}\{\|\nabla f(\mathbf{x})\|_2\}$.*

*Proof.* It holds that

$$
\begin{aligned}
\|v^{k+1} - w^k\|_2 &= \|v^{k+1} - v^k + v^k - w^k\|_2 \\
&\geq \|v^{k+1} - v^k\|_2 - \|v^k - w^k\|_2 \\
&\geq \|v^{k+1} - v^k\|_2 - \bar{\theta}\|v^k - v^{k-1}\|_2.
\end{aligned}
\tag{45}
$$

Thus,

$$
\begin{aligned}
\|v^{k+1} - w^k\|_2^2 &\geq \left(\|v^{k+1} - v^k\|_2 - \bar{\theta}\|v^k - v^{k-1}\|_2\right)^2 \\
&= \|v^{k+1} - v^k\|_2^2 - 2\bar{\theta}\|v^k - v^{k-1}\|_2\|v^k - v^{k-1}\|_2 + \bar{\theta}^2\|v^k - v^{k-1}\|_2^2 \\
&\geq (1-\bar{\theta})\|v^{k+1} - v^k\|_2^2 - \bar{\theta}(1-\bar{\theta})\|v^{k+1} - v^k\|_2^2.
\end{aligned}
\tag{46}
$$

Summing (46) from $k = 1$ to $K$, we get

$$(1-\bar{\theta})^2 \sum_{k=1}^{K} \|v^k - v^{k-1}\|_2^2 \leq \sum_{k=1}^{K} \|v^{k+1} - w^k\|_2^2 \leq r^2 K R^2. \tag{47}$$

$\square$

In the following, we denote

$$\mathcal{A} := \{k \in Z^+ | \mathbb{E}f(v^k) \geq \mathbb{E}f(v^{k-1})\}.$$

**Theorem 5** (Convergence of SRSGD). *(Theorem 2 with detailed bound) Suppose $f(w)$ is $L$-smooth. Consider the sequence $\{w^k\}_{k\geq0}$ generated by (10) with stochastic gradient that is bounded and has bound variance. Using any restart frequency $F$ and any constant step size $s_k := s \leq 1/L$. Assume that $\sum_{k\in\mathcal{A}} \left(\mathbb{E}f(w^{k+1}) - \mathbb{E}f(w^k)\right) = \bar{R} < +\infty$, then we have*

$$\min_{1\leq k\leq K} \left\{\mathbb{E}\|\nabla f(w^k)\|_2^2\right\} \leq \frac{rR^2}{(1-\bar{\theta})^2} \frac{L(1+\tilde{\theta})}{2} + \frac{rLR^2}{2} + \frac{\tilde{\theta}\tilde{R}}{rK}. \tag{48}$$

*If $f(w)$ is further convex and the set $\mathcal{B} := \{k \in \mathbb{Z}^+ | \mathbb{E}\|w^{k+1} - w^*\|^2 \geq \mathbb{E}\|w^k - w^*\|^2\}$ obeys $\sum_{k\in\mathcal{B}} \left(\mathbb{E}f(w^{k+1}) - \mathbb{E}f(w^k)\right) = \hat{R} < +\infty$, then*

$$\min_{1\leq k\leq K} \left\{\mathbb{E}\left(f(w^k) - f(w^*)\right)\right\} \leq \frac{\|w^0 - w^*\|^2 + \hat{R}}{2\gamma k} + \frac{\gamma R^2}{2}, \tag{49}$$

*where $w^*$ is the minimum of $f$. To obtain $\epsilon$ ($\forall \epsilon > 0$) error, we set $s = O(\epsilon)$ and $K = O(1/\epsilon^2)$.*

*Proof.* Firstly, we show the convergence of SRSGD for nonconvex optimization. $L$-smoothness of $f$, i.e., Lipschitz gradient continuity, gives us

$$f(v^{k+1}) \leq f(w^k) + \langle \nabla f(w^k), v^{k+1} - w^k \rangle + \frac{L}{2}\|v^{k+1} - w^k\|_2^2. \tag{50}$$

Taking expectation, we get

$$\mathbb{E}f(\boldsymbol{v}^{k+1}) \le \mathbb{E}f(\boldsymbol{w}^k) - r\mathbb{E}\|\nabla f(\boldsymbol{w}^k)\|_2^2 + \frac{r^2 LR^2}{2}. \tag{51}$$

On the other hand, we have

$$f(\boldsymbol{w}^k) \le f(\boldsymbol{v}^k) + \hat{\theta}^k\langle\nabla f(\boldsymbol{v}^k), \boldsymbol{v}^k - \boldsymbol{v}^{k-1}\rangle + \frac{L(\hat{\theta}^k)^2}{2}\|\boldsymbol{v}^k - \boldsymbol{v}^{k-1}\|_2^2. \tag{52}$$

Then, we have

$$
\begin{aligned}
\mathbb{E}f(\boldsymbol{v}^{k+1}) &\le \mathbb{E}f(\boldsymbol{v}^k) + \hat{\theta}^k\mathbb{E}\langle\nabla f(\boldsymbol{v}^k), \boldsymbol{v}^k - \boldsymbol{v}^{k-1}\rangle \\
&+ \frac{L(\hat{\theta}^k)^2}{2}\mathbb{E}\|\boldsymbol{v}^k - \boldsymbol{v}^{k-1}\|_2^2 - r\mathbb{E}\|\nabla f(\boldsymbol{w}^k)\|_2^2 + \frac{r^2 LR^2}{2}.
\end{aligned} \tag{53}
$$

We also have

$$\hat{\theta}^k\langle\nabla f(\boldsymbol{v}^k), \boldsymbol{v}^k - \boldsymbol{v}^{k-1}\rangle \le \hat{\theta}^k\left(f(\boldsymbol{v}^k) - f(\boldsymbol{v}^{k-1}) + \frac{L}{2}\|\boldsymbol{v}^k - \boldsymbol{v}^{k-1}\|_2^2\right). \tag{54}$$

We then get that

$$\mathbb{E}f(\boldsymbol{v}^{k+1}) \le \mathbb{E}f(\boldsymbol{v}^k) + \hat{\theta}^k\left(\mathbb{E}f(\boldsymbol{v}^k) - \mathbb{E}f(\boldsymbol{v}^{k-1})\right) - r\mathbb{E}\|\nabla f(\boldsymbol{w}^k)\|_2^2 + A_k, \tag{55}$$

where

$$A_k := \mathbb{E}\frac{L}{2}\|\boldsymbol{v}^k - \boldsymbol{v}^{k-1}\|_2^2 + \frac{L(\hat{\theta}^k)^2}{2}\mathbb{E}\|\boldsymbol{v}^k - \boldsymbol{v}^{k-1}\|_2^2 + \frac{r^2 LR^2}{2}.$$

Summing the inequality gives us

$$
\begin{aligned}
\mathbb{E}f(\boldsymbol{v}^{K+1}) \le \mathbb{E}f(\boldsymbol{v}^0) &+ \tilde{\theta}\sum_{k\in\mathcal{A}}\left(\mathbb{E}f(\boldsymbol{v}^k) - \mathbb{E}f(\boldsymbol{v}^{k-1})\right) \\
&- r\sum_{k=1}^{K}\mathbb{E}\|\nabla f(\boldsymbol{w}^k)\|_2^2 + \sum_{k=1}^{K}A_k.
\end{aligned} \tag{56}
$$

It is easy to see that

$$\tilde{\theta}\sum_{k\in\mathcal{A}}\left(\mathbb{E}f(\boldsymbol{v}^k) - \mathbb{E}f(\boldsymbol{v}^{k-1})\right) = \tilde{\theta}\tilde{R}.$$

We get the result by using Lemma 6

Secondly, we prove the convergence of SRSGD for convex optimization. Let $\boldsymbol{w}^*$ be the minimizer of $f$. We have

$$
\begin{aligned}
\mathbb{E}\|\boldsymbol{v}^{k+1} - \boldsymbol{w}^*\|_2^2 &= \mathbb{E}\|\boldsymbol{w}^k - \gamma\nabla f(\boldsymbol{w}^k) - \boldsymbol{w}^*\|_2^2 \\
&= \mathbb{E}\|\boldsymbol{w}^k - \boldsymbol{w}^*\|_2^2 - 2\gamma\mathbb{E}\langle\nabla f(\boldsymbol{w}^k), \boldsymbol{w}^k - \boldsymbol{w}^*\rangle + \gamma^2\mathbb{E}\|\nabla f(\boldsymbol{w}^k)\|_2^2 \\
&\le \mathbb{E}\|\boldsymbol{w}^k - \boldsymbol{x}^*\|_2^2 - 2\gamma\mathbb{E}\langle\nabla f(\boldsymbol{w}^k), \boldsymbol{w}^k - \boldsymbol{w}^*\rangle + \gamma^2 R^2.
\end{aligned} \tag{57}
$$

We can also derive

$$
\begin{aligned}
\mathbb{E}\|\boldsymbol{w}^k - \boldsymbol{w}^*\|_2 &= \mathbb{E}\|\boldsymbol{v}^k + \hat{\theta}^k(\boldsymbol{v}^k - \boldsymbol{v}^{k-1}) - \boldsymbol{w}^*\|_2^2 \\
&= \mathbb{E}\|\boldsymbol{v}^k - \boldsymbol{w}^*\|_2^2 + 2\hat{\theta}^k\mathbb{E}\langle\boldsymbol{v}^k - \boldsymbol{v}^{k-1}, \boldsymbol{v}^k - \boldsymbol{w}^*\rangle + (\hat{\theta}^k)^2\mathbb{E}\|\boldsymbol{v}^k - \boldsymbol{v}^{k-1}\|_2^2 \\
&= \mathbb{E}\|\boldsymbol{v}^k - \boldsymbol{w}^*\|_2^2 + \hat{\theta}^k\mathbb{E}\left(\|\boldsymbol{v}^k - \boldsymbol{w}^*\|_2^2 + \|\boldsymbol{v}^{k-1} - \boldsymbol{v}^k\|_2^2 - \|\boldsymbol{v}^{k-1} - \boldsymbol{w}^*\|_2^2\right) \\
&+ (\hat{\theta})^2\mathbb{E}\|\boldsymbol{v}^k - \boldsymbol{v}^{k-1}\|_2^2 \\
&= \mathbb{E}\|\boldsymbol{v}^k - \boldsymbol{w}^*\|_2^2 + \hat{\theta}^k\mathbb{E}\left(\|\boldsymbol{v}^k - \boldsymbol{w}^*\|_2^2 - \|\boldsymbol{v}^{k-1} - \boldsymbol{w}^*\|_2^2\right) + 2(\hat{\theta}^k)^2\mathbb{E}\|\boldsymbol{v}^k - \boldsymbol{v}^{k-1}\|_2^2,
\end{aligned}
$$

where we used the following identity

$$(\boldsymbol{a} - \boldsymbol{b})^T(\boldsymbol{a} - \boldsymbol{b}) = \frac{1}{2}[\|\boldsymbol{a} - \boldsymbol{d}\|_2^2 - \|\boldsymbol{a} - \boldsymbol{c}\|_2^2 + \|\boldsymbol{b} - \boldsymbol{c}\|_2^2 - \|\boldsymbol{b} - \boldsymbol{d}\|_2^2].$$

Then, we have

$$\mathbb{E}\|\boldsymbol{v}^{k+1} - \boldsymbol{w}^*\|_2^2 \leq \mathbb{E}\|\boldsymbol{v}^k - \boldsymbol{w}^*\|_2^2 - 2\gamma\mathbb{E}\langle\nabla f(\boldsymbol{w}^k), \boldsymbol{w}^k - \boldsymbol{w}^*\rangle + 2(\hat{\theta}^k)^2\mathbb{E}\|\boldsymbol{v}^k - \boldsymbol{v}^{k-1}\|_2^2 \quad (58)$$
$$+ r^2R^2 + \hat{\theta}^k\mathbb{E}(\|\boldsymbol{v}^k - \boldsymbol{w}^*\|_2^2 - \|\boldsymbol{v}^{k-1} - \boldsymbol{w}^*\|_2^2).$$

We then get that

$$2\gamma\mathbb{E}\left(f(\boldsymbol{w}^k) - f(\boldsymbol{w}^*)\right) \leq \mathbb{E}\|\boldsymbol{v}^k - \boldsymbol{w}^*\|_2^2 - \mathbb{E}\|\boldsymbol{v}^{k+1} - \boldsymbol{w}^*\|_2^2 \quad (59)$$
$$+ \hat{\theta}^k\left(\mathbb{E}\|\boldsymbol{v}^k - \boldsymbol{w}^*\|_2^2 - \mathbb{E}\|\boldsymbol{v}^{k-1} - \boldsymbol{w}^*\|_2^2\right) + r^2R^2.$$

Summing the inequality gives us the desired convergence result for convex optimization. □

## C.1 NUMERICAL VERIFICATION OF THE ASSUMPTIONS IN THEOREM 2

In this part, we numerically verify the assumptions in Theorem 2. In particular, we apply SRSGD with learning rate 0.1 to train LeNet [4] for MNIST classification (we test on MNIST due to extremely large computational cost). We conduct numerical verification as follows: starting from a given point $\boldsymbol{w}^0$, we randomly sample 469 mini-batches (note in total we have 469 batches in the training data) with batch size 128 and compute the stochastic gradient using each mini-batch. Next, we advance to the next step with each of these 469 stochastic gradients and get the approximated $\mathbb{E}f(\boldsymbol{w}^1)$. We randomly choose one of these 469 positions as the updated weights of our model. By iterating the above procedure, we can get $\boldsymbol{w}^1, \boldsymbol{w}^2, \cdots$ and $\mathbb{E}f(\boldsymbol{w}^1), \mathbb{E}f(\boldsymbol{w}^2), \cdots$ and we use these values to verify our assumptions in Theorem 2. We set restart frequencies to be 20, 40, and 80, respectively. Figure 6 top panels plot $k$ vs. the cardinality of the set $\mathcal{A} := \{k \in \mathbb{Z}^+ | \mathbb{E}f(\boldsymbol{w}^{k+1}) \geq \mathbb{E}f(\boldsymbol{w}^k)\}$, and Figure 6 bottom panels plot $k$ vs. $\sum_{k \in \mathcal{A}}\left(\mathbb{E}f(\boldsymbol{w}^{k+1}) - \mathbb{E}f(\boldsymbol{w}^k)\right)$. Figure 6 shows that $\sum_{k \in \mathcal{A}}\left(\mathbb{E}f(\boldsymbol{w}^{k+1}) - \mathbb{E}f(\boldsymbol{w}^k)\right)$ converges to a constant $\bar{R} < +\infty$. We also noticed that when the training gets plateaued, $\mathbb{E}(f(\boldsymbol{w}^k))$ still oscillates, but the magnitude of the oscillation diminishes as iterations goes, which is consistent with our plots that the cardinality of $\mathcal{A}$ increases linearly, but $\bar{R}$ converges to a finite number. These numerical results show that our assumption in Theorem 2 is reasonable.

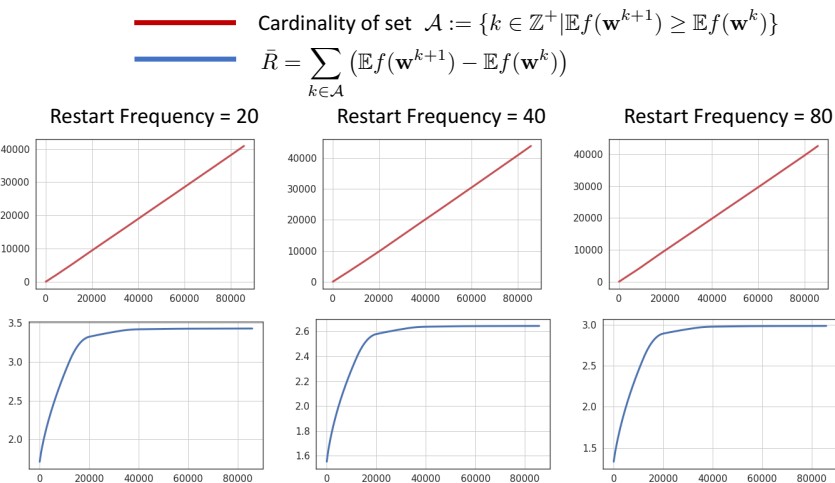

Figure 6: Cardinality of the set $\mathcal{A} := \{k \in \mathbb{Z}^+ | \mathbb{E}f(\boldsymbol{w}^{k+1}) \geq \mathbb{E}f(\boldsymbol{w}^k)\}$ (Top panels) and the value of $\bar{R} = \sum_{k \in \mathcal{A}}\left(\mathbb{E}f(\boldsymbol{w}^{k+1}) - \mathbb{E}f(\boldsymbol{w}^k)\right)$ (Bottom panels). We notice that when the training gets plateaued, $\mathbb{E}(f(\boldsymbol{w}^k))$ still oscillates, but the magnitude of the oscillation diminishes as iterations goes, which is consistent with our plots that the cardinality of $\mathcal{A}$ increases linearly, but $\bar{R}$ converges to a finite number under different restart frequencies. These results confirm that our assumption in Theorem 2 is reasonable.

---

[4]We used the PyTorch implementation of LeNet at https://github.com/activatedgeek/LeNet-5.

# D DATASETS AND IMPLEMENTATION DETAILS

## D.1 CIFAR

The CIFAR10 and CIFAR100 datasets Krizhevsky et al. (2009) consist of 50K training images and 10K test images from 10 and 100 classes, respectively. Both training and test data are color images of size $32 \times 32$. We run our CIFAR experiments on Pre-ResNet-110, 290, 470, 650, and 1001 with 5 different seeds He et al. (2016b). We train each model for 200 epochs with batch size of 128 and initial learning rate of 0.1, which is decayed by a factor of 10 at the 80th, 120th, and 160th epoch. The weight decay rate is $5 \times 10^{-5}$ and the momentum for the SGD baseline is 0.9. Random cropping and random horizontal flipping are applied to training data. Our code is modified based on the Pytorch classification project Yang (2017),[5] which was also used by Liu et al. Liu et al. (2020). We provide the restarting frequencies for the exponential and linear scheme for CIFAR10 and CIFAR100 in Table 7 below. Using the same notation as in the main text, we denote $F_i$ as the restarting frequency at the $i$-th learning rate.

Table 7: Restarting frequencies for CIFAR10 and CIFAR100 experiments

|  | CIFAR10 | CIFAR100 |
|---|---|---|
| Linear schedule | $F_1 = 30, F_2 = 60, F_3 = 90, F_4 = 120 \ (r = 2)$ | $F_1 = 50, F_2 = 100, F_3 = 150, F_4 = 200 \ (r = 2)$ |
| Exponential schedule | $F_1 = 40, F_2 = 50, F_3 = 63, F_4 = 78 \ (r = 1.25)$ | $F_1 = 45, F_2 = 68, F_3 = 101, F_4 = 152 \ (r = 1.50)$ |

## D.2 IMAGENET

The ImageNet dataset contains roughly 1.28 million training color images and 50K validation color images from 1000 classes Russakovsky et al. (2015). We run our ImageNet experiments on ResNet-50, 101, 152, and 200 with 5 different seeds. Following He et al. (2016a;b), we train each model for 90 epochs with a batch size of 256 and decrease the learning rate by a factor of 10 at the 30th and 60th epoch. The initial learning rate is 0.1, the momentum is 0.9, and the weight decay rate is $1 \times 10^{-5}$. Random $224 \times 224$ cropping and random horizontal flipping are applied to training data. We use the official Pytorch ResNet implementation Paszke et al. (2019),[6] and run our experiments on 8 Nvidia V100 GPUs. We report single-crop top-1 and top-5 errors of our models. In our experiments, we set $F_1 = 40$ at the 1st learning rate, $F_2 = 80$ at the 2nd learning rate, and $F_3$ is linearly decayed from 80 to 1 at the 3rd learning rate (see Table 8).

Table 8: Restarting frequencies for ImageNet experiments

|  | ImageNet |
|---|---|
| Linear schedule | $F_1 = 40, F_2 = 80, F_3$: linearly decayed from 80 to 1 in the last 30 epochs |

## D.3 TRAINING IMAGENET IN FEWER NUMBER OF EPOCHS:

Table 9 contains the learning rate and restarting frequency schedule for our experiments on training ImageNet in fewer number of epochs, i.e. the reported results in Table 6 in the main text. Other settings are the same as in the full-training ImageNet experiments described in Section D.2 above.

**Additional Implementation Details:** Implementation details for the ablation study of error rate vs. reduction in epochs and the ablation study of impact of restarting frequency are provided in Section F and G below.

## D.4 DETAILS ON RESTARTING HYPER-PARAMETERS SEARCH

In our CIFAR10 and CIFAR100 experiments, for both linear and exponential schedule, we conduct hyperparameter searches over the restarting frequencies using our smallest model, Pre-ResNet-110,

---

[5]Implementation available at https://github.com/bearpaw/pytorch-classification
[6]Implementation available at https://github.com/pytorch/examples/tree/master/imagenet

Table 9: Learning rate and restarting frequency schedule for ImageNet short training, i.e. Table 6 in the main text.

| | ImageNet |
|---|---|
| ResNet-50 | Decrease the learning rate by a factor of 10 at the 30th and 56th epoch. Train for a total of 80 epochs. |
| | $F_1 = 60, F_2 = 105, F_3$: linearly decayed from 105 to 1 in the last 24 epochs |
| ResNet-101 | Decrease the learning rate by a factor of 10 at the 30th and 56th epoch. Train for a total of 80 epochs. |
| | $F_1 = 40, F_2 = 80, F_3$: linearly decayed from 80 to 1 in the last 24 epochs |
| ResNet-152 | Decrease the learning rate by a factor of 10 at the 30th and 51th epoch. Train for a total of 75 epochs. |
| | $F_1 = 40, F_2 = 80, F_3$: linearly decayed from 80 to 1 in the last 24 epochs |
| ResNet-200 | Decrease the learning rate by a factor of 10 at the 30th and 46th epoch. Train for a total of 60 epochs. |
| | $F_1 = 40, F_2 = 80, F_3$: linearly decayed from 80 to 1 in the last 14 epochs |

making choices based on final validation performance. The same chosen restarting frequencies are applied for all models including Pre-ResNet-110, 290, 470, 650, and 1001. In particular, we use 10,000 images from the original training set as a validation set. This validation set contains 1,000 and 100 images from each class for CIFAR10 and CIFAR100, respectively. We first train Pre-ResNet-110 on the remaining 40,000 training images and use the performance on the validation set averaged over 5 random seeds to select the initial restarting frequency $F_1$ and the growth rate $r$. Both $F_1$ and $r$ are selected using grid search from the sets of $\{20, 25, 30, 35, 40, 45, 50\}$ and $\{1, 1.25, 1.5, 1.75, 2\}$, respectively. We then train all models including Pre-ResNet-110, 290, 470, 650, and 1001 on all 50,000 training images using the selected values of $F_1$ and $r$ and report the results on the test set which contains 10,000 test images. The reported test performance is averaged over 5 random seeds. We also use the same selected values of $F_1$ and $r$ for our short training experiments in Section 4.3.

For ImageNet experiments, we use linear scheduling and sweep over the initial restarting frequency $F_1$ and the growth rate $r$ in the set of $\{20, 30, 40, 50, 60\}$ and $\{1, 1.25, 1.5, 1.75, 2\}$, respectively. We select the values of $F_1 = 40$ and $r = 2$ which have the highest final validation accuracy averaged over 5 random seeds. Same as in CIFAR10 and CIFAR100 experiments, we select $F_1$ and $r$ using our smallest model, ResNet-50, and apply the same selected hyperparameter values for all models including ResNet-50, 101, 152, and 200. We also use the same selected values of $F_1$ and $r$ for our short training experiments in Section 4.3. However, for ResNet-50, we observe that $F_1 = 60$ and $r = 1.75$ yields better performance in short training. All reported results are averaged over 5 random seeds.

# E SRSGD VS. SGD AND SGD + NM ON IMAGENET CLASSIFICATION AND OTHER TASKS

## E.1 COMPARING WITH SGD WITH NESTEROV MOMENTUM ON IMAGENET CLASSIFICATION

In this section, we compare SRSGD with SGD with Nesterov constant momentum (SGD + NM) in training ResNets for ImageNet classification. All hyper-parameters of SGD with constant Nesterov momentum used in our experiments are the same as those of SGD described in section D.2. We list the results in Table 10. Again, SRSGD remarkably outperforms SGD + NM in training ResNets for ImageNet classification, and as the network goes deeper the improvement becomes more significant.

## E.2 LONG SHORT-TERM MEMORY (LSTM) TRAINING FOR PIXEL-BY-PIXEL MNIST

In this task, we examine the advantage of SRSGD over SGD and SGD with Nesterov Momentum in training recurrent neural networks. In our experiments, we use an LSTM with different numbers of hidden units (128, 256, and 512) to classify samples from the well-known MNIST dataset LeCun & Cortes (2010). We follow the implementation of Le et al. (2015) and feed each pixel of the image into the RNN sequentially. In addition, we choose a random permutation of $28 \times 28 = 784$ elements at the beginning of the experiment. This fixed permutation is applied to training and testing sequences. This task is known as permuted MNIST classification, which has become standard to measure the performance of RNNs and their ability to capture long term dependencies.

Table 10: Single crop validation errors (%) on ImageNet of ResNets trained with SGD + NM and SRSGD. We report the results of SRSGD with the increasing restarting frequency in the first two learning rates. In the last learning rate, the restarting frequency is linearly decreased from 70 to 1. For baseline results, we also include the reported single-crop validation errors He et al. (2016c) (in parentheses).

| Network | # Params | SGD + NM | | SRSGD | | Improvement | |
|---|---|---|---|---|---|---|---|
| | | top-1 | top-5 | top-1 | top-5 | top-1 | top-5 |
| ResNet-50 | 25.56M | $24.27 \pm 0.07$ | $7.17 \pm 0.07$ | $\mathbf{23.85 \pm 0.09}$ | $\mathbf{7.10 \pm 0.09}$ | 0.42 | 0.07 |
| ResNet-101 | 44.55M | $22.32 \pm 0.05$ | $6.18 \pm 0.05$ | $\mathbf{22.06 \pm 0.10}$ | $\mathbf{6.09 \pm 0.07}$ | 0.26 | 0.09 |
| ResNet-152 | 60.19M | $21.77 \pm 0.14$ | $5.86 \pm 0.09$ | $\mathbf{21.46 \pm 0.07}$ | $\mathbf{5.69 \pm 0.03}$ | 0.31 | 0.17 |
| ResNet-200 | 64.67M | $21.98 \pm 0.22$ | $5.99 \pm 0.20$ | $\mathbf{20.93 \pm 0.13}$ | $\mathbf{5.57 \pm 0.05}$ | 1.05 | 0.42 |

**Implementation and Training Details:** For the LSTM model, we initialize the forget bias to 1 and other biases to 0. All weights matrices are initialized orthogonally except for the hidden-to-hidden weight matrices, which are initialized to be identity matrices. We train each model for 350 epochs with the initial learning rate of 0.01. The learning rate was reduced by a factor of 10 at epoch 200 and 300. The momentum is set to 0.9 for SGD with standard and Nesterov constant momentum. The restart schedule for SRSGD is set to 90, 30, 90 . The restart schedule changes at epoch 200 and 300. In all experiments, we use batch size 128 and the gradients are clipped so that their L2 norm are at most 1. Our code is based on the code from the exponential RNN's Github.[7]

**Results:** Our experiments corroborate the superiority of SRSGD over the two baselines. SRSGD yields much smaller test error and converges faster than SGD with standard and Nesterov constant momentum across all settings with different number of LSTM hidden units. We summarize our results in Table 11 and Figure 7.

Table 11: Test errors (%) on Permuted MNIST of trained with SGD, SGD + NM and SRSGD. The LSTM model has 128 hidden units. In all experiments, we use the initial learning rate of 0.01, which is reduced by a factor of 10 at epoch 200 and 300. All models are trained for 350 epochs. The momentum for SGD and SGD + NM is set to 0.9. The restart schedule in SRSGD is set to 90, 30, and 90.

| Network | No. Hidden Units | SGD | SGD + NM | SRSGD | Improvement over SGD/SGD + NM |
|---|---|---|---|---|---|
| LSTM | 128 | $10.10 \pm 0.57$ | $9.75 \pm 0.69$ | $\mathbf{8.61 \pm 0.30}$ | 1.49/1.14 |
| LSTM | 256 | $10.42 \pm 0.63$ | $10.09 \pm 0.61$ | $\mathbf{9.03 \pm 0.23}$ | 1.39/1.06 |
| LSTM | 512 | $10.04 \pm 0.35$ | $9.55 \pm 1.09$ | $\mathbf{8.49 \pm 1.59}$ | 1.55/1.06 |

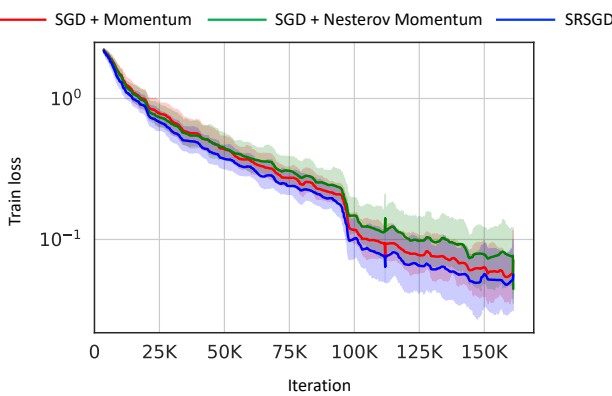

Figure 7: Training loss vs. training iterations of LSTM trained with SGD (red), SGD + NM (green), and SRSGD (blue) for PMNIST classification tasks.

### E.3 WASSERSTEIN GENERATIVE ADVERSARIAL NETWORKS (WGAN) TRAINING ON MNIST

We investigate the advantage of SRSGD over SGD with standard and Nesterov momentum in training deep generative models. In our experiments, we train a WGAN with gradient penalty Gulrajani

---

[7]Implementation available at https://github.com/Lezcano/expRNN

et al. (2017) on MNIST. We evaluate our models using the discriminator's loss, i.e. the Earth Moving distance estimate, since in WGAN lower discriminator loss and better sample quality are correlated Arjovsky et al. (2017).

**Implementation and Training Details:** The detailed implementations of our generator and discriminator are given below. For the generator, we set latent_dim to 100 and d to 32. For the discriminator, we set d to 32. We train each model for 350 epochs with the initial learning rate of 0.01. The learning rate was reduced by a factor of 10 at epoch 200 and 300. The momentum is set to 0.9 for SGD with standard and Nesterov constant momentum. The restart schedule for SRSGD is set to 60, 120, 180. The restart schedule changes at epoch 200 and 300. In all experiments, we use batch size 64. Our code is based on the code from the Pytorch WGAN-GP Github.[8]

```python
import torch
import torch.nn as nn

class Generator(nn.Module):
    def __init__(self, latent_dim, d=32):
        super().__init__()
        self.net = nn.Sequential(
            nn.ConvTranspose2d(latent_dim, d * 8, 4, 1, 0),
            nn.BatchNorm2d(d * 8),
            nn.ReLU(True),

            nn.ConvTranspose2d(d * 8, d * 4, 4, 2, 1),
            nn.BatchNorm2d(d * 4),
            nn.ReLU(True),

            nn.ConvTranspose2d(d * 4, d * 2, 4, 2, 1),
            nn.BatchNorm2d(d * 2),
            nn.ReLU(True),

            nn.ConvTranspose2d(d * 2, 1, 4, 2, 1),
            nn.Tanh()
        )
    def forward(self, x):
        return self.net(x)

class Discriminator(nn.Module):
    def __init__(self, d=32):
        super().__init__()
        self.net = nn.Sequential(
            nn.Conv2d(1, d, 4, 2, 1),
            nn.InstanceNorm2d(d),
            nn.LeakyReLU(0.2),

            nn.Conv2d(d, d * 2, 4, 2, 1),
            nn.InstanceNorm2d(d * 2),
            nn.LeakyReLU(0.2),

            nn.Conv2d(d * 2, d * 4, 4, 2, 1),
            nn.InstanceNorm2d(d * 4),
            nn.LeakyReLU(0.2),

            nn.Conv2d(d * 4, 1, 4, 1, 0),
        )
    def forward(self, x):
        outputs = self.net(x)
        return outputs.squeeze()
```

**Results:** Our SRSGD is still better than both the baselines. SRSGD achieves smaller discriminator loss, i.e. Earth Moving distance estimate, and converges faster than SGD with standard and Nesterov constant momentum. We summarize our results in Table 12 and Figure 8. We also demonstrate the

---

[8]Implementation available at https://github.com/arturml/pytorch-wgan-gp

digits generated by the trained WGAN in Figure 9. By visually evaluation, we observe that samples generated by the WGAN trained with SRSGD look slightly better than those generated by the WGAN trained with SGD with standard and Nesterov constant momentum.

Table 12: Discriminator loss (i.e. Earth Moving distance estimate) of the WGAN with gradient penalty trained on MNIST with SGD, SGD + NM and SRSGD. In all experiments, we use the initial learning rate of 0.01, which is reduced by a factor of 10 at epoch 200 and 300. All models are trained for 350 epochs. The momentum for SGD and SGD + NM is set to 0.9. The restart schedule in SRSGD is set to 60, 120, and 180.

| Task | SGD | SGD + NM | SRSGD | Improvement over SGD/SGD + NM |
|---|---|---|---|---|
| MNIST | $0.71 \pm 0.10$ | $0.58 \pm 0.03$ | **$0.44 \pm 0.06$** | 0.27/0.14 |

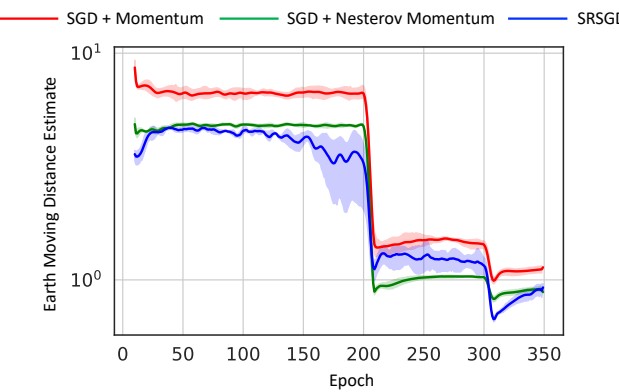

Figure 8: Earth Moving distance estimate (i.e. discriminator loss) vs. training epochs of WGAN with gradient penalty trained with SGD (red), SGD + NM (green), and SRSGD (blue) on MNIST.

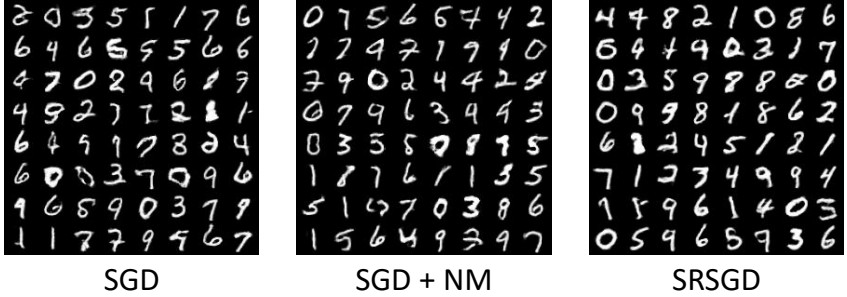

Figure 9: MNIST digits generated by WGAN trained with gradient penalty by SGD (left), SGD + NM (middle), and SRSGD (right).

## F ERROR RATE VS. REDUCTION IN TRAINING EPOCHS

### F.1 IMPLEMENTATION DETAILS

**CIFAR10 (Figure 4, left, in the main text) and CIFAR100 (Figure 11 in this Appendix):** Except for learning rate schedule, we use the same setting described in Section D.1 above and Section 4.1 in the main text. Table 13 contains the learning rate schedule for each number of epoch reduction in Figure 4 (left) in the main text and Figure 11 below.

**ImageNet (Figure 4, right, in the main text):** Except for the total number of training epochs, other settings are similar to experiments for training ImageNet in fewer number of epochs described in Section D.3. In particular, the learning rate and restarting frequency schedule still follow those in Table 9 above. We examine different numbers of training epochs: 90 (0 epoch reduction), 80 (10 epochs reduction), 75 (15 epochs reduction), 70 (20 epochs reduction), 65 (25 epochs reduction), and 60 (30 epochs reduction).

Table 13: Learning rate (LR) schedule for the ablation study of error rate vs. reduction in training epochs for CIFAR10 experiments, i.e. Figure 4 in the main text and for CIFAR100 experiments, i.e. Figure 11 in this Appendix.

| #of Epoch Reduction | LR Schedule |
|---|---|
| 0 | Decrease the LR by a factor of 10 at the 80th, 120th and 160th epoch. Train for a total of 200 epochs. |
| 15 | Decrease the LR by a factor of 10 at the 80th, 115th and 150th epoch. Train for a total of 185 epochs. |
| 30 | Decrease the LR by a factor of 10 at the 80th, 110th and 140th epoch. Train for a total of 170 epochs. |
| 45 | Decrease the LR by a factor of 10 at the 80th, 105th and 130th epoch. Train for a total of 155 epochs. |
| 60 | Decrease the LR by a factor of 10 at the 80th, 100th and 120th epoch. Train for a total of 140 epochs. |
| 75 | Decrease the LR by a factor of 10 at the 80th, 95th and 110th epoch. Train for a total of 125 epochs. |
| 90 | Decrease the LR by a factor of 10 at the 80th, 90th and 100th epoch. Train for a total of 110 epochs. |
| 100 | Decrease the LR by a factor of 10 at the 80th, 90th and 95th epoch. Train for a total of 100 epochs. |

Table 14: Classification test error (%) of SGD short training (100 epochs), SGD full training (200 epochs), SRSGD short training (100 epochs), and SRSGD full training (200 epochs) on CIFAR10. SGD short training yields much worse test errors than the others while SRSGD short training yields either comparable or even better results than SGD full training.

| Network | SGD short training | SGD full training | SRSGD short training | SRSGD full training |
|---|---|---|---|---|
| Pre-ResNet-110 | $6.36 \pm 0.12$ | $5.25 \pm 0.14$ | $5.43 \pm 0.18$ | $4.93 \pm 0.13$ |
| Pre-ResNet-290 | $5.81 \pm 0.10$ | $5.05 \pm 0.23$ | $4.83 \pm 0.11$ | $4.37 \pm 0.15$ |
| Pre-ResNet-470 | $5.59 \pm 0.19$ | $4.92 \pm 0.10$ | $4.64 \pm 0.17$ | $4.18 \pm 0.09$ |

Table 15: Classification test error (%) of SGD short training (100 epochs), SGD full training (200 epochs), SRSGD short training (100 epochs), and SRSGD full training (200 epochs) on CIFAR100. SGD short training yields worse test errors than the others while SRSGD short training yields either comparable or even better results than SGD full training.

| Network | SGD short training | SGD full training | SRSGD short training | SRSGD full training |
|---|---|---|---|---|
| Pre-ResNet-110 | $24.34 \pm 0.21$ | $23.75 \pm 0.20$ | $23.85 \pm 0.19$ | $23.49 \pm 0.23$ |
| Pre-ResNet-290 | $22.70 \pm 0.25$ | $21.78 \pm 0.21$ | $21.77 \pm 0.43$ | $21.49 \pm 0.27$ |
| Pre-ResNet-470 | $22.43 \pm 0.18$ | $21.43 \pm 0.30$ | $21.42 \pm 0.19$ | $20.71 \pm 0.32$ |

## F.2 SHORT TRAINING ON CIFAR10/CIFAR100 USING SGD

For better comparison between SRSGD training using fewer epochs and SGD full training, we also conduct experiments with SGD training using fewer epochs on CIFAR10 and CIFAR100. Table 14 and 15 compares SRSGD short training using 100 epoch, SGD short training using 100 epochs, SRSGD full training using 200 epochs, and SGD full training using 200 epochs for Pre-ResNet-110, 290, and 470 on CIFAR10 and CIFAR100, respectively. The learning rate schedule for SGD short training using 100 epochs is the same as the learning rate schedule for SRSGD short training using 100 epoch given in Section 4 and in Table 13 above. In particular, for both SGD and SRSGD training using 100 epochs, we decrease the learning rate by a factor of 10 at the 80th, 90th, and 95th epoch. We observe that SGD short training has the worst performance compared to the others while SRSGD short training yields either comparable or even better results than SGD full training.

## F.3 ADDITIONAL EXPERIMENTAL RESULTS

Figure 10 shows error rate vs. reduction in epochs for all models trained on CIFAR10 and ImageNet. It is a more complete version of Figure 4 in the main text. Table 16 and Table 17 provide detailed test errors vs. number of training epoch reduction reported in Figure 4 and Figure 10 . We also conduct an additional ablation study of error rate vs. reduction in epochs for CIFAR100 and include the results in Figure 11 and Table 18 below.

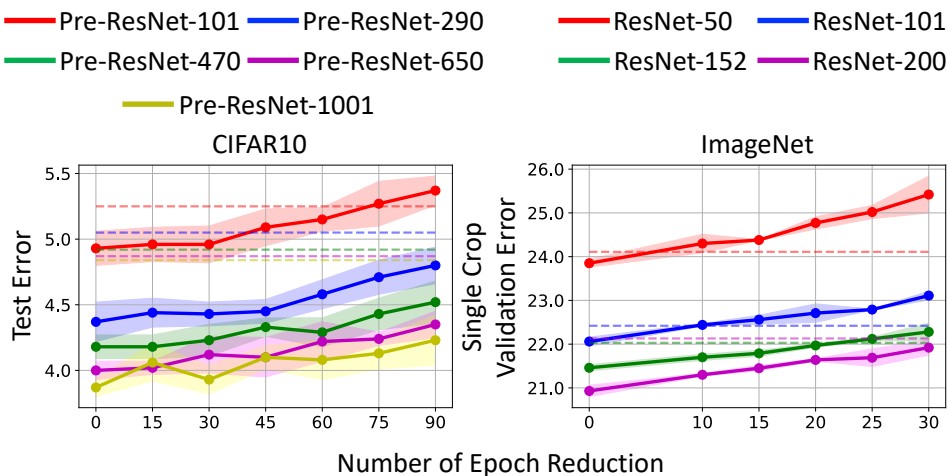

Figure 10: Test error vs. number of training epochs. Dashed lines are test errors of SGD trained in 200 epochs for CIFAR10 and 90 epochs for ImageNet. For CIFAR, SRSGD with fewer epochs achieves comparable results to SRSGD with 200 epochs. For ImageNet, training with less epochs slightly decreases the performance of SRSGD but still achieves comparable results to 200-epoch SGD.

Table 16: Test error vs. number of training epochs for CIFAR10

| Network | 110 (90 less) | 125 (75 less) | 140 (60 less) | 155 (45 less) | 170 (30 less) | 185 (15 less) | 200 (full trainings) |
|---|---|---|---|---|---|---|---|
| Pre-ResNet-110 | $5.37 \pm 0.11$ | $5.27 \pm 0.17$ | $5.15 \pm 0.09$ | $5.09 \pm 0.14$ | $4.96 \pm 0.14$ | $4.96 \pm 0.13$ | $\mathbf{4.93 \pm 0.13}$ |
| Pre-ResNet-290 | $4.80 \pm 0.14$ | $4.71 \pm 0.13$ | $4.58 \pm 0.11$ | $4.45 \pm 0.09$ | $4.43 \pm 0.09$ | $4.44 \pm 0.11$ | $\mathbf{4.37 \pm 0.15}$ |
| Pre-ResNet-470 | $4.52 \pm 0.16$ | $4.43 \pm 0.12$ | $4.29 \pm 0.11$ | $4.33 \pm 0.07$ | $4.23 \pm 0.12$ | $\mathbf{4.18 \pm 0.09}$ | $\mathbf{4.18 \pm 0.09}$ |
| Pre-ResNet-650 | $4.35 \pm 0.10$ | $4.24 \pm 0.05$ | $4.22 \pm 0.15$ | $4.10 \pm 0.15$ | $4.12 \pm 0.14$ | $4.02 \pm 0.05$ | $\mathbf{4.00 \pm 0.07}$ |
| Pre-ResNet-1001 | $4.23 \pm 0.19$ | $4.13 \pm 0.12$ | $4.08 \pm 0.15$ | $4.10 \pm 0.09$ | $3.93 \pm 0.11$ | $4.06 \pm 0.14$ | $\mathbf{3.87 \pm 0.07}$ |

Table 17: Top 1 single crop validation error vs. number of training epochs for ImageNet

| Network | 60 (30 less) | 65 (25 less) | 70 (20 less) | 75 (15 less) | 80 (10 less) | 90 (full trainings) |
|---|---|---|---|---|---|---|
| ResNet-50 | $25.42 \pm 0.42$ | $25.02 \pm 0.15$ | $24.77 \pm 0.14$ | $24.38 \pm 0.01$ | $24.30 \pm 0.21$ | $\mathbf{23.85 \pm 0.09}$ |
| ResNet-101 | $23.11 \pm 0.10$ | $22.79 \pm 0.01$ | $22.71 \pm 0.21$ | $22.56 \pm 0.10$ | $22.44 \pm 0.03$ | $\mathbf{22.06 \pm 0.10}$ |
| ResNet-152 | $22.28 \pm 0.20$ | $22.12 \pm 0.04$ | $21.97 \pm 0.04$ | $21.79 \pm 0.07$ | $21.70 \pm 0.07$ | $\mathbf{21.46 \pm 0.07}$ |
| ResNet-200 | $21.92 \pm 0.17$ | $21.69 \pm 0.20$ | $21.64 \pm 0.03$ | $21.45 \pm 0.06$ | $21.30 \pm 0.03$ | $\mathbf{20.93 \pm 0.13}$ |

Table 18: Test error vs. number of training epochs for CIFAR100

| Network | 110 (90 less) | 125 (75 less) | 140 (60 less) | 155 (45 less) | 170 (30 less) | 185 (15 less) | 200 (full trainings) |
|---|---|---|---|---|---|---|---|
| Pre-ResNet-110 | $24.06 \pm 0.26$ | $23.82 \pm 0.24$ | $23.82 \pm 0.28$ | $23.58 \pm 0.18$ | $23.69 \pm 0.21$ | $23.73 \pm 0.34$ | $\mathbf{23.49 \pm 0.23}$ |
| Pre-ResNet-290 | $21.96 \pm 0.45$ | $21.77 \pm 0.21$ | $21.67 \pm 0.37$ | $21.56 \pm 0.33$ | $\mathbf{21.38 \pm 0.44}$ | $21.47 \pm 0.32$ | $21.49 \pm 0.27$ |
| Pre-ResNet-470 | $21.35 \pm 0.17$ | $21.25 \pm 0.17$ | $21.21 \pm 0.18$ | $21.09 \pm 0.28$ | $20.87 \pm 0.28$ | $20.81 \pm 0.32$ | $\mathbf{20.71 \pm 0.32}$ |
| Pre-ResNet-650 | $21.18 \pm 0.27$ | $20.95 \pm 0.13$ | $20.77 \pm 0.31$ | $20.61 \pm 0.19$ | $20.57 \pm 0.13$ | $20.47 \pm 0.07$ | $\mathbf{20.36 \pm 0.25}$ |
| Pre-ResNet-1001 | $20.27 \pm 0.17$ | $20.03 \pm 0.13$ | $20.05 \pm 0.22$ | $19.74 \pm 0.18$ | $19.71 \pm 0.22$ | $\mathbf{19.67 \pm 0.22}$ | $19.75 \pm 0.11$ |

# G  IMPACT OF RESTARTING FREQUENCY FOR IMAGENET AND CIFAR100

## G.1  IMPLEMENTATION DETAILS

For the CIFAR10 experiments on Pre-ResNet-290 in Figure 5 in the main text, as well as the CIFAR100 and ImageNet experiments in Figure 14 and 15 in this Appendix, we vary the initial restarting frequency $F_1$. Other settings are the same as described in Section D above.

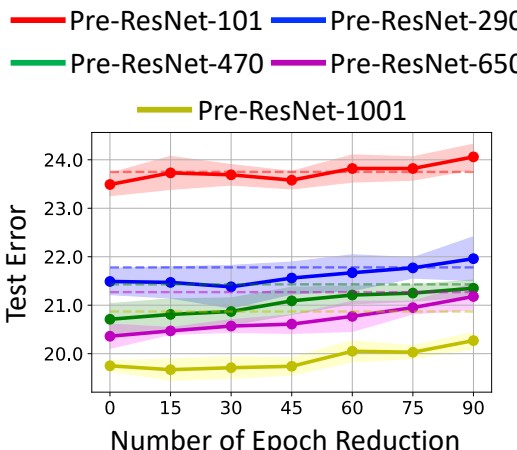

Figure 11: Test error vs. number of epoch reduction in CIFAR100 training. The dashed lines are test errors of the SGD baseline. For CIFAR100, SRSGD training with fewer epochs can achieve comparable results to SRSGD training with full 200 epochs. In some cases, such as with Pre-ResNet-290 and 1001, SRSGD training with fewer epochs achieves even better results than SRSGD training with full 200 epochs.

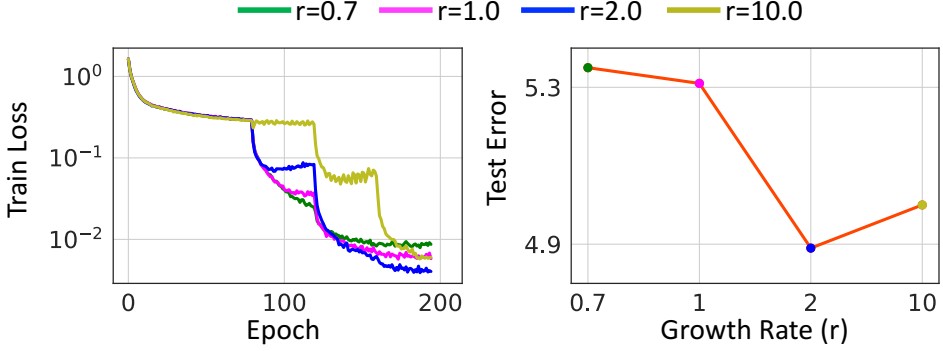

Figure 12: Training loss (left) and test error (right) of Pre-ResNet-110 trained on CIFAR10 with different growth rate $r$ (linear schedule). Here, we fix the initial restarting frequency $F_1 = 30$ for all trainings. Increasing the restarting frequency during training yields better results than decreasing the restarting frequency, but increasing the restarting frequency too fast and too much also diminishes the performance of SRSGD.

### G.2   IMPACT OF THE GROWTH RATE $r$

We do an ablation study for the growth rate $r$ to understand its impact on the behavior of SRSGD. We choose a case study of training a Pre-ResNet-110 on CIFAR10 using SRSGD with a linear schedule scheme for the restarting frequency. We fix the initial restarting frequency $F_1 = 30$ and vary the growth rate $r$. We choose $r$ from the set of $\{0.7, 1.0, 2.0, 10.0\}$. These values of $r$ represent four different scenarios. When $r = 0.7$, the restarting frequency decreases every time the learning rate is reduced by a factor of 10. When $r = 1.0$, the restarting frequency stays constant during the training. When $r = 2.0$, the restarting frequency increases every time the learning rate is reduced by a factor of 10. Finally, when $r = 10.0$, it is similar to when $r = 2.0$, but the restarting frequency increases much faster and to larger values. Figure 12 summarizes the results of our ablation study. We observe that for CIFAR10, decreasing the restarting frequency or keeping it constant during training yields worse results than increasing the restarting frequency. However, increasing the restarting frequency too much also diminishes the performance of SRSGD.

### G.3 ADDITIONAL EXPERIMENTAL RESULTS

To complete our study on the impact of restarting frequency in Section 5.2 in the main text, we examine the case of CIFAR100 and ImageNet in this section. We summarize our results in Figure 14 and 15 below. Also, Figure 13 is a more detailed version of Figure 5 in the main text.

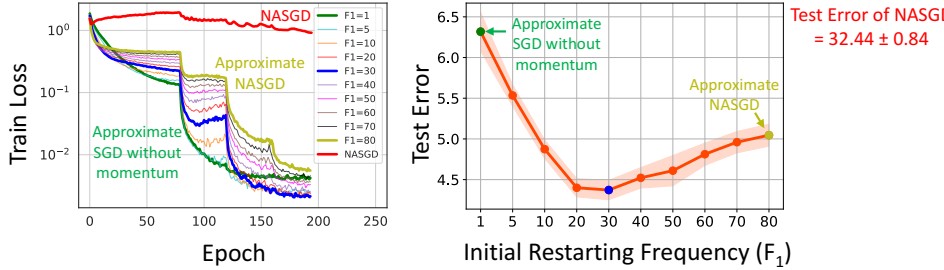

Figure 13: Training loss (left) and test error (right) of Pre-ResNet-290 trained on CIFAR10 with different initial restarting frequencies $F_1$ (linear schedule). SRSGD with small $F_1$ approximates SGD without momentum, while SRSGD with large $F_1$ approximates NASGD. The training loss curve and test accuracy of NASGD are shown in red and confirm the result of Theorem 1 that NASGD accumulates error due to the stochastic gradients.

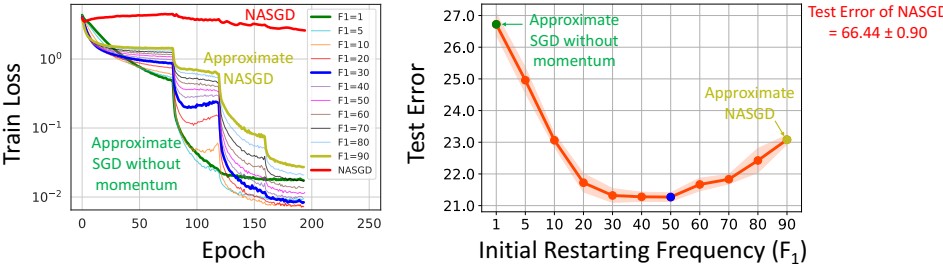

Figure 14: Training loss and test error of Pre-ResNet-290 trained on CIFAR100 with different initial restarting frequencies $F_1$ (linear schedule). SRSGD with small $F_1$ approximates SGD without momentum, while SRSGD with large $F_1$ approximates NASGD. The training loss curve and test accuracy of NASGD are shown in red and confirm the result of Theorem 1 that NASGD accumulates error due to the stochastic gradients.

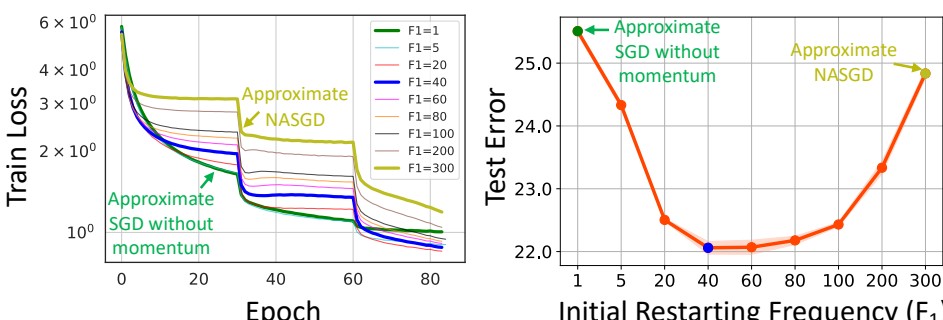

Figure 15: Training loss and test error of ResNet-101 trained on ImageNet with different initial restarting frequencies $F_1$. We use linear schedule and linearly decrease the restarting frequency to 1 at the last learning rate. SRSGD with small $F_1$ approximates SGD without momentum, while SRSGD with large $F_1$ approximates NASGD.

# H FULL TRAINING WITH LESS EPOCHS AT THE INTERMEDIATE LEARNING RATES

We explore SRSGD full training (200 epochs on CIFAR and 90 epochs on ImageNet) with less number of epochs at the intermediate learning rates and report the results in Table 19, 20, 21 and Figure 16, 17, 18 below. The settings and implementation details here are similar to those in Section F, but using all 200 epochs for CIFAR experiments and 90 epochs for ImageNet experiments.

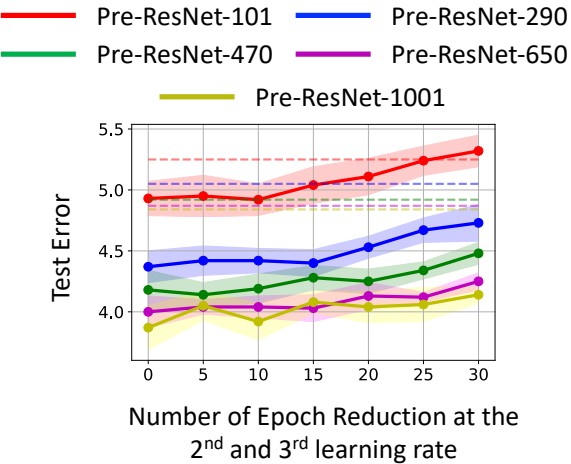

Figure 16: Test error when using new learning rate schedules with less training epochs at the 2nd and 3rd learning rate for CIFAR10. We still train in full 200 epochs in this experiment. On the x-axis, 10, for example, means we reduce the number of training epochs by 10 at each intermediate learning rate, i.e. the 2nd and 3rd learning rate. The dashed lines are test errors of the SGD baseline.

Table 19: Test error when using new learning rate schedules with less training epochs at the 2nd and 3rd learning rate for CIFAR10. We still train in full 200 epochs in this experiment. In the table, 80-90-100, for example, means we reduce the learning rate by factor of 10 at the 80th, 90th, and 100th epoch.

| Network | 80 - 90 - 100 | 80 - 95 - 110 | 80 - 100 - 120 | 80 - 105 - 130 | 80 - 110 - 140 | 80 - 115 - 150 | 80 - 120 - 160 |
|---|---|---|---|---|---|---|---|
| Pre-ResNet-110 | $5.32 \pm 0.14$ | $5.24 \pm 0.17$ | $5.11 \pm 0.13$ | $5.04 \pm 0.15$ | $\mathbf{4.92 \pm 0.15}$ | $4.95 \pm 0.12$ | $4.93 \pm 0.13$ |
| Pre-ResNet-290 | $4.73 \pm 0.13$ | $4.67 \pm 0.12$ | $4.53 \pm 0.10$ | $4.40 \pm 0.11$ | $4.42 \pm 0.09$ | $4.42 \pm 0.10$ | $\mathbf{4.37 \pm 0.15}$ |
| Pre-ResNet-470 | $4.48 \pm 0.16$ | $4.34 \pm 0.10$ | $4.25 \pm 0.12$ | $4.28 \pm 0.10$ | $4.19 \pm 0.10$ | $\mathbf{4.14 \pm 0.07}$ | $4.18 \pm 0.09$ |
| Pre-ResNet-650 | $4.25 \pm 0.13$ | $4.12 \pm 0.06$ | $4.13 \pm 0.09$ | $4.03 \pm 0.11$ | $4.04 \pm 0.11$ | $4.04 \pm 0.04$ | $\mathbf{4.00 \pm 0.07}$ |
| Pre-ResNet-1001 | $4.14 \pm 0.18$ | $4.06 \pm 0.12$ | $4.04 \pm 0.15$ | $4.08 \pm 0.09$ | $3.92 \pm 0.13$ | $4.05 \pm 0.14$ | $\mathbf{3.87 \pm 0.07}$ |

Table 20: Test error when using new learning rate schedules with less training epochs at the 2nd and 3rd learning rate for CIFAR100. We still train in full 200 epochs in this experiment. In the table, 80-90-100, for example, means we reduce the learning rate by factor of 10 at the 80th, 90th, and 100th epoch.

| Network | 80 - 90 - 100 | 80 - 95 - 110 | 80 - 100 - 120 | 80 - 105 - 130 | 80 - 110 - 140 | 80 - 115 - 150 | 80 - 120 - 160 |
|---|---|---|---|---|---|---|---|
| Pre-ResNet-110 | $23.65 \pm 0.14$ | $23.96 \pm 0.26$ | $23.97 \pm 0.31$ | $23.53 \pm 0.13$ | $23.57 \pm 0.36$ | $23.68 \pm 0.24$ | $\mathbf{23.49 \pm 0.23}$ |
| Pre-ResNet-290 | $21.94 \pm 0.44$ | $21.71 \pm 0.27$ | $21.55 \pm 0.40$ | $21.44 \pm 0.31$ | $\mathbf{21.37 \pm 0.45}$ | $21.47 \pm 0.32$ | $21.49 \pm 0.27$ |
| Pre-ResNet-470 | $21.29 \pm 0.11$ | $21.21 \pm 0.14$ | $21.17 \pm 0.18$ | $20.99 \pm 0.28$ | $20.81 \pm 0.22$ | $20.80 \pm 0.31$ | $\mathbf{20.71 \pm 0.32}$ |
| Pre-ResNet-650 | $21.11 \pm 0.24$ | $20.91 \pm 0.17$ | $20.66 \pm 0.33$ | $20.52 \pm 0.18$ | $20.51 \pm 0.16$ | $20.43 \pm 0.10$ | $\mathbf{20.36 \pm 0.25}$ |
| Pre-ResNet-1001 | $20.21 \pm 0.15$ | $20.00 \pm 0.11$ | $19.86 \pm 0.19$ | $\mathbf{19.55 \pm 0.19}$ | $19.69 \pm 0.21$ | $19.60 \pm 0.17$ | $19.75 \pm 0.11$ |

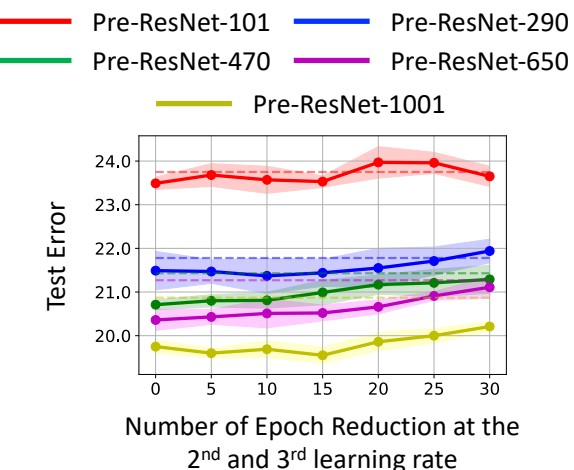

Figure 17: Test error when using new learning rate schedules with less training epochs at the 2nd and 3rd learning rate for CIFAR100. We still train in full 200 epochs in this experiment. On the x-axis, 10, for example, means we reduce the number of training epochs by 10 at each intermediate learning rate, i.e. the 2nd and 3rd learning rate. The dashed lines are test errors of the SGD baseline.

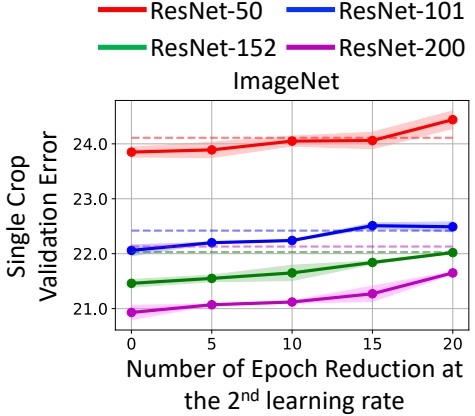

Figure 18: Test error when using new learning rate schedules with less training epochs at the 2nd learning rate for ImageNet. We still train in full 90 epochs in this experiment. On the x-axis, 10, for example, means we reduce the number of training epochs by 10 at the 2nd learning rate. The dashed lines are test errors of the SGD baseline.

Table 21: Top 1 single crop validation error when using new learning rate schedules with less training epochs at the 2nd learning rate for ImageNet. We still train in full 90 epochs in this experiment. In the table, 30-40, for example, means we reduce the learning rate by factor of 10 at the 30th and 40th epoch.

| Network | 30 - 40 | 30 - 45 | 30 - 50 | 30 - 55 | 30 - 60 |
|---------|---------|---------|---------|---------|---------|
| ResNet-50 | $24.44 \pm 0.16$ | $24.06 \pm 0.15$ | $24.05 \pm 0.09$ | $23.89 \pm 0.14$ | $\mathbf{23.85 \pm 0.09}$ |
| ResNet-101 | $22.49 \pm 0.09$ | $22.51 \pm 0.05$ | $22.24 \pm 0.01$ | $22.20 \pm 0.01$ | $\mathbf{22.06 \pm 0.10}$ |
| ResNet-152 | $22.02 \pm 0.01$ | $21.84 \pm 0.03$ | $21.65 \pm 0.14$ | $21.55 \pm 0.06$ | $\mathbf{21.46 \pm 0.07}$ |
| ResNet-200 | $21.65 \pm 0.02$ | $21.27 \pm 0.14$ | $21.12 \pm 0.02$ | $21.07 \pm 0.01$ | $\mathbf{20.93 \pm 0.13}$ |

# I  VISUALIZATION OF SRSGD'S TRAJECTORY

Here we visualize the training trajectory through bad minima of SRSGD, SGD with constant momentum, and SGD. In particular, we train a neural net classifier on a swiss roll data as in Huang et al. (2019) and find bad minima along its training. Each red dot in Figure 19 represents the trained model after each 10 epochs in the training. From each red dot, we search for nearby bad local minima,

which are the blue dots. Those bad local minima achieve good training error but bad test error. We plots the trained models and bad local minima using PCA Wold et al. (1987) and t-SNE Maaten & Hinton (2008) embedding. The blue color bar is for the test accuracy of bad local minima; the red color bar is for the number of training epochs.

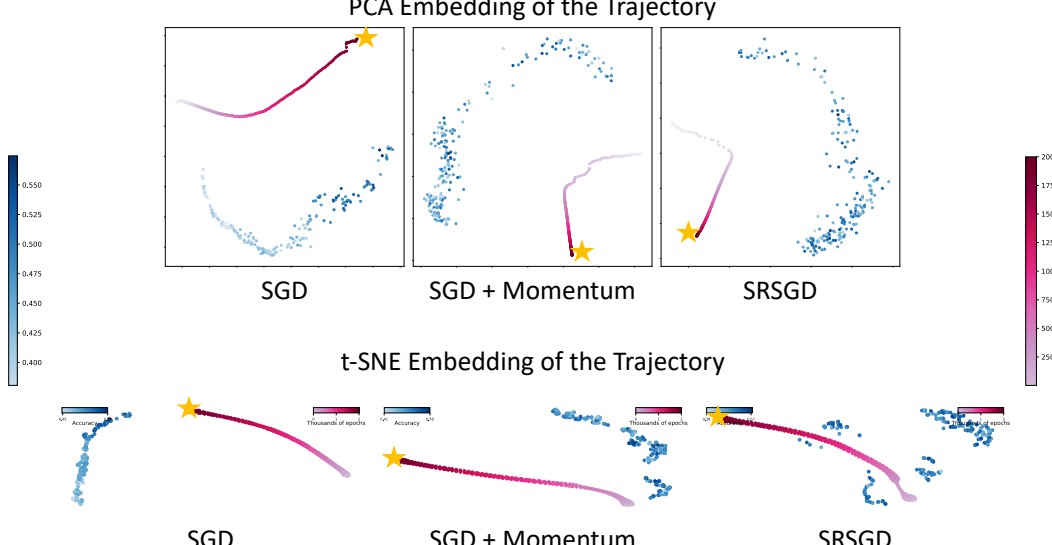

Figure 19: Trajectory through bad minima of SGD, SGD with constant momentum, and SRSGD during the training: we train a neural net classifier and plot the iterates of SGD after each ten epoch (red dots). We also plot locations of nearby "bad" minima with poor generalization (blue dots). We visualize these using PCA and t-SNE embedding. The blue color bar is for the test accuracy of bad local minima while the red color bar is for the number of training epochs. All blue dots for SGD with constant momentum and SRSGD achieve near perfect train accuracy, but with test accuracy below 59%. All blue dots for SGD achieves average train accuracy of 73.11% and with test accuracy also below 59%. The final iterate (yellow star) of SGD, SGD with constant momentum, and SRSGD achieve 73.13%, 99.25%, and 100.0% test accuracy, respectively.

(CONTINUED NEXT PAGE)

## J  SRSGD IMPLEMENTATION IN PYTORCH

```python
import torch
from .optimizer import Optimizer, required

class SRSGD(Optimizer):
    """
    Scheduled Restart SGD.
    Args:
        params (iterable): iterable of parameters to optimize
                           or dicts defining parameter groups.
        lr (float): learning rate.
        weight_decay (float, optional): weight decay (L2 penalty) (
            default: 0)
        iter_count (integer): count the iterations mod 200
    Example:
        >>> optimizer = torch.optim.SRSGD(model.parameters(), lr=0.1,
                        weight_decay=5e-4, iter_count=1)
        >>> optimizer.zero_grad()
        >>> loss_fn(model(input), target).backward()
        >>> optimizer.step()
        >>> iter_count = optimizer.update_iter()
    Formula:
        v_{t+1} = p_t - lr*g_t
        p_{t+1} = v_{t+1} + (iter_count)/(iter_count+3)*(v_{t+1} - v_t)
    """
    def __init__(self, params, lr=required, weight_decay=0.,
                 iter_count=1, restarting_iter=100):
        if lr is not required and lr < 0.0:
            raise ValueError("Invalid learning rate: {}".format(lr))
        if weight_decay < 0.0:
            raise ValueError("Invalid weight_decay value: {}".format(
                weight_decay))
        if iter_count < 1:
            raise ValueError("Invalid iter count: {}".format(iter_count))
        if restarting_iter < 1:
            raise ValueError("Invalid iter total: {}".format(
                restarting_iter))

        defaults = dict(lr=lr, weight_decay=weight_decay, iter_count=
            iter_count,
                        restarting_iter=restarting_iter)
        super(SRSGD, self).__init__(params, defaults)

    def __setstate__(self, state):
        super(SRSGD, self).__setstate__(state)

    def update_iter(self):
        idx = 1
        for group in self.param_groups:
            if idx == 1:
                group['iter_count'] += 1
                if group['iter_count'] >= group['restarting_iter']:
                    group['iter_count'] = 1
            idx += 1
        return group['iter_count'], group['restarting_iter']

    def step(self, closure=None):
        """
        Perform a single optimization step.
        Arguments: closure (callable, optional): A closure that
                   reevaluates the model and returns the loss.
        """
        loss = None
        if closure is not None:
```

```
            loss = closure()

        for group in self.param_groups:
            weight_decay = group['weight_decay']
            momentum = (group['iter_count'] - 1.)/(group['iter_count'] +
                2.)
            for p in group['params']:
                if p.grad is None:
                    continue
                d_p = p.grad.data
                if weight_decay !=0:
                    d_p.add_(weight_decay, p.data)

                param_state = self.state[p]

                if 'momentum_buffer' not in param_state:
                    buf0 = param_state['momentum_buffer'] = torch.clone(p
                        .data).detach()
                else:
                    buf0 = param_state['momentum_buffer']

                buf1 = p.data - group['lr']*d_p
                p.data = buf1 + momentum*(buf1 - buf0)
                param_state['momentum_buffer'] = buf1

        iter_count, iter_total = self.update_iter()

        return loss
```

## K   SRSGD IMPLEMENTATION IN KERAS

```
import numpy as np
import tensorflow as tf
from keras import backend as K
from keras.optimizers import Optimizer
from keras.legacy import interfaces
if K.backend() == 'tensorflow':
    import tensorflow as tf

class SRSGD(Optimizer):
    """Scheduled Restart Stochastic gradient descent optimizer.
    Includes support for Nesterov momentum
    and learning rate decay.
    # Arguments
        learning_rate: float >= 0. Learning rate.
    """

    def __init__(self, learning_rate=0.01, iter_count=1, restarting_iter
        =40, **kwargs):
        learning_rate = kwargs.pop('lr', learning_rate)
        self.initial_decay = kwargs.pop('decay', 0.0)
        super(SRSGD, self).__init__(**kwargs)
        with K.name_scope(self.__class__.__name__):
            self.iterations = K.variable(0, dtype='int64', name='
                iterations')
            self.learning_rate = K.variable(learning_rate, name='
                learning_rate')
            self.decay = K.variable(self.initial_decay, name='decay')
            # for srsgd
            self.iter_count = K.variable(iter_count, dtype='int64', name=
                'iter_count')
            self.restarting_iter = K.variable(restarting_iter, dtype='
                int64', name='restarting_iter')
        self.nesterov = nesterov
```

```python
@interfaces.legacy_get_updates_support
@K.symbolic
def get_updates(self, loss, params):
    grads = self.get_gradients(loss, params)
    self.updates = [K.update_add(self.iterations, 1)]

    momentum = (K.cast(self.iter_count, dtype=K.dtype(self.decay)) -
        1.)/(K.cast(self.iter_count, dtype=K.dtype(self.decay)) + 2.)

    lr = self.learning_rate
    if self.initial_decay > 0:
        lr = lr * (1. / (1. + self.decay * K.cast(self.iterations,
                                                  K.dtype(self.decay)
                                                  )))
    # momentum
    shapes = [K.int_shape(p) for p in params]

    moments = [K.variable(value=K.get_value(p), dtype=K.dtype(self.
        decay), name='moment_' + str(i))
               for (i, p) in enumerate(params)]

    self.weights = [self.iterations] + moments + [self.iter_count]
    for p, g, m in zip(params, grads, moments):
        v = p - lr * g
        new_p = v + momentum * (v - m)
        self.updates.append(K.update(m, v))

        # Apply constraints.
        if getattr(p, 'constraint', None) is not None:
            new_p = p.constraint(new_p)

        self.updates.append(K.update(p, new_p))

    condition = K.all(K.less(self.iter_count, self.restarting_iter))
    new_iter_count = K.switch(condition, self.iter_count + 1, self.
        iter_count - self.restarting_iter + 1)
    self.updates.append(K.update(self.iter_count, new_iter_count))

    return self.updates

def get_config(self):
    config = {'learning_rate': float(K.get_value(self.learning_rate))
        ,
              'decay': float(K.get_value(self.decay)),
              'iter_count': int(K.get_value(self.iter_count)),
              'restarting_iter': int(K.get_value(self.restarting_iter
                  ))}
    base_config = super(SRSGD, self).get_config()
    return dict(list(base_config.items()) + list(config.items()))
```

