# OpenReview forum: "Scheduled Restart Momentum for Accelerated Stochastic Gradient Descent"
_ICLR.cc/2021/Conference — Reject_

### Official Review · AnonReviewer3 · 2020-10-24
**I tend to accept this paper due to its impressive experimental results. However, there are some theoretical issues need to be addressed.**

**Rating:** 6
**Confidence:** 4

**Review:**

This paper proposes to restart the momentum parameter in SGD (with Nesterov's momentum) according to some carefully chosen schedules in training deep neural network, which is named as SRSGD. Two different restarting schedules are proposed: linear schedule and exponential schedule. The strong point of this paper is its extensive experimental evaluations, which justify that SRSGD significantly improves the convergence speed and generalization over standard momentum SGD. The empirical analysis also sheds some light on the parameter tuning and interpretation of SRSGD.

This paper is well-written and easy to follow. Overall, I like this work and see the following strengths of this paper:
- The proposed approach significantly outperforms standard momentum SGD in several DNN tasks and the advantage grows with the depth of DNN, which is appealing to practitioners.
- The experiments are comprehensive and constructive. Various image classification benchmarks were tested and the confidence intervals are also provided. I appreciate the effort the authors put into that.
- Although the proposed restarting schedules seem to be hard to tune in practice, the authors provide empirical analysis on the impacts of different parameters, which gives some guidance.

On the other side, I have some concerns on the theoretical aspects:

- Theorem 1 (and 2) assumes a bounded variance of the stochastic gradient, which may not be true for ERM (e.g., for least squares "Jain, P., et al. (2018). Accelerating Stochastic Gradient Descent for Least Squares Regression. In COLT, pages 545–604."). Thus, the "mini-batch stochastic gradient" is not precise in Theorem 1 (and 2) and should be replaced with certain assumptions.
- Under the classic bounded variance assumption, there exists a nice workaround of the non-convergence issue of NASGD in Theorem 1: the AC-SA approach proposed in "Lan, G. (2012). An optimal method for stochastic composite optimization. Math. Program., 133(1-2):365–397.". AC-SA is basically NASGD with some parameter constraints. It can be shown that AC-SA maps to scheme (5) with $\frac{t_k - 1}{t_{k+1}} = \frac{\beta_k - 1}{\beta_{k+1}}$ and $s_k=\frac{\gamma_k}{\beta_k}$. Corollary 1 in (Lan, G., 2012) states that if $\frac{t_k - 1}{t_{k+1}} = \frac{k-1}{k+2}$ and $s_k= \min\{\frac{1}{2L}, \frac{C}{N^{1.5}}\}$, where $N$ is the total number of iterations, NASGD converges at the optimal $O(\frac{L}{N^2} + \frac{\sigma}{\sqrt{N}})$ rate. Thus, the non-convergence issue can be solved by simply fixing $N$ in advance and setting $s_k$ accordingly.

Due to these points, the discussion in Section 3.1 is not very insightful to me. There are some recent works studying the non-acceleration issues of SGD with HB and NAG momentum for least squares:
- Kidambi, R., et al. (2018).On the insufficiency of existing momentum schemes for Stochastic Optimization. In ICLR.
- Liu, C. and Belkin, M. (2020). Accelerating SGD with momentum for over-parameterized learning. In ICLR.

Their setting (stochastic unbounded variance) is somewhat closer to ERM. Since this line of work is quite relevant, a proper literature review and adding relevant bibliographical entries might be necessary.

Other comments:
- In fact, both the constant scheme (3) and the variable-pararmeter one are referred to as NAG in optimization community. In the strongly convex case, NAG uses a constant $\mu$ to achieve acceleration and is a non-monotone method (this relates to the claim under ARNAG in Section 2). See his book "Nesterov, Y. (2018). Lectures on convex optimization, volume 137. Springer".
- "Constant momentum achieves state-of-the-art result in DL (Sutskever et al., 2013)". Actually, Sutskever et al. (2013) tried a variable momentum similar to the one in the paper.
- The second sentence in Related Work. "SGD with scheduled momentum" -> "These works all use constant momentum"?
- The discussion about NAG with $\delta$-inexact oracle does not seem to be quite connected with other parts of the paper since this inexactness is deterministic.
- "dom(J)" is not defined in Appendix A.

---

> ### Author Response · Authors · 2020-11-13
> **Response to Reviewer 3**
>
> Thank you for your valuable feedback and thoughtful reviews. We have revised our manuscript as you suggested, and the revised part has been highlighted in blue.  Below we address your concerns.
>
> ---
>
> Q1. “Theorem 1 (and 2) assumes a bounded variance of the stochastic gradient, which may not be true for ERM (e.g., for least squares "Jain, P., et al. (2018). Accelerating Stochastic Gradient Descent for Least Squares Regression. In COLT, pages 545–604."). Thus, the "mini-batch stochastic gradient" is not precise in Theorem 1 (and 2) and should be replaced with certain assumptions.”
>
> Answer: Thanks for pointing this to us. As you suggested, we have replaced the “mini-batch stochastic gradient” with the bounded variance assumptions in the statement of our theorems. We have also included additional theoretical analysis under the conditions in "Jain, P., et al. (2018). Accelerating Stochastic Gradient Descent for Least Squares Regression. In COLT, pages 545–604" as future work.
>
> To the best of our knowledge, the bounded variance assumption is quite common in analyzing SGD-type algorithms. For instance, that assumption is used in “S. Bubeck. Convex optimization: Algorithms and complexity. Foundations and Trends in Machine Learning, 8(3-4):231–357, 2015.” “L. Bottou, F. E. Curtis, and J. Nocedal. Optimization methods for large-scale machine learning. arxiv:1606.04838, 2016.”
>
> ---
>
> Q2. “Under the classic bounded variance assumption, there exists a nice workaround of the non-convergence issue of NASGD in Theorem 1: the AC-SA approach proposed in "Lan, G. (2012). An optimal method for stochastic composite optimization. Math. Program., 133(1-2):365–397.". AC-SA is basically NASGD with some parameter constraints. It can be shown that AC-SA maps to scheme (5) with $(t_{k}-1)/t_{k+1}=(\beta_{k}-1)/\beta_{k+1}$ and $s_k=\gamma_k/\beta_k$. Corollary 1 in (Lan, G., 2012) states that if $(t_k-1)/t_{k+1}=(k-1)/k+2$ and $s_k=\min\{1/2L, C/N^{1.5}\}$, where $N$ is the total number of iterations, NASGD converges at the optimal $O(L/N^2+\sigma/\sqrt{N})$ rate. Thus, the non-convergence issue can be solved by simply fixing $N$ in advance and setting $s_k$ accordingly.”
>
> Answer: We agree that the non-convergence issue can be solved by fixing $N$ in advance and setting $s_k$ accordingly. To our understanding, Lan. G. (2012) proposes a mirror descent type of algorithm, and it uses a decay learning rate, which might have some challenges that need to be overcome in order to be efficient for improving the training of deep neural networks.
>
> ---
>
> Q3. “Due to these points in Q1 and Q2, the discussion in Section 3.1 is not very insightful to me.”
>
> Answer: In section 3.1, we aim to convey that the bound of the NASGD is out of control when the stochastic gradient of bounded variance is used. Numerically, we showed the divergence of training the logistic regression model for the MNIST classification task. Our proof helps explain the divergence phenomenon. In particular, from our analysis, we can see that the uncontrollable error bound is due to the fact that the time-varying momentum gets closer and closer to 1 as iteration goes. Based on this analysis, we applied the restart scheme, which makes the momentum parameter to be smaller than a constant that is less than 1. We believe that the discussion in section 3.1 provides a better understanding of NASGD and motivates the need for the SRSGD that we propose. We have clarified these points in the revised manuscript.
>
> ---
>
> Q4. “There are some recent works studying the non-acceleration issues of SGD with HB and NAG momentum for least squares: 1) Kidambi, R., et al. (2018).On the insufficiency of existing momentum schemes for Stochastic Optimization. In ICLR. 2) Liu, C. and Belkin, M. (2020). Accelerating SGD with momentum for over-parameterized learning. In ICLR. Their setting (stochastic unbounded variance) is somewhat closer to ERM. Since this line of work is quite relevant, a proper literature review and adding relevant bibliographical entries might be necessary.”
>
> Answer: We have added all these papers into the related work and reference of our revised manuscript. Thanks for pointing these papers to us.
>
> ---
>
> Q5. “In fact, both the constant scheme (3) and the variable-parameter one are referred to as NAG in optimization community. In the strongly convex case, NAG uses a constant mu to achieve acceleration and is a non-monotone method (this relates to the claim under ARNAG in Section 2). See his book "Nesterov, Y. (2018). Lectures on convex optimization, volume 137. Springer".”
>
> Answer: We have clarified that we consider NAG with a time-varying momentum parameter at the beginning of our revised paper.

---

> > ### Author Response · Authors · 2020-11-13
> > **Response to Reviewer 3 -- Cont'd**
> >
> > Q6. “"Constant momentum achieves state-of-the-art results in DL (Sutskever et al., 2013)". Actually, Sutskever et al. (2013) tried a variable momentum similar to the one in the paper.”
> >
> > Answer: Yes, Sutskever et al. (2013) tried variable momentum similar to the time-varying one in their experiments. We have replaced the sentence “Constant momentum achieves state-of-the-art result in DL” with “In many DL practices constant momentum achieves state-of-the-art result in DL. For instance, training DNNs for image classification.”
> >
> > ---
> >
> > Q7. “The second sentence in Related Work. "SGD with scheduled momentum" -> "These works all use constant momentum"?”
> >
> > Answer: We have corrected the second statement "These works all use constant momentum"?” in the revised manuscript.
> >
> > ----
> >
> > Q8. “The discussion about NAG with delta-inexact oracle does not seem to be quite connected with other parts of the paper since this inexactness is deterministic.”
> >
> > Answer: We provided a short review of the result of NAG with delta-inexact oracle in the appendix to give the reader a more complete picture of NAG with an inexact gradient. If you think it is not necessary, we can remove it from the appendix.
> >
> > ---
> >
> > Q9. “"dom(J)" is not defined in Appendix A.”
> >
> > Answer: It is a typo. Thanks for pointing out this.
> >
> >
> > =======================
> >
> > We hope we have cleared your concerns about our work. We have also revised our manuscript according to your comments, and we would appreciate it if we can get your further feedback at your earliest convenience.

---

> > > ### Comment · AnonReviewer3 · 2020-11-16
> > > **Response to Author**
> > >
> > > Thank you for the detailed feedback and revision. For Section 3.1, I think the authors want to convey that NASGD with constant $s_k$ suffers from error accumulation, which leads to an $O(k(\delta + R^2))$ bound similar to the results in Devolder et al. (2014) (it seems that $R$ is not defined in the proof? If it is an assumed upper bound of gradient norm, it should be clearly stated in the theorem). In this case, the section title "NASGD does not converge" is not appropriate since plain SGD with constant stepsize also does not converge (to any accuracy). A decaying stepsize is necessary for convergence (to any accuracy) in this setting. The section title could be something like "NASGD suffers from error accumulation".
> > >
> > > I actually feel that this error accumulation bound can be directly obtained by choosing $\gamma_t = \beta_t \frac{C}{L}, \beta_t = \frac{t+1}{2}$ in AC-SA for some constant $C$, which is equivalent to choosing constant $s_k$ in NASGD. And then, from Theorem 2(a) in Lan, G. (2012), the guarantee becomes $O(t(M^2 + \sigma^2))$.
> > >
> > > Moreover, there are some issues in the proof of Lemma 4: (20) cannot be obtained from (25).

---

> > > > ### Author Response · Authors · 2020-11-16
> > > > **Thank you for your reponse, and response to your further concerns**
> > > >
> > > > Thank you for your very valuable feedback and very thoughtful and careful review again, which indeed significantly improves the quality of our paper. We appreciate it. Below we address your further concerns.
> > > >
> > > > ---
> > > >
> > > > Q1. “For Section 3.1, I think the authors want to convey that NASGD with constant $s_k$ suffers from error accumulation, which leads to an $O(k(\delta + R^2))$ bound similar to the results in Devolder et al. (2014) (it seems that $R$ is not defined in the proof? If it is an assumed upper bound of gradient norm, it should be clearly stated in the theorem). In this case, the section title "NASGD does not converge" is not appropriate since plain SGD with constant stepsize also does not converge (to any accuracy). A decaying stepsize is necessary for convergence (to any accuracy) in this setting. The section title could be something like "NASGD suffers from error accumulation".”
> > > >
> > > > Answer: As you suggested, we have replaced “NASGD does not converge” with “NASGD suffers from error accumulation” in our revision. Also, we have stated the gradient norm bounded by $R$ in the revised manuscript.
> > > >
> > > > ---
> > > >
> > > > Q2. “I actually feel that this error accumulation bound can be directly obtained by choosing $\gamma_t = \beta_t \frac{C}{L}, \beta_t = \frac{t+1}{2}$ in AC-SA for some constant $C$, which is equivalent to choosing constant $s_k$ in NASGD. And then, from Theorem 2(a) in Lan, G. (2012), the guarantee becomes $O(t(M^2 + \sigma^2))$.”
> > > >
> > > > Answer: Thank you for clarifying this. We have added this point to our revision.
> > > >
> > > > ----
> > > >
> > > > Q3. “Moreover, there are some issues in the proof of Lemma 4: (20) cannot be obtained from (25).”
> > > >
> > > > Answer: There are a few typos in our proof, and we have fixed them in the revised manuscript. Basically, notations $\tilde{v}^{k+1}$ and $v^{k+1}$ get confused in (23) and (25) in our proof. Again, we do appreciate you for pointing out this.
> > > >
> > > > ---
> > > >
> > > > We hope we have cleared your further concerns about our work. We have also revised our manuscript according to your comments, and we would appreciate it if we can get your further feedback at your earliest convenience.

---

### Official Review · AnonReviewer4 · 2020-10-24
**The theoretical analysis is not convincing.**

**Rating:** 4
**Confidence:** 4

**Review:**

To summary, the authors proposed an algorithm named Scheduled Restart SGD (SRSGD) to solve the empirical risk minimization problem. They provided a theoretical analysis along with experiments to support the benefits of their new algorithm.

The good points are:

- The authors proposed a fairly new algorithm that integrates scheduled restart Nesterov accelerated gradient (NAG) momentum with SGD.
- They conducted experiments to show the benefit of SRSGD: reduce overfitting and the training time, generalize better as the neural network grows deeper,..
- They discussed the theoretical aspects and provided a convergence analysis.

The weak points are:

- The idea of using Nesterov momentum for SGD is not new. (To be more clear, the algorithm SGD with Nesterov momentum is already in PyTorch’s library [[1]]). As the authors said, the restart momentum schedule is also in literature as well. The novelty (as far as I understand), is only combining these things to SGD, which is a well-known algorithm.
- There are some specific reasons that make the theoretical analysis of the paper not convincing (please see details below). Moreover, the convergence rate doesn’t reflect any novelty. Therefore, their analysis can not persuade the readers.
- There are clearly some benefits of SRSGD, however, it requires two additional hyper-parameters compared to other adaptive methods and SGD with momentum (as in Section 4.1). This could increase the tuning time and could be a disadvantage of SRSGD.

Because of the weak points stated above, I cannot accept this paper at the moment. The detail concern and comments are below:

- About Theorem 1: The authors claimed that they 'proved the error accumulation of Nesterov accelerated SGD’, as a reason to use the restart momentum schedule. However, Theorem 1 states that $\mathbb{E} (f(w^k) - f(w^*)) = O (k)$, which means that $\mathbb{E} (f(w^k) - f(w^*)) \leq C \cdot k$ for a positive constant $C$ (by their definition in Notation, page 3). This gives no information on the non-convergence of NASGD. I checked the appendix to look closer, however the proof is not rigorous enough to see what the authors want to prove.

- About Theorem 2: The authors imposed a new assumption: the set $ \mathcal{A} := $ { $ \{ k \in \mathbb{Z}^+ | \mathbb{E} f(w^{k+1}) \geq \mathbb{E} f(w^{k})  \}$ } is finite. The concerns are as follows: Firstly, it is really hard and maybe impossible to check this condition. Secondly, in the appendix, the set $\mathcal{A}$ directly involves a term $\bar{R}$, which is finite thanks to the assumption. But $\bar{R}$ can be arbitrarily large and make the right-hand side unbounded. Furthermore, the analysis also relies on another term $R$, which is $\sup_x || \nabla f(x) ||_2 $. This term could also be unbounded if the authors don’t assume the bounded gradient for the problem.

- For the above reasons, there seems to be no theoretical guarantee for the new algorithm. About the empirical aspects, it is nice that the authors provided numerous experiments using different data sets and network architectures. The improvement is understandable because SRSGD needs 2 additional hyper-parameters ($F_1$ and $r$) and requires more tuning effort. The comparison with Adam and RMSProp are limited, while they are the popular algorithms for deep learning models. The authors also claimed that the training time and overfitting can be reduced by half. I think it could be better to see this aspect compared not only to plain SGD but also to Adam and other adaptive algorithms.

Minor comments:

- It is much better if the authors state the theorem results more clearly before using the asymptotic notation. The current bound is not rigorous, and people cannot see how the final bound looks like.
- I think the 'Case study - Quadratic function’ in (8) should be written before Figure 2, so that the readers can see the problem you are optimizing easily.

[1]:  https://pytorch.org/docs/stable/optim.html

---

> ### Author Response · Authors · 2020-11-13
> **Response to Reviewer 4**
>
> Thank you for your review. We have revised our manuscript as you suggested, and the revised part has been highlighted in blue. We believe there are serious misunderstandings of our paper, which we feel is mainly due to the confusion between Nesterov momentum for SGD in PyTorch and the Nesterov accelerated gradient with time-varying momentum. Below we address your concerns.
>
> ----
>
> Q1. “The idea of using Nesterov momentum for SGD is not new. (To be more clear, the algorithm SGD with Nesterov momentum is already in PyTorch’s library [1]). As the authors said, the restart momentum schedule is also in literature as well. The novelty (as far as I understand), is only combining these things to SGD, which is a well-known algorithm.”
>
> Answer: We believe there is a misunderstanding of the momentum we studied and the momentum used in the PyTorch library. The momentum we used is scaled by a time-varying parameter, $(k-1)/(k+2)$ with $k$ being the iteration number, which achieves $1/k^2$ convergence rate for convex optimization. If we replace this time-varying momentum with a constant one, which is the Nesterov momentum in the PyTorch library, the convergence rate for convex optimization is only $1/k$. The time-varying Nesterov momentum is indeed one of the most beautiful results in the first-order optimization; simply replacing a constant with a time-varying coefficient can accelerate the convergence rate from $1/k$ to $1/k^2$.
>
> However, the time-varying momentum $(k-1)/(k+2)$ cannot accelerate SGD, as shown in Theorem 1 of our paper, it will accumulate error due to the stochastic/noisy gradients in SGD. We fix this issue by using the scheduled restart strategy, which restarts the momentum according to a well-designed schedule. This leads to the proposed SRSGD algorithm. Moreover, we prove the convergence of the proposed SRSGD, which we believe is nontrivial due to the time-varying momentum together with the scheduled restart.
>
> ----
>
> Q2. “There are clearly some benefits of SRSGD, however, it requires two additional hyper-parameters compared to other adaptive methods and SGD with momentum (as in Section 4.1). This could increase the tuning time and could be a disadvantage of SRSGD.”
>
> Answer:  We tune these two parameters on a small network and then apply the obtained parameters to other larger networks. Based on our experience, it is not difficult to tune these two parameters in practice, and we have provided the default parameters in the supplementary material and in the submitted code. Furthermore, we just have one additional hyperparameter compared to the existing SGD with constant momentum, where there is also a momentum weight to be well-tuned. Moreover, according to the ablation study on the impact of restart frequency in section 4.2, if we restart too often, SRSGD degenerates to SGD without momentum; while if we restart too infrequently, SRSGD will accumulate error. These ablation studies can give some intuition to tune the two hyperparameters of SRSGD.
>
> ----
>
> Q3. “Moreover, the convergence rate doesn’t reflect any novelty. Therefore, their analysis can not persuade the readers.”
>
> Answer: Our theoretical analysis contains two main parts. First, we prove that the time-varying momentum will accumulate error when the stochastic gradient is used. Second, we prove the convergence rate of the proposed SRSGD algorithm. The convergence rate proof of our proposed SRSGD has a major difference compared to the convergence rate proof of SGD, which mainly due to the time-varying momentum with a restart. We used an induction technique to bound the sequences’ squared sum, and we used a novel Lyapunov function to get the convergence results. Also, without extra assumption and other information, the known convergence rate for general convex optimization with stochastic gradient is only $O(1/\epsilon)$. In practice, we show that  SRSGD remarkably outperforms the benchmark SGD with constant momentum via a variety of numerical tests. Under which condition SRSGD can get a better theoretical guarantee than SGD with constant momentum is under our investigation.

---

> > ### Author Response · Authors · 2020-11-13
> > **Response to Reviewer 4 -- Cont'd**
> >
> > Q4. “About Theorem 1: The authors claimed that they ‘proved the error accumulation of Nesterov accelerated SGD’, as a reason to use the restart momentum schedule. However, Theorem 1 states that $E(f(w_k)-f(w^*))=O(k)$, which means that $E(f(w_k)-f(w^*))\leq Ck$ for a positive constant $C$ (by their definition in Notation, page 3). This gives no information on the non-convergence of NASGD. I checked the appendix to look closer, however the proof is not rigorous enough to see what the authors want to prove.”
> >
> > Answer: The theoretical results in Theorem 1 aims to convey that the bound of the NASGD is out of control. Numerically, we show the divergence of training a logistic regression model for the MNIST classification task. Our proof helps explain the numerically observed divergence phenomenon. In particular, from our analysis, we can see that the uncontrollable error bound is due to the fact that the time-varying momentum gets closer and closer to 1 as iteration goes. Based on this analysis, we apply the restart scheme, which makes the momentum parameter to be smaller than a constant that is less than 1. The uncontrollable bound result is consistent with the analysis under the delta-inexact gradient scenario in O. Devolder, F. Glineur, and Y. Nesterov. First-Order Methods of Smooth Convex Optimization with Inexact Oracle. Mathematical Programming volume 146, 37–75(2014). We have made these points clear in our revision.
> >
> > ----
> >
> > Q5. “About Theorem 2: The authors imposed a new assumption: the set A:=\{k\in Z^+|Ef(w_{k+1})\geq Ef(w_k)\} is finite. The concerns are as follows: Firstly, it is really hard and maybe impossible to check this condition. Secondly, in the appendix, the set A directly involves a term \bar{R}, which is finite thanks to the assumption. But \bar{R} can be arbitrarily large and make the right-hand side unbounded. Furthermore, the analysis also relies on another term R, which is sup_x||]\nabla f(x)||_2. This term could also be unbounded if the authors don’t assume the bounded gradient for the problem.”
> >
> > Answer: We can replace this finite set assumption with $\sum_{k\in A}(\mathbb{E}f(v_k)-\mathbb{E}f(v_{k-1}))=\bar{R}<+\infty$. The new assumption indicates that the algorithm can be non-monotone, but the sum of the change of the expected function values at all changing points is finite. Compared to the finite set assumption, the new assumption is weaker and more general. Here, we intuitively argue that the new assumption can be true even when the cardinality of $A$ ($\sharp A$) is very large: First, if the function $f$ is Lipschitz, the new assumption holds if $\sum_{k\in A}\|v_k-v_{k-1}\|<+\infty$. Second, according to Lemma 6 in Appendix C (note that Lemma 6 is independent on the new assumption), we can see that the difference of sequence decays almost as $\|v_k-v_{k-1}\|=O(r)$ (not rigorous). Thus, the new assumption holds if $\sharp A=O(\frac{1}{r})$ with $r$ being the learning rate, which can be very small and thus $\sharp$ can be quite large. We leave the further simplification of this assumption as future work.
> >
> > In the proof, we assumed the boundedness of the gradient, which is a common assumption in the deep learning community.
> >
> > ----
> >
> > Q6. ”The comparison with Adam and RMSProp are limited, while they are the popular algorithms for deep learning models. The authors also claimed that the training time and overfitting can be reduced by half. I think it could be better to see this aspect compared not only to plain SGD but also to Adam and other adaptive algorithms.”
> >
> > Answer: As you mentioned, Adam and other adaptive algorithms use adaptive step size or both adaptive step size and constant momentum. These adaptive step size algorithms can be very fast but usually do not give comparable results as SGD with momentum in training deep neural networks for image classification, see Table 4 for comparison among SRSGD, Adam, and RMSProp. We leave integrating scheduled restart time-varying momentum with adaptive step size as future work.
> >
> > ----
> >
> > Q7. “It is much better if the authors state the theorem results more clearly before using the asymptotic notation. The current bound is not rigorous, and people cannot see how the final bound looks like.”
> >
> > Answer: We have revised these in the updated manuscript. The detailed bounds are provided in the appendix for the sake of readability. Thanks for your suggestion.
> >
> > ----
> >
> > Q8. ”I think the 'Case study - Quadratic function’ in (8) should be written before Figure 2, so that the readers can see the problem you are optimizing easily.”
> >
> > Answer: We have fixed it as you suggested.
> >
> > ==========================
> >
> > We hope we have cleared your concerns about our work. We have also revised our manuscript according to your comments, and we would appreciate it if we can get your further feedback at your earliest convenience.

---

> ### Comment · AnonReviewer4 · 2020-11-22
> **Response to authors**
>
> First, I would like to appreciate the authors’ effort to improve the manuscript, and the amount of work the authors put in the impressive experiments of this paper. I think we all agree that SRSGD shows very promising empirical benefits.
>
> However, at the moment, I still have major theoretical concerns about this paper.
> - I understand that the authors have replaced the finite set assumption. But the new assumption is still too strong, hence the final result is not quite a good contribution. It basically says that the term that contributes to the divergence of the algorithm is bounded. This is very similar to the final result. I see that the authors have some intuition about how to prove this, but it is possible that the term \bar{R} will depend on $k$ (not a constant) and ruin the bound. Hence, at the moment I don’t think this assumption is realistic, unless there is more affirmative proof.
> - My other concern is the result of Theorem 2, where the author proved that $\mathbb{E} (f(w^k) - f(w^*)) \leq C k$. I appreciate that the title of this section has been revised, however, this result conveys no information about the non-convergence or accumulating errors of the algorithm. It is totally possible that some algorithm may converges with rate $\mathbb{E} (f(w^k) - f(w^*)) = O (1/k)$, and then it automatically satisfies the equation $\mathbb{E} (f(w^k) - f(w^*)) \leq C k$ for some constant $C$. Unless a lower bound is provided, I think this theoretical result cannot support your argument that NASGD accumulates errors. And it is a different story if the authors want to show this empirically, it would be harder to convince the readers.
>
> I have read all other replies from the authors and reviewers, and once again, I appreciate the revised effort from the authors. You can be assured that I fully understand the proposed algorithms and have agreed with your other responses. However, my theoretical concerns are not addressed and I will continue discussing with other reviewers before any final decision.

---

> > ### Author Response · Authors · 2020-11-22
> > **Thank you for your comments, and response to your further concerns**
> >
> > Thank you for your very responsible and valuable review, also considering others’ reviews. We appreciate your further feedback. Below we address your further concerns.
> >
> > ---
> >
> > Q1 “I understand that the authors have replaced the finite set assumption. But the new assumption is still too strong, hence the final result is not quite a good contribution. It basically says that the term that contributes to the divergence of the algorithm is bounded. This is very similar to the final result. I see that the authors have some intuition about how to prove this, but it is possible that the term \bar{R} will depend on $k$ (not a constant) and ruin the bound. Hence, at the moment I don’t think this assumption is realistic, unless there is more affirmative proof.”
> >
> > Answer:  The analysis of SRSGD is harder than the analysis of SGD with constant momentum, and our experimental results confirm our convergence results of SRSGD. Regarding your comment on the bounded $\bar{R}$ assumption, as discussed in our previous response, it can be true even when the cardinality of the set $\mathcal{A}$ is of order $O(k)$ with $k$ being the iteration number. We can numerically show that the cardinality of the set $\mathcal{A}$ is much less than $O(k)$ for all the experiments done in our paper.
> >
> > We leave the convergence analysis without bounded $\bar{R}$ assumption as an open problem, and we have commented on this in our revised manuscript.
> >
> > ---
> >
> > Q2. My other concern is the result of Theorem 2, where the author proved that $\mathbb{E} (f(w^k) - f(w^*)) \leq C k$. I appreciate that the title of this section has been revised, however, this result conveys no information about the non-convergence or accumulating errors of the algorithm. It is totally possible that some algorithm may converges with rate $\mathbb{E} (f(w^k) - f(w^*)) = O (1/k)$, and then it automatically satisfies the equation $\mathbb{E} (f(w^k) - f(w^*)) \leq C k$ for some constant $C$. Unless a lower bound is provided, I think this theoretical result cannot support your argument that NASGD accumulates errors. And it is a different story if the authors want to show this empirically, it would be harder to convince the readers.
> >
> > Answer: Thanks for clarifying this.
> >
> > Analyzing error accumulation by upper bound analysis was used in “O. Devolder, F. Glineur, and Y. Nesterov. First-Order Methods of Smooth Convex Optimization with Inexact Oracle. Mathematical Programming volume 146, 37–75(2014)” for NAG when an $\delta$-inexact gradient is used.
> >
> > Theorem 1 shows that the upper bound of $\mathbb{E} (f(w^k) - f(w^*))$ cannot be bounded by a convergent sequence. Based on our numerical results, there may exist cases that $\mathbb{E} (f(w^k) - f(w^*))$ can increase as fast as $Ck$ (see Fig.~3(a) for a numerical example). However, what you pointed out is about the lower bound, which aims to show that for any function $f$, $\mathbb{E} (f(w^k) - f(w^*))$ diverges. We cannot prove the lower bound, which is very difficult. As far as we know, a lower bound has not been established even for the $\delta$-inexact gradient (see O. Devolder, F. Glineur, and Y. Nesterov. First-Order Methods of Smooth Convex Optimization with Inexact Oracle. Mathematical Programming volume 146, 37–75(2014).) We have added a comment on the lower bound in our revised manuscript.
> >
> > ========
> >
> > We hope we have cleared your further concerns about our work. We would appreciate it if we can get your further feedback at your earliest convenience.

---

> ### Author Response · Authors · 2020-11-25
> **Numerical Verification of the assumption in Theorem 2, and some more remarks on the bounded $\bar{R}$ assumption**
>
> We have added a numerical verification of the assumptions in Theorem 2 in our revised manuscript; see Appendix C.1 for details. In particular, we apply SRSGD to train LeNet (We used the PyTorch implementation of LeNet at https://github.com/activatedgeek/LeNet-5.) for MNIST classification (due to extremely large computational cost). We conduct numerical verification as follows: starting from a given point $\mathbf{w}^0$, we randomly sample 469 mini-batches (note in total we have 469 batches in the training data) and compute the stochastic gradient using each mini-batch. Next, we advance to the next step with each of these 469 stochastic gradients and get the approximated $\mathbb{E}f(\mathbf{w}^1)$. We randomly choose one of these 469 positions as the updated weights of our model. By iterating the above procedure, we can get $\mathbf{w}^1, \mathbf{w}^2,\cdots$ and $\mathbb{E}f(\mathbf{w}^1), \mathbb{E}f(\mathbf{w}^2),\cdots$ and we use these values to verify our assumptions in Theorem 2. We set restart frequencies to be 20, 40, and 80, respectively. Figure. 6 (top panels) plot $k$ vs. the cardinality of the set $\mathcal{A} := \{k\in \mathbb{Z}^+| \mathbb{E}f(\mathbf{w}^{k+1})\geq \mathbb{E}f(\mathbf{w}^k) \}$ (the small oscillation leads to the cardinality of $\mathcal{A}$ keep increasing, and the gradually diminishing oscillation is due to the variance in the stochastic gradient),  and  Figure. 6 bottom panels plot $k$ vs. $\sum_{k\in \mathcal{A}}\left(\mathbb{E}f(\mathbf{w}^{k+1}) - \mathbb{E}f(\mathbf{w}^{k}) \right)$. Figure. 6 shows that $\sum_{k\in \mathcal{A}}\left(\mathbb{E}f(\mathbf{w}^{k+1}) - \mathbb{E}f(\mathbf{w}^{k}) \right)$ converges to a constant $\bar{R}<+\infty$. Note that $\bar{R}$ can be bounded even when the cardinality of the set $\mathcal{A}$ grows linearly as pointed out in our previous reply.  These numerical results show that our assumption in Theorem 2 is reasonable.
>
>
>
> NAG is a non-monotone algorithm (as mentioned by Reviewer 5), and this leads to the concern about our assumption in Theorem 2. The non-monotonicity of NAG was interpreted as that the momentum has exceeded a critical value in “B. O’Donoghue, E. Candès. Adaptive restart for accelerated gradient schemes. Foundations of computational mathematics. 2013.” NAG with restart can dramatically reduce oscillation and even almost becomes a monotonic algorithm. See Fig. 3 of “B. O’Donoghue, E. Candès. Adaptive restart for accelerated gradient schemes. Foundations of computational mathematics. 2013.”

---

### Official Review · AnonReviewer1 · 2020-10-27

**Rating:** 5
**Confidence:** 3

**Review:**

The author propose a scheduled restart scheme for SGD with Nesterov momentum named SRSGD, convergence result and various experiments are provided. However, the idea is the combination of restart schedule and NGD, and the convergence result is not that hard to go through with the existence of current proof of SGD.  In specific,

1. the convergence result of SRSGD is O(1/epsilon^2), which is exactly the same with the convergence result of vanilla SGD, this implies that the the NAG momentum does not contribute to the theoretical part. Therefore, it will be easier  to prove the convergence of SRSGD following the the typical proof structure of SGD with some additional error term, e.g.,

"Stochastic First- and Zeroth-order Methods for Nonconvex Stochastic Programming",

"Accelerated gradient methods for nonconvex nonlinear and stochastic programming".

2. I understand the acceleration of convergence may happens in experiments but not in theory, and the main focus of this paper lies on the experiments. However,   although success in experiments,  the intuition why restart scheme can improve generalization error is not convinced to me.  At least, the authors should provide more explanation on this.

Overall, I think the contribution here are not enough to be published in ICLR.

---

> ### Author Response · Authors · 2020-11-13
> **Response to Reviewer 1**
>
> Thank you for your reviews. We have revised our manuscript as you suggested, and the revised part has been highlighted in blue.  Below we address your concerns point-by-point.
>
> ----
>
> Q1. “However, the idea is the combination of restart schedule and NGD, and the convergence result is not that hard to go through with the existence of current proof of SGD.”
>
> Answer: We believe that there is a misunderstanding. Please allow us to clear this misunderstanding by clarifying our major contribution. It is known that the Nesterov accelerated gradient (NAG) with a time-varying momentum parameter, $(k-1)/(k+2)$ with $k$ being the iteration number, can accelerate general convex optimization with a much better convergence rate. Our work focuses on leveraging the idea of NAG with time-varying momentum to improve the training of deep neural networks. Technically, in our paper, we first prove that the time-varying momentum will accumulate error when the stochastic gradient is used. To remedy this issue, we use a very simple idea of a scheduled restart with two specific strategies for momentum scheduling. The resulting algorithm remarkably improves the training of deep neural networks for image classification and provably converges as fast as the existing SGD algorithm. As far as we know, our proposed SRSGD is the first algorithm that makes time-varying momentum outperform the benchmarks by remarkable margins in training various deep neural networks. SRSGD is very simple to use and can be easily applied to a wide range of tasks. Reviewer 2 pointed out that “The motivation is sufficiently original in my opinion and very interesting!”
>
> ----
>
> Q2. “The convergence result of SRSGD is $O(1/\epsilon^2)$, which is exactly the same with the convergence result of vanilla SGD, this implies that the NAG momentum does not contribute to the theoretical part. Therefore, it will be easier to prove the convergence of SRSGD following the typical proof structure of SGD with some additional error term, e.g., "Stochastic First- and Zeroth-order Methods for Nonconvex Stochastic Programming", "Accelerated gradient methods for nonconvex nonlinear and stochastic programming”.”
>
> Answer: The convergence rate proof of our SRSGD has a major difference compared to the convergence rate proof of SGD, which mainly due to the time-varying momentum with a restart. We used an induction technique to bound the sequences’ squared sum, and we used a novel Lyapunov function to get the convergence results. Also, without extra assumption and other information, the known convergence rate for general convex optimization with stochastic gradient is only $O(1/\epsilon)$. In practice, we showed that SRSGD remarkably outperforms the benchmark SGD with constant momentum via a variety of numerical tests. Under which condition SRSGD can get a better theoretical guarantee than SGD with constant momentum is under our investigation.
>
> Thanks for pointing out these two papers to us, which is indeed interesting and helpful. We have added them to our related work and references.
>
> ---
>
> Q3. “I understand the acceleration of convergence may happen in experiments but not in theory, and the main focus of this paper lies on the experiments. However, although success in experiments, the intuition why restart scheme can improve generalization error is not convinced to me. At least, the authors should provide more explanation on this.”
>
> Answer: As we showed in Theorem 1, the time-varying momentum, $(k-1)/(k+2)$, accumulates errors in stochastic/noisy gradients. When we restart less often, it will still accumulate error locally to some degree. Intuitively, the noise accumulation can help escape some bad local minima, which can lead to another local minimum that generalizes better. By modulating the restart schedule, we can give SRSGD a good chance to escape some bad local minima. Theoretical analysis of the generalization behavior of using time-varying momentum with a restart is a very interesting problem, which is under our investigation.
>
> ===========================
>
> We hope we have cleared your concerns about our work. We have also revised our manuscript according to your comments, and we would appreciate it if we can get your further feedback at your earliest convenience.

---

> > ### Comment · AnonReviewer1 · 2020-11-23
> > **Response to Author**
> >
> > Dear authors,
> >
> > Thank you for your responses. I have gone through the responses and the other reviews. However, as I mentioned in the previous paper, I still no convinced about the technical difficult about restart scheme with Nesterov acceleration, and the intuition that why restart scheme improves generalization.
> >
> > Moreover after reading other reviews, I realize that the finite non-monotone assumption is not that suitable as mentioned by review 5 that It is well known that Nesterov acceleration is non-monotone.
> >
> > Overall, I will keep my score at this time.

---

> > > ### Author Response · Authors · 2020-11-23
> > > **Thank you for your reponse, and response to your further concerns**
> > >
> > > Thank you for your reply and for considering others’ reviews. We appreciate your further feedback. Below we address your further concerns.
> > >
> > > ---
> > >
> > > Q1. “I still no convinced about the technical difficult about restart scheme with Nesterov acceleration, and the intuition that why restart scheme improves generalization.”
> > >
> > > Answer: Regarding the difficulty of analyzing Nesterov acceleration with restart and stochastic gradient. There are a few recent papers that consider Nesterov restart with restart under the exact gradient scenario: 1) B. O’Donoghue, E. Candès. Adaptive restart for accelerated gradient schemes. Foundations of computational mathematics. 2013. In this paper, the authors analyzed the convergence of Nesterov accelerated gradient with adaptive restart and exact gradient for quadratic functions. 2) W. Su, S. Boyd, E. Candes. A Differential Equation for Modeling Nesterov's Accelerated Gradient Method: Theory and Insights, JMLR, 2016. In this paper, the authors analyzed the speed restart with exact gradient. 3) V. Roulet, A. d'Aspremont. Sharpness, Restart and Acceleration, NeurIPS, 2017. This is perhaps the one we think offers the strongest theoretical results on scheduled restart and adaptive restart for Nesterov acceleration with exact gradient, and their results further depend on an extra sharpness assumption. We believe these papers are from very top researchers in the community.
> > >
> > > Regarding your concern on why SRSGD can improve generalization, we have not claimed a theoretical guarantee for this in our paper. We only have an intuitive understanding of this, which we have mentioned in our paper and previous reply. Again, since SRSGD with appropriate restart can accumulate error locally, which can help escape some bad local minima (see Fig.~18 for a visualization study).
> > >
> > >
> > > ----
> > >
> > > Q2. “I realize that the finite non-monotone assumption is not that suitable as mentioned by review 5 that It is well known that Nesterov acceleration is non-monotone.”
> > >
> > > Answer: We will provide numerical evidence to show this assumption is reasonable. It will take several more hours to finish these numerical verifications.
> > >
> > > ========
> > >
> > > We hope we have cleared your further concerns about our work. We would appreciate it if we can get your further feedback at your earliest convenience.

---

> > > > ### Author Response · Authors · 2020-11-25
> > > > **Numerical Verification of the assumption in Theorem 2, and some more remarks on the bounded $\bar{R}$ assumption**
> > > >
> > > > We have added a numerical verification of the assumptions in Theorem 2 in our revised manuscript; see Appendix C.1 for details. In particular, we apply SRSGD to train LeNet (We used the PyTorch implementation of LeNet at https://github.com/activatedgeek/LeNet-5.) for MNIST classification (due to extremely large computational cost). We conduct numerical verification as follows: starting from a given point $\mathbf{w}^0$, we randomly sample 469 mini-batches (note in total we have 469 batches in the training data) and compute the stochastic gradient using each mini-batch. Next, we advance to the next step with each of these 469 stochastic gradients and get the approximated $\mathbb{E}f(\mathbf{w}^1)$. We randomly choose one of these 469 positions as the updated weights of our model. By iterating the above procedure, we can get $\mathbf{w}^1, \mathbf{w}^2,\cdots$ and $\mathbb{E}f(\mathbf{w}^1), \mathbb{E}f(\mathbf{w}^2),\cdots$ and we use these values to verify our assumptions in Theorem 2. We set restart frequencies to be 20, 40, and 80, respectively. Figure. 6 (top panels) plot $k$ vs. the cardinality of the set $\mathcal{A} := \{k\in \mathbb{Z}^+| \mathbb{E}f(\mathbf{w}^{k+1})\geq \mathbb{E}f(\mathbf{w}^k) \}$ (the small oscillation leads to the cardinality of $\mathcal{A}$ keep increasing, and the gradually diminishing oscillation is due to the variance in the stochastic gradient),  and  Figure. 6 bottom panels plot $k$ vs. $\sum_{k\in \mathcal{A}}\left(\mathbb{E}f(\mathbf{w}^{k+1}) - \mathbb{E}f(\mathbf{w}^{k}) \right)$. Figure. 6 shows that $\sum_{k\in \mathcal{A}}\left(\mathbb{E}f(\mathbf{w}^{k+1}) - \mathbb{E}f(\mathbf{w}^{k}) \right)$ converges to a constant $\bar{R}<+\infty$. Note that $\bar{R}$ can be bounded even when the cardinality of the set $\mathcal{A}$ grows linearly as pointed out in our previous reply.  These numerical results show that our assumption in Theorem 2 is reasonable.
> > > >
> > > >
> > > >
> > > > NAG is a non-monotone algorithm (as mentioned by Reviewer 5), and this leads to the concern about our assumption in Theorem 2. The non-monotonicity of NAG was interpreted as that the momentum has exceeded a critical value in “B. O’Donoghue, E. Candès. Adaptive restart for accelerated gradient schemes. Foundations of computational mathematics. 2013.” NAG with restart can dramatically reduce oscillation and even almost becomes a monotonic algorithm. See Fig. 3 of “B. O’Donoghue, E. Candès. Adaptive restart for accelerated gradient schemes. Foundations of computational mathematics. 2013.”

---

### Official Review · AnonReviewer2 · 2020-10-28
**Nice idea and many experiments, but stability is under-explored**

**Rating:** 6
**Confidence:** 4

**Review:**

UPDATE:

I find the main contribution of the work to be the empirical analysis. I personally liked this paper, but I must agree with the other reviewers that improving the theoretical results will strengthen the paper's impact. This would require a major revision not suited for a conference rebuttal, and so, for now, I have downgraded my score to a borderline accept, but I look forward to seeing a future revision of this paper.

#### Summary
This work proposes to combine Nesterov momentum (with the Nesterov (1983) $\frac{t_{k}-1}{t_{k+1}}$ momentum schedule for convex functions) and the scheduled restart method (of Roulet and d’Aspremont (2017)) for minimizing finite-sums with stochastic mini-batch gradients. This work theoretically argues that adaptive restarts can prevent the error accumulation identified by Devolder et al. (2017) for $\delta$-inexact gradients.  The stabilizing property is empirically explored (to a small degree) on a Laplacian cycle graph quadratic function with additive Gaussian noise. The most extensive results, which are conducted with mini-batch gradients on CIFAR10 and ImageNet, indicate that adaptive restarts accelerate convergence of deep neural networks.

Adaptive restarts with Nesterov momentum is shown to be very effective for deep learning, hence my strong accept score, though I think there is room for improvement.

---

#### Originality

While adaptive restarts have previously been used to improve the convergence of Nesterov's method, this work focuses on using these adaptive restarts to prevent error accumulation with stochastic gradients. The motivation is sufficiently original in my opinion, and very interesting!

---

#### Quality and Significance

I think this work does not reach its full potential. Acceleration of adaptive restarts has been demonstrated in previous work with deterministic gradients (e.g., Roulet and d’Aspremont (2017)). This work provides empirical evidence of similar behaviour for deep learning with mini-batch gradients, but the purpose (and novelty) in this work is the argument that adaptive restarts stabilize convergence.

*On the results:*
The deep learning experiments are very thorough. One minor observation is that the accuracies in Table 5 are slightly lower than established benchmarks, and fall within the margin of improvement (at least on ResNet-50). It is common to include an additional decay at the 80th epoch (not just 30 and 60) on ImageNet; so I suspect this may be the reason for the slightly weaker baselines.

*Room for improvement*:
The theorem of error accumulation is conducted in the Appendix under a $\delta$-bounded variance assumption. This is an interesting result, but unfortunately wouldn't even hold for least-squares problems when using mini-batch gradients, and so seems disconnected from practical machine learning problems. The assumption of Theorem 2 that the cardinality of $\text{set}( k | E[f(w_{k+1})] \geq E[f(w_{k}] )$ is finite also seems strong. Moreover, the stabilizing hypothesis is only weakly evaluated on a single run of a quadratic and logistic MNIST problem. Extending results to finite-sum settings (i.e., without the bounded variance assumption) would be a great improvement. A reference is missed to [1], where instability of Nesterov's method is demonstrated for smooth strongly-convex finite-sums under interpolation. In Section 5 in that work (Conclusion), they posit that adaptive restarts can be used to stabilize the method to the destabilizing behaviour they identify, but do not explore this direction. There may be possible avenues for extending the stabilizing results in this section to finite-sums

[1] M. Assran and M. Rabbat, "On the Convergence of Nesterov's Accelerated Gradient Method in Stochastic Settings," ICML 2020

---

#### Clarity

Clarity leaves room for improvement. Firstly, it should be clearly stated in the introduction and the abstract that theoretical results in this work considers Nesterov momentum with time-varying momentum (i.e., following Nesterov (1983)). Assumptions for the theorems should also be stated in the main text (not buried in the appendix), e.g., the $\delta$-bounded variance assumption, and the update rule for the momentum parameter.

Note, that NAG can be run with time-varying momentum, which as indicated achieves the optimal rate on convex functions, or it can be run with constant momentum (Nesterov, 2004), which achieves the optimal rate for strongly-convex functions. Therefore, the subtle point that NAG is given by equation (3), but with time-varying momentum, is not good practice in my opinion.

The following claim is not true in general "Since NAG momentum achieves a much better convergence rate than constant momentum with exact gradient, we consider the following question:”  this holds to the best of my knowledge for convex functions, but NAG achieves the optimal convergence rate with constant momentum for strongly-convex function (Nesterov, 2004).

Please consider rewording "SGD" to "HB-SGD" since you use it to refer to SGD with heavy-ball momentum. Also consider renaming the proposed momentum restarted method since it does not indicate that momentum is used.

---

> ### Author Response · Authors · 2020-11-13
> **Response to Reviewer 2**
>
> Thank you for your reviews. We have revised our manuscript as you suggested, and the revised part is highlighted in blue. Below we address your concerns point-by-point.
>
> ----
>
> Q1. “One minor observation is that the accuracies in Table 5 are slightly lower than established benchmarks, and fall within the margin of improvement (at least on ResNet-50). It is common to include an additional decay at the 80th epoch (not just 30 and 60) on ImageNet; so I suspect this may be the reason for the slightly weaker baselines.”
>
> Answer: As you suggested, we have launched the ImageNet experiments on ResNet-50 with SGD and SRSGD using an additional decay at the 80th epoch. Our proposed SRSGD still outperforms SGD in this setting. In particular, SRSGD achieves a single crop top-1 validation error of $23.64 \pm 0.12$ \%, while SGD achieves a single crop top-1 validation error of $23.80 \pm 0.14$ \%. All experiments are averaged over 3 runs due to the limited amount of computing resources and time.
>
> ----
>
> Q2. “The theorem of error accumulation is conducted in the Appendix under a delta-bounded variance assumption. This is an interesting result, but unfortunately wouldn't even hold for least-squares problems when using mini-batch gradients, and so seems disconnected from practical machine learning problems. Extending results to finite-sum settings (i.e., without the bounded variance assumption) would be a great improvement.”
>
> Answer: The delta-bounded variance is quite common in analyzing stochastic gradient optimization algorithms, as can be seen, for instance, in “S. Bubeck. Convex optimization: Algorithms and complexity. Foundations and Trends in Machine Learning, 8(3-4):231–357, 2015.” “L. Bottou, F. E. Curtis, and J. Nocedal. Optimization methods for large-scale machine learning. arxiv:1606.04838, 2016.” In the revised manuscript, we have clearly stated all the assumptions that we made in our analysis and replaced the words of mini-batch SGD with the delta-bounded variance assumption.
>
> Thanks for pointing out this direction, and eliminating the assumption of the bounded variance in the analysis is under our investigation.
>
> ----
>
> Q3. “The assumption of Theorem 2 that the cardinality of set $\{k|E[f(w_{k+1})]\geq E[f(w_k)]\}$ is finite also seems strong. Moreover, the stabilizing hypothesis is only weakly evaluated on a single run of a quadratic and logistic MNIST problem.”
>
> Answer: We can directly assume that $\sum_{k\in A}(\mathbb{E}f(v_k)-\mathbb{E}f(v_{k-1}))=\bar{R}<+\infty
> $. This assumption is weaker than the finite set assumption. Here, we intuitive argue that the new assumption can be true even when the cardinality of $A$ ($\sharp A$) is very large: First, if the function $f$ is Lipschitz, the new assumption holds if $\sum_{k\in A}\|v_k-v_{k-1}\|<+\infty$. Second, according to Lemma 6 in Appendix C (note that Lemma 6 is independent on the new assumption), we can see that the difference of sequence decays almost as $\|v_k-v_{k-1}\|=O(r)$ (not rigorous). Thus, the new assumption holds if $\sharp A=O(\frac{1}{r})$ with $r$ being the learning rate, which can be very small and thus $\sharp $ can be quite large.
>
> In the case studies discussed in our paper, we verified the stabilizing effect of scheduled restart on a quadratic and logistic MNIST problem. However, the stabilizing effect of scheduled restart has also been thoroughly verified in all the following experiments on training deep neural networks; without restart, the loss of training deep neural networks will go to a very large number and blow up very quickly. We have made this point clear in the revised manuscript.
>
> ----
>
> Q4. “A reference is missed to [1], where instability of Nesterov's method is demonstrated for smooth strongly-convex finite-sums under interpolation. In Section 5 in that work (Conclusion), they posit that adaptive restarts can be used to stabilize the method to the destabilizing behaviour they identify, but do not explore this direction. There may be possible avenues for extending the stabilizing results in this section to finite-sums.”
>
> Answer: Thanks for pointing out this very interesting and important reference to us, and we have added this paper to the Introduction section of our paper. Indeed, adaptive restart is a promising idea, and we tried this approach when we started our work. The problem we encountered was that the adaptive restart made momentum to restart too often and become almost always zero. Thus, we switched to the scheduled restart approach, as presented in our paper. Making adaptive restart works for large scale machine learning problems is certainly an interesting avenue for us.

---

> > ### Author Response · Authors · 2020-11-13
> > **Response to Reviewer 2 -- Cont'd**
> >
> > Q5. “It should be clearly stated in the introduction and the abstract that theoretical results in this work considers Nesterov momentum with time-varying momentum (i.e., following Nesterov (1983)). Assumptions for the theorems should also be stated in the main text (not buried in the appendix), e.g., the $\delta$-bounded variance assumption, and the update rule for the momentum parameter.”
> >
> > Answer: We have revised our manuscript as you suggested, and this indeed helps improve the readability of our paper significantly.
> >
> > ---
> >
> > Q6. “NAG can be run with time-varying momentum, which as indicated achieves the optimal rate on convex functions, or it can be run with constant momentum (Nesterov, 2004), which achieves the optimal rate for strongly-convex functions. Therefore, the subtle point that NAG is given by equation (3), but with time-varying momentum, is not good practice in my opinion.”
> >
> > Answer: Thanks for pointing this out, and we have clarified this point in the updated manuscript according to your suggestion.
> >
> > ---
> >
> > Q7. “The following claim is not true in general "Since NAG momentum achieves a much better convergence rate than constant momentum with exact gradient, we consider the following question:” this holds to the best of my knowledge for convex functions, but NAG achieves the optimal convergence rate with constant momentum for strongly-convex function (Nesterov, 2004).”
> >
> > Answer: We have corrected that statement in the revision.
> >
> > ----
> >
> > Q8. “Please consider rewording "SGD" to "HB-SGD" since you use it to refer to SGD with heavy-ball momentum. Also, consider renaming the proposed momentum restarted method since it does not indicate that momentum is used.”
> >
> > Answer: As you suggested, we have clearly pointed out the time-varying momentum that we restarted at the very beginning of the proposed restart method. Regarding rename "SGD " to "HB-SGD", which may lead to lots of other changes. To avoid confusion and making potential mistakes during the rebuttal period, we clearly stated the meaning of SGD in our revision. Later, we will rename it systematically later.
> >
> >
> > ===========
> >
> > We hope we have cleared your concerns about our work. We have also revised our manuscript according to your comments, and we would appreciate it if we can get your further feedback at your earliest convenience.

---

> ### Comment · AnonReviewer2 · 2020-11-21
> **Thank you for your comments**
>
> Dear authors,
>
> Thank you for your responses. I have gone through the changes and also read the other reviews. Here are my thoughts:
>
> On Q1. Thank you for running the ImageNet results with the updated schedule, these new numbers are more consistent with existing benchmarks.
>
> On Q2.
> I am aware that the delta-bounded assumption is common, but my point is that it wouldn’t generally hold in the finite-sum setting (not even for simple least-squares), which is the setting in which the restarted-scheme is proposed. AnonReviewer3 raised a similar point so I'm curious to hear their thoughts. Generally, this is not a major issue for me, but I don’t feel this point was addressed.
>
> On Q4.
> This was my biggest issue, and I am happy with how it was addressed. My main concern was that the restarting scheme was proposed to stabilize convergence with stochastic gradients (which I found interesting and expected to be true), but was hardly empirically verified. The fact that the experiments diverge without the restarts is sufficient empirical evidence to satisfy this concern, but I would still appreciate it if you could show a figure or table to really drive this point home (e.g., I would expect the loss to converge at first, and then blow-up at some point during training without the restart scheme).
>
> On Q3.
> I appreciate you moving the assumptions into the main paper from the appendix, and for weakening it slightly. However, I still find the updated assumption $\sum_{k \in A} \mathbb{E}f^k - \mathbb{E}f^{k-1} < \infty$ to be quite strong. I am curious to see if other reviewers (AnonReviewer5 and AnonReviewer4) feel satisfied by this update, but, in short, while the updated assumption is slightly weaker, it essentially has a very similar implication. However, this is still not a major issue for me.
>
> Concern:
> My only remaining concern is the theoretical issue pointed out by AnonReviewer 4. Indeed, on second glance, the result that $f^k-f^\star = \mathcal{O}(k)$ is somewhat uninformative. Instead, a lower bound must be shown to prove that the method can exhibit instability with the time-varying momentum under the delta-bounded variance assumption. If I recall correctly, Devolder et al., 2014 show upper bounds for illustrating the impact of convergence rate on error accumulation, but they also prove lower bounds on the rate of error accumulation. In order to prove that the restart scheme stabilizes convergence, I would also expect this lower bound to be proven under similar set of assumptions to the convergence guarantees with restarts.
>
> In conclusion, I find the empirical results to be very good, and I really like the idea behind the paper. My only remaining concern is in the theoretical contribution. As long as this theory is not misleading or erroneous, I would be ok with accepting this paper without stronger guarantees.

---

> > ### Author Response · Authors · 2020-11-22
> > **Thank you for your comments, and response to your further concerns**
> >
> > Thank you for your very responsible and valuable review, also considering others’ reviews. We appreciate your further feedback. Below we address your further concerns.
> >
> > ---
> >
> > Q1. “I am aware that the delta-bounded assumption is common, but my point is that it wouldn’t generally hold in the finite-sum setting, which is the setting in which the restarted-scheme is proposed. This is not a major issue for me, but I don’t feel this point was addressed.”
> >
> > Answer: Thanks for pointing out this again. We have restated Theorem 1 and stated the assumption of bounded variance stochastic gradient instead of the mini-batch gradient in our revision.
> >
> > ---
> >
> > Q2. “I would still appreciate it if you could show a figure or table to really drive this point home (e.g., I would expect the loss to converge at first, and then blow-up at some point during training without the restart scheme).”
> >
> > Answer: As you suggested, we have provided empirical verification on error accumulation in our revised manuscript. In particular, Fig. 5 and Fig. 14 show that NASGD (in red) accumulates error in training Pre-ResNet-290 for CIFAR10 and CIFAR100 classification, resp. The training losses of NASGD in both cases fail to converge to good values, which is then reflected in the bad test accuracies of the models trained with NASGD. These plots are averaged over 5 runs.
> >
> > ---
> >
> > Q3. “I still find the updated assumption $\sum_{k\in A}Ef_k-Ef_{k-1}<\infty$ to be quite strong.”
> >
> > Answer: The analysis of SRSGD is harder than the analysis of SGD with constant momentum, and our experiments confirm our convergence results of SRSGD. Regarding your comment on the bounded $\bar{R}$ assumption, as discussed in our previous response, it can be true even when the cardinality of the set $\mathcal{A}$ is of order $O(k)$ with $k$ being the iteration number. We can numerically show that the cardinality of the set $\mathcal{A}$ is much less than $O(k)$ for all the experiments we have done in our paper.
> >
> > ---
> >
> > Q4. “My only remaining concern is the theoretical issue pointed out by Reviewer 4. Indeed, on second glance, the result that $f_k-f_*=O(k)$ is completely uninformative. Instead, a lower bound must be shown to prove that the method can exhibit instability with the time-varying momentum under the delta-bounded variance assumption. In order to prove that the restart scheme stabilizes convergence, I would also expect this lower bound to be proven under a similar set of assumptions to the convergence guarantees with restarts.”
> >
> > Answer: Let us clarify this point, and we believe there is a misunderstanding of our purpose of Theorem 1 by Reviewer 4. What we want to show is that the upper bound of $\mathbb{E} (f(w^k) - f(w^*))$ cannot be bounded by a convergent sequence. Based on our numerical results, there may exist cases that $\mathbb{E} (f(w^k) - f(w^*))$ increases as fast as $Ck$ (see Fig.~3(a) for a numerical example). However, what  AnonReviewer 4 pointed out is about the lower bound, which aims to show that for any function $f$, $\mathbb{E} (f(w^k) - f(w^*))$ diverges. Let us elaborate on this below.
> >
> > First, analyzing error accumulation by upper bound analysis was used in “O. Devolder, et al. 2014” for NAG when an $\delta$-inexact gradient is used.
> >
> > Second, we named Theorem 1 as “Uncontrolled bound of NASGD” to emphasize that $\mathbb{E} (f(w^k) - f(w^*))$ cannot be controlled by a convergent sequence. The theoretical results in Theorem 1 aim to convey that the bound of the NASGD can be out of control. Numerically, we show the divergence of training a logistic regression model for the MNIST classification task. Our proof helps explain the numerically observed divergence phenomenon. In particular, from our analysis, we can see that the uncontrollable error bound is due to the fact that the time-varying momentum gets closer and closer to 1 as iteration goes. Based on this analysis, we apply the restart scheme, which makes the momentum parameter smaller than a constant that is less than 1. The uncontrollable bound result is consistent with the analysis under the delta-inexact gradient scenario in Devolder et al. 2014.
> >
> > Third, the lower bound in this case is very hard to establish. We have added a comment on the lower bound in our revision. As far as we know, even under the $\delta$-inexact gradient assumption, the lower bound of $\mathbb{E} (f(w^k) - f(w^*))$ has not been established in the literature yet. The results in Sec. 8.3 of “ Devolder et al. 2014” seem different from the lower bound of  $\mathbb{E} (f(w^k) - f(w^*))$, which discusses when a first-order method will accumulate error. Please correct us if we misunderstood this.
> >
> > ===
> >
> > We hope we have cleared your further concerns about our work. We have also revised our manuscript according to your comments, and we would appreciate it if we can get your further feedback at your earliest convenience.

---

> > > ### Author Response · Authors · 2020-11-25
> > > **Numerical Verification of the assumption in Theorem 2, and some more remarks on the bounded $\bar{R}$ assumption**
> > >
> > > We have added a numerical verification of the assumptions in Theorem 2 in our revised manuscript; see Appendix C.1 for details. In particular, we apply SRSGD to train LeNet (We used the PyTorch implementation of LeNet at https://github.com/activatedgeek/LeNet-5.) for MNIST classification (due to extremely large computational cost). We conduct numerical verification as follows: starting from a given point $\mathbf{w}^0$, we randomly sample 469 mini-batches (note in total we have 469 batches in the training data) and compute the stochastic gradient using each mini-batch. Next, we advance to the next step with each of these 469 stochastic gradients and get the approximated $\mathbb{E}f(\mathbf{w}^1)$. We randomly choose one of these 469 positions as the updated weights of our model. By iterating the above procedure, we can get $\mathbf{w}^1, \mathbf{w}^2,\cdots$ and $\mathbb{E}f(\mathbf{w}^1), \mathbb{E}f(\mathbf{w}^2),\cdots$ and we use these values to verify our assumptions in Theorem 2. We set restart frequencies to be 20, 40, and 80, respectively. Figure. 6 (top panels) plot $k$ vs. the cardinality of the set $\mathcal{A} := \{k\in \mathbb{Z}^+| \mathbb{E}f(\mathbf{w}^{k+1})\geq \mathbb{E}f(\mathbf{w}^k) \}$ (the small oscillation leads to the cardinality of $\mathcal{A}$ keep increasing, and the gradually diminishing oscillation is due to the variance in the stochastic gradient),  and  Figure. 6 bottom panels plot $k$ vs. $\sum_{k\in \mathcal{A}}\left(\mathbb{E}f(\mathbf{w}^{k+1}) - \mathbb{E}f(\mathbf{w}^{k}) \right)$. Figure. 6 shows that $\sum_{k\in \mathcal{A}}\left(\mathbb{E}f(\mathbf{w}^{k+1}) - \mathbb{E}f(\mathbf{w}^{k}) \right)$ converges to a constant $\bar{R}<+\infty$. Note that $\bar{R}$ can be bounded even when the cardinality of the set $\mathcal{A}$ grows linearly as pointed out in our previous reply.  These numerical results show that our assumption in Theorem 2 is reasonable.
> > >
> > >
> > >
> > > NAG is a non-monotone algorithm (as mentioned by Reviewer 5), and this leads to the concern about our assumption in Theorem 2. The non-monotonicity of NAG was interpreted as that the momentum has exceeded a critical value in “B. O’Donoghue, E. Candès. Adaptive restart for accelerated gradient schemes. Foundations of computational mathematics. 2013.” NAG with restart can dramatically reduce oscillation and even almost becomes a monotonic algorithm. See Fig. 3 of “B. O’Donoghue, E. Candès. Adaptive restart for accelerated gradient schemes. Foundations of computational mathematics. 2013.”

---

### Official Review · AnonReviewer5 · 2020-11-08
**New insight into applying momentum restart in stochastic optimization; concerns regarding theoretical contribution**

**Rating:** 5
**Confidence:** 4

**Review:**

This paper focuses on Nesterov's Accelerated Gradient (NAG) method with a scheduled restart for stochastic optimization. The authors particularly provide a particular momentum restart scheme and characterize its convergence for both nonconvex and convex objective functions. Furthermore, various experiments have been conducted to highlight the strength of their algorithm.

Overall, the paper is well-written and provides new insights into the benefits of using momentum restart in accelerated methods. That said, I believe the authors could improve the related work section to add further discussions on previous works on this topic, especially in the context of stochastic optimization. For instance, [1] has also developed a multistage variant of NAG with momentum restart between stages. The idea of applying restart techniques to stochastic methods is studied in [2] as well.

My major concern is regarding the theoretical contribution on this paper.

First, I encourage the authors to clearly state the assumptions that they are making to obtain their results such as Theorem 2. For instance, looking into the details of their analysis (Lemma 6 and Theorem 5 in Section C of the appendix), it seems that the authors are assuming the gradients are bounded (by $R$). In addition, I am suspicious the authors are somehow assuming the function $f$ is bounded when they bound
$\sum_{k \in \mathcal{A}}\left(\mathbb{E} f\left({v}^{k}\right)-\mathbb{E} f\left({v}^{k-1}\right)\right)=\bar{R}<+\infty$
in the proof of Theorem 5. Anyhow, these assumptions should be stated upfront so that the readers could compare these results with other works.

Second, I am not sure how much assumptions such as
$\mathcal{A}:=$ { $k \in \mathbb{Z}^{+} \mid \mathbb{E} f\left({w}^{k+1}\right) \geq \mathbb{E} f\left({w}^{k}\right)$}
being finite are realistic. Of course for finite number of iterations $\mathcal{A}$ is finite, but as the proof shows, its growth should be constant relative to $K$. However, Nesterov's method is known to be non-monotone, and hence, this is not a straightforward assumption to make (see Figure 1 in [3] for instance). A discussion regarding this assumption (and the other similar assumption for the convex case) is appreciated.

References

[1] Necdet Serhat Aybat, Alireza Fallah, Mert Gurbuzbalaban, and Asuman Ozdaglar. A universally optimal multistage accelerated sto chastic gradient metho d. In Advances in Neural Information Processing Systems, pages 8525 8536, 2019.

[2] Andrei Kulunchakov and Julien Mairal. A generic acceleration framework for sto chastic comp osite optimization. In Advances in Neural Information Process- ing Systems, pages 1255612567, 2019.

[3] Weijie Su, Stephen Boyd, and Emmanuel J Candes. A dierential equation for mo deling nesterov's accelerated gradient metho d: theory and insights. The Journal of Machine Learning Research, 17(1):53125354, 2016.

---

> ### Author Response · Authors · 2020-11-13
> **Response to Reviewer 5**
>
> Thank you for your reviews. We have revised our manuscript as you suggested, and the revised part has been highlighted in blue. Below we address your concerns point-by-point.
>
> ----
>
> Q1. “I believe the authors could improve the related work section to add further discussions on previous works on this topic, especially in the context of stochastic optimization. For instance, [1] has also developed a multistage variant of NAG with momentum restart between stages. The idea of applying restart techniques to stochastic methods is studied in [2] as well.”
>
> Answer: Thanks for pointing out these very exciting papers to us. We have added these references to our revised manuscript, and the revised part is highlighted in blue.
>
> ---
>
> Q2. “I encourage the authors to clearly state the assumptions that they are making to obtain their results such as Theorem 2. For instance, looking into the details of their analysis (Lemma 6 and Theorem 5 in Section C of the appendix), it seems that the authors are assuming the gradients are bounded (by R). In addition, I am suspicious the authors are somehow assuming the function f is bounded when they bound $\sum_{k\in A}(\mathbb{E}f(v_k)-\mathbb{E}f(v_{k-1}))=\bar{R}<+\infty$ in the proof of Theorem 5. Anyhow, these assumptions should be stated upfront so that the readers could compare these results with other works.”
>
> Answer: We agree with your comment that the boundedness of function values can yield the uniform bound $\sum_{k\in A}(\mathbb{E}f(v_k)-\mathbb{E}f(v_{k-1}))=\bar{R}<+\infty.$ According to your comments and the comments from the other reviewers, we can combine the previous two assumptions into the following weaker assumption
> $$
> \sum_{k\in A}(\mathbb{E}f(v_k)-\mathbb{E}f(v_{k-1}))=\bar{R}<+\infty.
> $$
> Here, we intuitive argue that the new assumption can be true even when the cardinality of $A$ ($\sharp A$) is very large: First, if the function $f$ is Lipschitz, the new assumption holds if $\sum_{k\in A}\|v_k-v_{k-1}\|<+\infty$. Second, according to Lemma 6 in Appendix C (note that Lemma 6 is independent on the new assumption), we can see that the difference of sequence decays almost as $\|v_k-v_{k-1}\|=O(r)$ (not rigorous). Thus, the new assumption holds if $\sharp A=O(\frac{1}{r})$ with $r$ being the learning rate, which can be very small and thus $\sharp$ can be quite large.
>
> As you suggested, we have stated the assumptions clearly in our theorem in the revised manuscript.
>
> ---
>
> Q3. “I am not sure how much assumptions such as $A:=\{k\in \mathbb{Z}^+|\mathbb{E}f(w^{k+1})\geq \mathbb{E}f(w^k) \}$ being finite are realistic. However, Nesterov's method is known to be non-monotone, and hence, this is not a straightforward assumption to make. A discussion regarding this assumption (and the other similar assumption for the convex case) is appreciated.”
>
> Answer: Again, we can replace this finiteness assumption by
>
> $$
> \sum_{k\in A}(\mathbb{E}f(v_k)-\mathbb{E}f(v_{k-1}))=\bar{R}<+\infty,
> $$
>
> This new assumption indicates that the algorithm can be non-monotone, but the sum of function values changes at the changing point is finite.
>
> =====
>
> We hope we have cleared your concerns about our work. We have also revised our manuscript according to your comments, and we would appreciate it if we can get your further feedback at your earliest convenience.

---

> > ### Comment · AnonReviewer5 · 2020-11-22
> > **Response to authors**
> >
> > First and foremost, I would like to thank the authors for revising the manuscript. I reviewed the changes as well as other reviews' comments. I also appreciate that the authors moved the assumptions to the body of the paper and made it more evident for the readers.
> >
> > Overall I believe the manuscript has improved during the rebuttal. However, I am still not convinced that my main concern regarding the assumption of the function value increase's boundedness has been fully addressed as the new assumption still seems pretty strong.  In fact, it somehow assumes the term that causes divergence in the analysis is bounded automatically. As a consequence, it looks like part of the result that the authors are trying to show is assumed to be true in advance.
> >
> > Again, I want to highlight that the insights provided in this paper, especially in experiments, are definitely interesting to the community.  However, at this point, my main concerns, mostly related to the theoretical aspect of the paper, are not addressed. Nevertheless, I am looking forward to discussing with other reviewers and reading further replies before the final decision.

---

> > > ### Author Response · Authors · 2020-11-22
> > > **Thank you for your comments, and response to your further concerns**
> > >
> > > Thank you for your very responsible and valuable review, also considering others’ reviews. We appreciate your further feedback. Below we address your further concerns.
> > >
> > > Q1. “I am still not convinced that my main concern regarding the assumption of the function value increase's boundedness has been fully addressed as the new assumption still seems pretty strong.  In fact, it somehow assumes the term that causes divergence in the analysis is bounded automatically. As a consequence, it looks like part of the result that the authors are trying to show is assumed to be true in advance.”
> > >
> > > Answer: The analysis of SRSGD is harder than the analysis of SGD with constant momentum, and our experimental results confirm our convergence results of SRSGD. Regarding your comment on the bounded $\bar{R}$ assumption, as discussed in our previous response, it can be true even when the cardinality of the set $\mathcal{A}$ is of order $O(k)$ with $k$ being the iteration number. We can numerically show that the cardinality of the set $\mathcal{A}$ is much less than $O(k)$ for all the experiments done in our paper.
> > >
> > > We leave the convergence analysis without bounded $\bar{R}$ assumption as an open problem, and we have commented on this in our revised manuscript.
> > >
> > > =====
> > >
> > > We hope we have cleared your further concerns about our work. We would appreciate it if we can get your further feedback at your earliest convenience.

---

> > > > ### Author Response · Authors · 2020-11-25
> > > > **Numerical Verification of the assumption in Theorem 2, and some more remarks on the bounded $\bar{R}$ assumption**
> > > >
> > > > We have added a numerical verification of the assumptions in Theorem 2 in our revised manuscript; see Appendix C.1 for details. In particular, we apply SRSGD to train LeNet (We used the PyTorch implementation of LeNet at https://github.com/activatedgeek/LeNet-5.) for MNIST classification (due to extremely large computational cost). We conduct numerical verification as follows: starting from a given point $\mathbf{w}^0$, we randomly sample 469 mini-batches (note in total we have 469 batches in the training data) and compute the stochastic gradient using each mini-batch. Next, we advance to the next step with each of these 469 stochastic gradients and get the approximated $\mathbb{E}f(\mathbf{w}^1)$. We randomly choose one of these 469 positions as the updated weights of our model. By iterating the above procedure, we can get $\mathbf{w}^1, \mathbf{w}^2,\cdots$ and $\mathbb{E}f(\mathbf{w}^1), \mathbb{E}f(\mathbf{w}^2),\cdots$ and we use these values to verify our assumptions in Theorem 2. We set restart frequencies to be 20, 40, and 80, respectively. Figure. 6 (top panels) plot $k$ vs. the cardinality of the set $\mathcal{A} := \{k\in \mathbb{Z}^+| \mathbb{E}f(\mathbf{w}^{k+1})\geq \mathbb{E}f(\mathbf{w}^k) \}$ (the small oscillation leads to the cardinality of $\mathcal{A}$ keep increasing, and the gradually diminishing oscillation is due to the variance in the stochastic gradient),  and  Figure. 6 bottom panels plot $k$ vs. $\sum_{k\in \mathcal{A}}\left(\mathbb{E}f(\mathbf{w}^{k+1}) - \mathbb{E}f(\mathbf{w}^{k}) \right)$. Figure. 6 shows that $\sum_{k\in \mathcal{A}}\left(\mathbb{E}f(\mathbf{w}^{k+1}) - \mathbb{E}f(\mathbf{w}^{k}) \right)$ converges to a constant $\bar{R}<+\infty$. Note that $\bar{R}$ can be bounded even when the cardinality of the set $\mathcal{A}$ grows linearly as pointed out in our previous reply.  These numerical results show that our assumption in Theorem 2 is reasonable.
> > > >
> > > >
> > > >
> > > > NAG is a non-monotone algorithm, and this leads to the concern about our assumption in Theorem 2. The non-monotonicity of NAG was interpreted as that the momentum has exceeded a critical value in “B. O’Donoghue, E. Candès. Adaptive restart for accelerated gradient schemes. Foundations of computational mathematics. 2013.” NAG with restart can dramatically reduce oscillation and even almost becomes a monotonic algorithm. See Fig. 3 of “B. O’Donoghue, E. Candès. Adaptive restart for accelerated gradient schemes. Foundations of computational mathematics. 2013.”

---

### Author Response · Authors · 2020-11-25
**Numerical Verification of the assumption in Theorem 2, and some more remarks on the bounded $\bar{R}$ assumption**

We have added a numerical verification of the assumptions in Theorem 2 in our revised manuscript; see Appendix C.1 for details. In particular, we apply SRSGD to train LeNet (We used the PyTorch implementation of LeNet at https://github.com/activatedgeek/LeNet-5.) for MNIST classification (due to extremely large computational cost). We conduct numerical verification as follows: starting from a given point $\mathbf{w}^0$, we randomly sample 469 mini-batches (note in total we have 469 batches in the training data) and compute the stochastic gradient using each mini-batch. Next, we advance to the next step with each of these 469 stochastic gradients and get the approximated $\mathbb{E}f(\mathbf{w}^1)$. We randomly choose one of these 469 positions as the updated weights of our model. By iterating the above procedure, we can get $\mathbf{w}^1, \mathbf{w}^2,\cdots$ and $\mathbb{E}f(\mathbf{w}^1), \mathbb{E}f(\mathbf{w}^2),\cdots$ and we use these values to verify our assumptions in Theorem 2. We set restart frequencies to be 20, 40, and 80, respectively. Figure. 6 (top panels) plot $k$ vs. the cardinality of the set $\mathcal{A} := \{k\in \mathbb{Z}^+| \mathbb{E}f(\mathbf{w}^{k+1})\geq \mathbb{E}f(\mathbf{w}^k) \}$ (the small oscillation leads to the cardinality of $\mathcal{A}$ keep increasing, and the gradually diminishing oscillation is due to the variance in the stochastic gradient),  and  Figure. 6 bottom panels plot $k$ vs. $\sum_{k\in \mathcal{A}}\left(\mathbb{E}f(\mathbf{w}^{k+1}) - \mathbb{E}f(\mathbf{w}^{k}) \right)$. Figure. 6 shows that $\sum_{k\in \mathcal{A}}\left(\mathbb{E}f(\mathbf{w}^{k+1}) - \mathbb{E}f(\mathbf{w}^{k}) \right)$ converges to a constant $\bar{R}<+\infty$. Note that $\bar{R}$ can be bounded even when the cardinality of the set $\mathcal{A}$ grows linearly as pointed out in our previous reply.  These numerical results show that our assumption in Theorem 2 is reasonable.



NAG is a non-monotone algorithm (as mentioned by Reviewer 5), and this leads to the concern about our assumption in Theorem 2. The non-monotonicity of NAG was interpreted as that the momentum has exceeded a critical value in “B. O’Donoghue, E. Candès. Adaptive restart for accelerated gradient schemes. Foundations of computational mathematics. 2013.” NAG with restart can dramatically reduce oscillation and even almost becomes a monotonic algorithm. See Fig. 3 of “B. O’Donoghue, E. Candès. Adaptive restart for accelerated gradient schemes. Foundations of computational mathematics. 2013.”

---

> ### Author Response · Authors · 2020-11-25
> **Some further clarifications of our paper**
>
> First, we want to emphasize our contributions in this paper again; see the “Contributions” section on page 2. In a nutshell, our contributions are about developing a simple and effective algorithm based on NAG with time-varying momentum to improve training neural networks. The proposed SRSGD can remarkably improve training neural networks and was verified by a large number of benchmark experiments. SRSGD is different from existing stochastic algorithms in PyTorch, and our work is not about analyzing an existing algorithm.
>
> Second, our upper bound analysis of NASGD (Theorem 1) is to show that Nesterov acceleration will accumulate error when stochastic gradients are used. The upper bound analysis was used in “O. Devolder, F. Glineur, and Y. Nesterov. First-Order Methods of Smooth Convex Optimization with Inexact Oracle. Mathematical Programming volume 146, 37–75(2014)”, in which the authors proved error accumulation of NAG when $\delta$-inexact gradient is used. Our argument is about stochastic gradient, and our result is backed up with enormous numerical evidence to support this error accumulation result.
>
> Third, Theorem 2 provides convergence results for SRSGD under certain assumptions. We have clearly stated our assumptions in our revision. Regarding these assumptions, we have provided some numerical evidence to show they are reasonable.
>
> Finally, we thank the reviewers for their thoughtful review and valuable feedback. Also, we thank the area chair for reviewing and handling our paper.

---

### Decision · Program_Chairs · 2021-01-07
**Final Decision**

**Decision:**

Reject

**Comment:**

The reviewers acknowledge that the paper has some promising experiments. However, they think that the theoretical contributions are not rigorous, specifically the assumption in Theorem 2. It is true that the main part of the proof relies on this assumption. The main question is whether this assumption holds or not. The way that the authors provide some numerical verification is not convincing and does not necessarily help in this situation:

1) MNIST data set with LeNet is definitely not enough to represent all situations.
2) This assumption depends on the algorithm so the parameter choices are also very important.

Some reviewers and I agree that this assumption may hold in some scenarios, but assuming it (without proving it) would significantly reduce the contribution of the paper.

The following is the suggestion to improve the paper. Since this assumption is not standard and hard to verify, the authors should verify more experiments with various data sets and network architectures with difference choices of the algorithm parameters to have some sense whether this assumption may be true or not. Next, please try to show that this assumption holds or come up with different analysis with more reasonable assumptions.

The authors should consider to improve the theory to strengthen the paper and resubmit this paper in the future venues.